# Aerosol-mid-latitude cyclone indirect effects in observations and high-resolution simulations

Daniel T. McCoy[1], Paul R. Field[1,4], Anja Schmidt[1,2,3], Daniel P. Grosvenor[1], Frida A.-M. Bender[5], Ben J. Shipway[4], Adrian A. Hill[4], Jonathan M. Wilkinson[4], Gregory S. Elsaesser[6]

[1]University of Leeds, Leeds LS2 9JT, UK
[2]Department of Chemistry, University of Cambridge, Cambridge, CB2 1EW, UK
[3]Department of Geography, University of Cambridge, Downing Place, Cambridge CB2 3EN, UK
[4]Met Office, Fitzroy Rd, Exeter EX1 3PB, UK
[5]University of Stockholm, Svante Arrhenius Väg 16C, Sweden
[6]Department of Applied Physics and Mathematics, Columbia University and NASA Goddard Institute for Space Studies, New York, NY, USA

*Correspondence to*: Daniel T. McCoy (D.T.McCoy@leeds.ac.uk)

**Abstract.** Aerosol-cloud interactions are a major source of uncertainty in inferring the climate sensitivity from the observational record of temperature. The adjustment of clouds to aerosol is a poorly constrained aspect of these aerosol-cloud interactions. Here, we examine the response of midlatitude cyclone cloud properties to a change in cloud droplet number concentration (CDNC). Idealized experiments in high-resolution, convection-permitting global aquaplanet simulations with constant CDNC are compared to thirteen years of remote-sensing observations. Observations and idealized aquaplanet simulations agree that increased warm conveyor belt (WCB) moisture flux into cyclones is consistent with higher cyclone liquid water path (CLWP). When CDNC is increased a larger LWP is needed to give the same rain rate. The LWP adjusts to allow the rain rate to be equal to the moisture flux into the cyclone along the warm conveyor belt. This results in an increased CLWP for higher CDNC at a fixed WCB moisture flux in both observations and simulations. If observed cyclones in the top and bottom tercile of CDNC are contrasted it is found that they not only have higher CLWP, but also cloud cover, and albedo. The difference in cyclone albedo between the cyclones in the top and bottom third of CDNC is observed by CERES to be between 0.018 and 0.032, which is consistent with a 4.6-8.3 $Wm^{-2}$ in-cyclone enhancement in upwelling shortwave when scaled by annual-mean insolation. Based on a regression model to observed cyclone properties, roughly 60% of the observed variability in CLWP can be explained by CDNC and WCB moisture flux.

## 1. Introduction

The degree to which the aerosol indirect effects that result from anthropogenic aerosol emissions have acted to increase planetary albedo and mask greenhouse gas warming is highly uncertain (Andreae et al., 2005;Carslaw et al., 2013;Boucher et al., 2014;Forster, 2016). Establishing how much the aerosol emitted during the 20[th] century has enhanced the liquid water amount and thus the albedo of midlatitude storm systems is a key step in constraining the climate sensitivity inferred from the observational record.

Extratropical cyclones play an important role in not only determining midlatitude albedo, but also the transport of moisture, heat, precipitation, and momentum (Hartmann, 2015;Catto et al., 2012;Hawcroft et al., 2012;Trenberth and Stepaniak, 2003;Schneider et al., 2006). Based on observational case-studies and modelling it is known that both the synoptic-scale atmospheric processes and much smaller scale cloud microphysical processes play a role in regulating the cyclone lifecycle (Naud et al., 2016;Grandey et al., 2013;Lu and Deng, 2016;Thompson and Eidhammer, 2014;Igel et al., 2013;Lu and Deng, 2015;Zhang et al., 2007;Naud et al., 2017).

In general, for warm rain processes, enhancement in aerosol that can act as cloud condensation nuclei (CCN) should enhance cloud droplet number concentration (CDNC, the first indirect, or Twomey effect) (Nakajima et al., 2001;Charlson et al., 1992;Twomey, 1977). This effect has the potential to suppress precipitation and lead to a greater retention of liquid water within the cloud (the second indirect, lifetime, cloud adjustment, or Albrecht effect) (Albrecht, 1989;Gryspeerdt et al., 2016;Sekiguchi et al., 2003). Empirical studies have established some evidence supporting the existence of these effects in liquid clouds (Gryspeerdt et al., 2016;Quaas et al., 2008;Nakajima et al., 2001;Sekiguchi et al., 2003;McCoy et al., 2017b;McCoy et al., 2015;Meskhidze and Nenes, 2006;Toll et al., 2017;Naud et al., 2017), although it has been argued that compensating physical processes may offset these microphysical perturbations(Stevens and Feingold, 2009;Malavelle et al., 2017;Igel et al., 2013;Sato et al., 2018). Covariability between aerosol optical depth (AOD) and cloud cover in extratropical cyclones has been shown by previous studies(Naud et al., 2017;Grandey et al., 2013)- supporting the idea that cloud adjustments occur in this regime. Here we use global, high-resolution simulations and remote-sensing observations to indicate that aerosol-cloud interactions produce an increase in the cloud liquid water content, cloud extent, and albedo of extratropical cyclones.

In section 2 we will discuss the observations, and idealized simulations of cloud responses to changes in CDNC used to examine the effects of aerosols and meteorology on cloud properties. We will also discuss the modelling of volcanic plumes from the 2014-2015 Holuhraun eruption used to provide preliminary comparison to recent results indicating an insensitivity of cloud water content to volcanic aerosol (Malavelle et al., 2017). In section 3 we present our analysis of our idealized aquaplanet simulation and we test the hypothesis arrived at in these simulations in the observational record. In section 4 we summarize our results. A list of the acronyms used in this study is provided in Table 1.

## 2. Methods

### 2.1 Cyclone compositing

Many previous studies have demonstrated the usefulness of averaging around cyclone centers to examine midlatitude behavior-including aerosol variability (Field et al., 2011;Field and Wood, 2007;Naud et al., 2016;Catto, 2016;Naud et al., 2017;Grandey et al., 2013). A variety of different techniques for locating cyclone centers and compositing around elements of cyclones exist in the literature utilizing pressure fields (Jung et al., 2006;Löptien et al., 2008;Hoskins and Hodges, 2002;Field et al., 2008); geopotential height (Blender and Schubert, 2000); and vorticity (Sinclair, 1994;Hoskins and Hodges, 2002;Catto et al., 2010). In this study we utilize the methodology described in Field and Wood (2007). This algorithm locates cyclone centers based on sea level pressure (SLP) and then composites around each center. In this study we use the same constants relating to minima, slope and concavity of SLP contours as defined by Field and Wood (2007) to locate cyclone centers. As in Field and Wood (2007) SLP is resolved at 2.5°, and each composite is 4000 km across. When cyclone compositing is performed on observations, only cyclone centers with 50% or more of the composite area located over ocean are considered valid. All observations that are over land are removed from the composite. Cyclone centers are located in both hemispheres, but southern hemisphere cyclone composites are shown oriented so that they have a consistent orientation with northern hemisphere cyclones (Fig. 1).

## 2.2 Observations

### 2.2.1 SLP

The Modern-Era Retrospective Analysis for Research and Applications version 2(Bosilovich et al., 2015) (MERRA2) daily-mean sea level pressure (SLP) was used to locate cyclone centers in the observational record from 2003-2015 using the algorithm described above.

### 2.2.2 MAC-LWP

The Multi-Sensor Advanced Climatology framework used for developing monthly cloud water products (Elsaesser et al., 2017) is adapted for use here to create diurnal-cycle corrected and bias-corrected daily datasets for total liquid water path (LWP, where path is the mass in an atmospheric column), 10-meter wind speed, and water vapor path (WVP).

One possible caveat in our analysis is that the radiative signal used to retrieve LWP may partly arise from upwelling radiation due to wind roughening of the ocean surface or emission from WVP. In such cases, LWP is biased in one direction, while wind and/or WVP may be biased in an opposite direction (Elsaesser et al., 2017). However, retrievals of WVP and wind speed have been shown to be unbiased relative to in situ observations and thus such issues are likely minimal (Mears et al., 2001;Wentz, 2015;Trenberth et al., 2005;Meissner et al., 2001;Elsaesser et al., 2017).

Because microwave radiometers must make assumptions regarding the partitioning of precipitating and non-precipitating liquid this represents a systematic uncertainty in the microwave LWP data set. To bypass this source of uncertainty we utilize the total LWP data product provided by MAC-LWP. The total LWP observations from this data set represent the precipitating and non-precipitating liquid water averaged over both cloudy- and clear-sky. In this study we define the sum of precipitating and non-precipitating LWP within the cyclone as cyclone-LWP (CLWP). It should be noted that the MERRA2 reanalysis total precipitable liquid water (the TQL data in MERRA2) was compared to the microwave CLWP as a rough indicator of how MERRA2's cyclone properties covaried with its predicted sulfate.

**2.2.3 CDNC**

Cloud droplet number concentration (CDNC) is the key state variable that moderates the relationship between aerosol and cloud properties such as LWP and cloud fraction (Wood, 2012). In this study we use two different data sets to describe CDNC: (1) the CDNC retrieved by the Moderate Resolution Imaging Spectroradiometer (MODIS) (King et al., 2003;Nakajima et al., 2001;Grosvenor and Wood, 2014) and (2) 910 hPa sulfate mass from the MERRA2 reanalysis. Data set (2) is used to assess the robustness of our analysis in regards to any remote-sensing retrieval errors in data set (1).

Retrievals of CDNC from the MODIS instrument were performed as described in Grosvenor and Wood (2014) and are the same data evaluated in McCoy et al. (2017a). In the present study, level-2 swath data (joint product) from MODIS collection 5.1 (King et al., 2003) is filtered by removing pixels with solar zenith angles greater than 65° to eliminate  problematic retrievals at a pixel-level (Grosvenor and Wood (2014). The daily-mean CDNC at 1°x1° resolution is calculated from the filtered level 2 swath data and only low (cloud tops below 3.2 km), liquid clouds were used to calculate CDNC. Only 1°x1° regions where the cloud fraction exceeds 80% are considered valid (Bennartz et al., 2011) and the CDNC is calculated using the 3.7μm MODIS channel effective radius.

The second estimate of CDNC is provided by MERRA2 using sulfate ($SO_4$) mass. Previous studies have shown that MERRA2 sulfate mass is a good predictor of CDNC as retrieved by MODIS (McCoy et al., 2017a;McCoy et al., 2017b). The relationship used in the present study to calculate CDNC from 910hPa sulface mass is $CDNC = 10^{0.41 log_{10} SO4 + 2.1}$, where CDNC is in units of $cm^{-3}$ and SO4 is in units of $\mu g/m^3$. Since MERRA2 aerosol assimilation does not ingest MODIS cloud properties the CDNC from MODIS should not influence MERRA2 sulfate(Randles et al., 2016). One caveat to using MERRA2 sulfate as a proxy for CDNC when investigating cloud-aerosol adjustments is that MERRA2 does ingest microwave-retrieved rain rates up until 2009 and clear-sky microwave WVP into its reanalysis(McCarty et al., 2016). The possible influence of the assimilation of these cloud and meteorological properties into the MERRA2 reanalysis are evaluated in section 3. It should be noted that support for the usefulness of this data product has been provided by studying long-term trends related to volcanism and pollution controls. These have shown consistency between MODIS CDNC and sulfate mass from MERRA2 as

well as observations of boundary-layer sulfur dioxide from the ozone monitoring instrument (OMI)(McCoy et al., 2017a).

These two datasets use independent approaches to estimate CDNC and will not be subject to the same errors in representing the true cloud microphysical state. Using estimate of CDNC from these two sources will yield insight into the observational uncertainty surrounding CDNC.

### 2.2.5 Albedo and cloud fraction from CERES

The analysis presented in this work focuses on changes in liquid water in cyclones. However, changes in the cloud fraction (CF) and the all-sky albedo are central in the evaluation of the forcing related to cloud adjustments to aerosol. We utilize observed albedo and CF from the CERES 3-hourly observations, where the all-sky albedo is for clear and cloudy regions. The decision to use all-sky albedo has been made to parallel previous studies (Bender et al., 2017;Bender et al., 2016;Engstrom et al., 2015b;Engstrom et al., 2015a) and has the benefit of not being sensitive to thresholding in the same way that cloud property retrievals are(Marchand et al., 2010). If we used an in-cloud albedo this would mean that only confidently cloudy pixels would be considered. By using an all-sky albedo this allows consideration of the contributions of broken and sub-pixel cloud cover to albedo. Broken cloud are a prominent feature in midlatitude cyclones and have the ability to substantially influence all-sky albedo (McCoy et al., 2017c). One important caveat to this methodology is that we cannot partition the albedo change into the direct effect of aerosols, the first indirect effect, and adjustments. To offer an estimate of the change due to the direct effect we examine the CERES clear-sky albedo.

The 3-hourly data is averaged to create a daily-mean albedo and CF. CF, clear-sky albedo, and all-sky albedo are provided in the CERES SYN1DEG data set edition 4(Wielicki et al., 1996;Doelling et al., 2013;Doelling et al., 2016). CF is calculated from MODIS and geostationary satellites based on the Minnis et al. (2011) cloud mask. It is used in the calculation of the albedo retrieved by CERES as described in Doelling et al. (2016) to create an angular distribution model and to interpret geostationary observations of albedo in relation to the observations from CERES. It should be noted that without utilizing a satellite simulator(Bodas-Salcedo et al., 2011) we cannot directly compare cloud fraction to the aquaplanet simulations presented in this work.

To calculate the shortwave (SW) forcing that is consistent with albedo differences we need to know the downwelling SW. Mean solar insolation (30°-80° ) was calculated using the CERES EBAF-TOA edition 4 data set(Loeb et al., 2009). This quantity was used to estimate the change in reflected SW from the difference in albedo.

5   The dependence of albedo on solar zenith angle (SZA) is well-documented and needs to be either removed or treated in order to contrast variations in albedo generated by clouds across latitudes and seasons (Bender et al., 2017). The dependence of albedo on cloud fraction and SZA in 3-hourly CERES data is shown in Fig. 2. Above a SZA of 45° the albedo depends strongly on SZA. While this is a real effect of low sun angles, we are more interested in understanding the albedo of cyclones without the SZA

10 effect. To mitigate this effect we remove observations where SZA exceeds 45° from the 3-hourly observations. To examine sensitivity to this cutoff we also utilize SZA cutoffs of 30° and 60°. The effect of these different cutoffs on the dependence of albedo on CF is shown in Fig. S2.

### 2.3 Models and Simulations

### 2.3.1 Aquaplanet

15   Two sets of simulations in the MetOffice Unified Model (UM) vn10.3 based on GA6 (Walters et al., 2017) were created to test the sensitivity of cloud adjustment to changes in CDNC to model resolution in an idealized aquaplanet setting. The simulations were performed in a GCM-surrogate setting and a convection-permitting setting. The GCM-surrogate model provides a comparison to the resolution of a typical GCM and was run at 1.89°x1.25° horizontal resolution. It incorporated a parameterized convection

20 scheme, but no cloud-scheme was implemented meaning that only convective and large-scale clouds were simulated. The convection-permitting model was run at 0.088°x0.059° and neither convection parametrization nor cloud scheme were used. It is accepted that using this resolution (roughly 6.8km in the midlatitudes) does put the convection-permitting simulation within the convective 'Grey Zone'. The use of simulations at this resolution presents both benefits and drawbacks. Without convection being

25 parameterized microphysics and aerosol explicitly interact at the model resolution allowing the cloud system to evolve in terms of changes to the rain and the anvils of the convection as well as cloud-to-cloud interactions mediated via cold pools and modifications to the thermodynamic and moisture profiles.

However, while we are able to afford global aquaplanet runs at this resolution, it is not sufficiently finely resolved to completely resolve convection (as noted above) and this may lead to unknown errors in the simulations. We acknowledge these potential shortcomings. However, our results are able to probe process-related interactions in a way that parametrized convection simulations are structurally incapable

of. Intercomparison of simulations at scales ranging from 1-16km show minimal change to the mean statistics of simulated cloud fields (Field et al., 2017). This gives us some confidence that our results will not just be a product of the resolution of the simulations. As discussed below, we find that both GCM-surrogate and convection permitting simulations increase CLWP as aerosol increases. The response of CLWP to aerosol in the convection-permitting simulation is more pronounced than the GCM-surrogate

simulation, but does not contradict it.

        Both convection-permitting and GCM-surrogate simulations were run with 70 vertical levels.  The Cloud-AeroSol Interacting Microphysics (CASIM) two-moment microphysics scheme (Hill et al., 2015;Shipway and Hill, 2012;Grosvenor et al., 2017;Miltenberger et al., 2018) was used for all clouds in the convection-permitting simulation and for large-scale cloud cover in the GCM-surrogate simulation.

The CASIM microphysics scheme is described in Shipway and Hill (2012). The warm rain processes in CASIM is compared to other microphysics schemes in Hill et al. (2015). The cloud physics parameterization used in CASIM is described in Khairoutdinov and Kogan (2000).  Convective clouds in the GCM-surrogate simulation do not parameterize aerosol-cloud interactions. This is consistent with most operational climate and global numerical weather prediction models (Boucher et al., 2014).

Sea surface temperature (SST) was held fixed in the simulations and the atmosphere was allowed to spin up for a week at low resolution and then for another week at high resolution. The SST profile used in the aquaplanet was derived from a 20-year climatology run from the UM in standard climate model configuration. The January SST was averaged with a north-south reflected version of itself and then zonally averaged to provide a symmetrical SST.

Aerosol concentration is constant in the simulations. We assume that the removal rate of aerosol is equal to the rate of replenishment and the time rate of change in aerosol concentration is zero, which is a reasonable approximation over much of the global oceans (Wood et al., 2012). However, we do not expect this to be an accurate representation of how real-world cyclones behave- for example precipitation

depletion of aerosol should be stronger in post-frontal region, which is not reflected in this simplistic set-up. We have chosen to represent aerosol in this way because it reduces the complexity of the idealized model and still allows us to gain insight into how modulation in CDNC alters cloud properties. With these caveats to our analysis in mind, we will now describe the aerosol profiles used in the control and

enhanced aerosol concentration simulations.

The aerosol profile in the control simulation was 100 cm$^{-3}$ in the accumulation mode at the surface up until 5km and then exponentially decreased after 5km with an e-folding of 1 km. Aerosol-cloud interactions were parameterized using a simple Twomey-type parameterization(Rogers and Yau, 1989) with $CDNC = 0.5N_{acc}\text{w}^{0.25}$ with $N_{acc}$ being accumulation mode aerosol number concentration and w

being updraft velocity limited such that at w=16m/s CDNC= $N_{acc}$. The vertical velocity was set to have a minimum value of 0.1m/s. The effects of enhanced aerosol on clouds was investigated by increasing aerosol at the surface to 2000 cm$^{-3}$ in a channel between 30°N and 60°N (with an exponential decay after 5km with an e-folding of 1 km, as in the control simulation). Ice number was controlled using a simple temperature-dependent relationship (Cooper, 1986). Simulations were run for 15 days. A single

simulation was run at each resolution and aerosol concentration, giving a total of four simulations of 15 days each.

It is important to note that an increase in CDNC with increasing $N_{acc}$ is guaranteed by the activation parameterization used in these simulations. However, the intention of these simulations is to evaluate the response of macrophysical cloud properties to changes in CDNC and these aquaplanet

simulations should be thought of in the context of an artificially constrained CDNC set of experiments as opposed to 'fixed-CCN' experiments. In addition to the large change in CDNC between the different sensitivity experiments a small amount of variability in CDNC is introduced by vertical velocities in excess of 0.1 m/s, as described above.

It is also important to note that the fixing of CDNC at a constant value means that precipitation

does not affect CDNC via the removal of aerosol and thus CCN. The simulations presented here are intended to examine the adjustment in cyclone clouds to changes in CDNC as opposed to a change in aerosol fluxes. If aerosol were allowed to respond to precipitation we may speculate as to how this might affect the behavior of the cloud adjustment simulated by CASIM. As described in the following sections,

the rain rate on a daily, cyclone-wide scale is determined by the large-scale environment. Subsequently, we may hypothesize that the feedback between aerosol, CDNC, and the rain rate is relatively weak, but we note that this assumption of fixed CDNC artificially removes this interaction pathway with the intent of understanding the adjustment in cloud properties to CDNC.

5          Finally, in the simulations presented in this paper we explore the response of the clouds in the UM treated by the CASIM cloud microphysics to changes in CDNC. A different cloud microphysics scheme would potentially yield a different adjustment to aerosol, but our results are unlikely to be qualitatively dependent on the simplistic activation scheme chosen here. We also acknowledge that the adjustment of cloud to aerosol in these idealized simulations will be a function of the CASIM microphysics scheme.

Examination of CASIM in relation to other multi-moment schemes suggests that if the adjustment works through the warm rain process another multi-moment scheme would produce a qualitatively similar result (Hill et al., 2015). It is important to note that the simulations presented in this work include ice processes, which may affect the susceptibility of rain rate to changes in CDNC (Koren et al., 2005;Rosenfeld and Woodley, 2000). These effects may be highly dependent on the choice of microphysics scheme. Further,

the representation of these effects in models is very uncertain and could substantially affect the predictions of our simulations. Lastly, the evaporation-entrainment feedback on aerosol-cloud interactions (Hill et al., 2009;Xue and Feingold, 2006;Xue et al., 2008) is not well represented in these simulations due to model vertical grid resolution and boundary layer treatment.

        In the context of the CASIM cloud scheme used here we note that an increase in LWP in response

to CDNC is guaranteed for a precipitating grid box (all else being equal). That is to say, if we examine a precipitating grid box in the model with a given liquid water content and instantaneously increase the CDNC, on the subsequent time step the grid box will have increased its liquid content because precipitation will be inhibited. If there is no precipitation the liquid content will remain unchanged. This is a common feature of warm clouds in models(Hill et al., 2015), and appears in the LWP response

simulated by higher horizontal resolution instances of the CASIM model(Grosvenor et al., 2017). While some LWP reduction effects such as evaporation entrainment will not be as efficacious in CASIM due to vertical grid resolution and boundary layer treatment, Miltenberger et al. (2018) showed, using CASIM, that the subsequent evolution of the clouds in the context of a realistic forcing may yield decreased LWP

in response to increased CDNC through interaction with the environment and between clouds. In summary, CASIM's vertical resolution and boundary layer treatment make it less likely that mechanisms such as the evaporation-entrainment feedback will be as efficacious and the LWP response to enhanced CDNC might be less pronounced in a different model that is able to capture these effects. Overall, we present these simulations as an exploration of how clouds within cyclones respond to changes in CDNC through the warm rain process. These simulations are used to contextualize the observations and evaluate whether we may reproduce observational variability utilizing this idealized set of simulations.

### 2.3.2 Dispersion model simulations of the 2014-2015 Holuhraun eruption in Iceland

The 2014-2015 eruption of Holuhraun in Iceland emitted a large quantity of sulfur into the troposphere (Gettelman et al., 2015;Schmidt et al., 2015) and served as a case study of how clouds respond to changes in aerosol (McCoy and Hartmann, 2015;Malavelle et al., 2017). Because Holuhraun is in the midlatitudes it offers an opportunity to examine how cyclone properties are altered by sulfate aerosol particles. The Numerical Atmospheric-dispersion Modelling Environment (NAME) is a Lagrangian dispersion model (Schmidt et al., 2015;Jones et al., 2007) that was used to simulate the chemical conversion and dispersion of sulfur dioxide and sulfate aerosol particles for the first two months of the Holuhraun eruption. Simulations were run using reanalysis meteorology for the eruptive period from the UM as described in Schmidt et al. (2015).

The output from the Holuhraun simulations was used to determine which cyclones had interacted with the volcanic sulfate plume. The simulations were configured using a time varying flux of $SO_2$ of 100 kt/d between 31 August 2014 and 13 September 2014 and 60 kt/d thereafter in line with observations and fluxes derived in a previous study (Schmidt et al., 2015). Emissions were distributed uniformly between 1500-3000m, consistent with observed plume heights during September 2014(Schmidt et al., 2015). Sensitivity to emission height was tested by running a second simulation with emissions between 0-1500m. The near-surface sulfate mass was calculated by taking the mean over the bottom five model levels (100m-900m). This sulfate mass was used to determine which cyclones interacted with the sulfate plumes from the eruption during September and October of 2014.

**3. Results and discussion**

In this section we present observational analysis showing that mid-latitude cyclone liquid water content, cloud cover, and ultimately, albedo covary with changes in cloud droplet number concentration (CDNC). This work was motivated by a set of idealized convection-permitting experiments designed to examine
how mid-latitude cyclone properties change in response to cloud microphysics. In section 3.1 we will discuss the characterization of cyclone systems in the midlatitudes and how we can stratify them in relation to the large-scale environment. In section 3.2 we will examine how meteorology determines cyclone properties and compare this dependence across our aquaplanet simulations, and observations. In section 3.3 the response of cyclones to a change in CDNC in the aquaplanet simulations will be contrasted
with the covariability between CDNC and cyclone-mean properties in the observational record. In section 3.4 we will examine which parts of midlatitude cyclones differ between high and low CDNC populations and will contrast these observations with the change in cyclone structure in the aquaplanet simulations. In section 3.5 we show that the all-sky albedo in midlatitude cyclones differs between high and low CDNC populations. In section 3.6 we fit a regression model to explain cyclone liquid water path as a function
of microphysics (CDNC) and meteorology (WCB moisture flux) and find we are able to explain the majority of extratropical cyclone variability by these two predictors. Finally, in section 3.7 we examine cyclones during the eruption of Holuraun utilizing dispersion modelling to examine the propagation of the volcanic plume.

**3.1 Large-scale environmental controls on mid-latitude cyclones in relation to microphysical perturbations**

Compared to the meteorological drivers of cyclone formation, aerosol-cloud interactions are subtle and difficult to observe. To understand the contributions of aerosol and meteorology to cyclones we need to characterize what constitutes a cyclone. Cyclone centers were identified using sea level pressure (SLP), in keeping with previous studies (Field and Wood, 2007) and as described in the methods
section. Cyclone centers were identified in both the northern and southern hemispheres between 30°-90° degrees latitude over ocean. Cyclone compositing was performed to identify centers for both observed and simulated cyclones. Because microwave CLWP cannot be retrieved over land surfaces, only cyclone

centers with a substantial fraction of the cyclone over ocean were considered valid. A minimum ocean coverage of 50% within the 2000 km radius composite was required to include cyclone centers in our analysis. As noted in the methods section, southern hemisphere cyclones are flipped so that their orientation is consistent with cyclones in the northern hemisphere (Fig. 1). That is to say, the poleward half of the cyclone is shown in the top part of the composite and the equatorward half is shown in the bottom half of the composite.

Now that we have created a database of observed cyclones we need to stratify them by the large-scale environmental factors that are controlling their development. Considerable research has been devoted to investigating the dependence of cyclone properties on meteorology using cyclone composites(Catto, 2016). One that has been found to be particularly useful is the so-called warm conveyor belt (WCB) metric  (Field and Wood, 2007;Pfahl and Sprenger, 2016;Harrold, 1973). This relies on a simple model of cyclone development as described in Harrold (1973) and is calculated as the product of cyclone-mean wind speed and water vapor path multiplied by a constant describing the width of the warm conveyor belt as defined in Field and Wood (2007). It should be noted that cyclone-mean here and in the rest of this article refers to an average taken within a 2000 km radius of the cyclone center. WCB moisture flux is a proxy for the moisture flux ingested by the cyclone and is a good predictor of the cyclone-mean rain rate in observations and global climate models (Field and Wood, 2007;Field et al., 2011).

We created a suite of simulations in the MetOffice Unified Model (UM) that is intended to explore aerosol-cloud interactions within mid-latitude cyclones, these simulations are described in the methods section in more detail. Because the focus of this study is to understand maritime, midlatitude storms, the model has no land surface (an aquaplanet) allowing an unbroken storm track providing more cyclones to be analyzed without the complications of landmasses on their evolution.  A control simulation and enhanced aerosol simulation were run at high and low resolution to see how cyclones differed when CDNC was increased. In the control simulation accumulation mode aerosol concentration was set at a value of 100 cm$^{-3}$ near the surface and in the enhanced aerosol simulation the accumulation mode aerosol concentration was set to 2000 cm$^{-3}$ near the surface in the 30°N-60°N latitude band.

Only liquid droplets are directly affected by the aerosol changes. For ice, number concentrations followed a simple temperature-dependent relationship, which typical of a GCM participating in the

climate model intercomparison project (CMIP). Minimal impact is made on ice concentrations through variations in $N_{acc}$ (hence small changes to longwave radiation). We do not vary the parametrizations that control the ice number when we vary $N_{acc}$.

Examination of the results of our convection-permitting simulations show that the relationship
between WCB flux and precipitation or rain rate is relatively invariant as a function of model resolution and aerosol concentration (Fig. S3). However, the slope of the relationship between precipitation or rain rate and WCB moisture flux is somewhat shallower in the low-resolution model. Further, use of this WCB metric is particularly useful in the context of our analysis because it can be measured by a microwave radiometer allowing us to readily compare simulations and observations.

This consistency across models of varying spatial resolution and observations of real-world cyclones seems reasonable because once in equilibrium, the water mass flux that goes into the cyclone must be precipitated out. The perturbed aerosol environment reduces the efficiency of warm rain production for a given water path and therefore should lead to a higher equilibrium water path for a given mean rain rate or WCB flux. Reliable observations of ice cloud properties are not available(Jiang et al.,
2012) so it is difficult to infer the importance of ice cloud in this mechanism. However, the frozen water path in the cyclones did not change between control and enhanced aerosol experiments, indicating that this aerosol-cyclone indirect effect primarily acts through the warm rain process, at least within our aquaplanet simulations (Fig. S4).

Casting our analysis as a function of WCB moisture flux means that we are investigating cyclone
responses to changes in CDNC at a set precipitation rate. One possibility is that this framework will prove expedient to our analysis of cloud adjustments to aerosol changes in cyclones given the divergence in precipitation responses in previous studies, ranging from intensification of precipitation (Zhang et al., 2007;Thompson and Eidhammer, 2014;Wang et al., 2014), to unchanged precipitation(Igel et al., 2013), or suppression of precipitation(Lu and Deng, 2016).

**3.2 Comparison between observed and simulated cyclone properties and their dependence on meteorology**

Before examining the response of cyclones to changes in cloud microphysics we compare observed and simulated cyclones. In particular, we examine their response to changes in synoptic

meteorology as characterized by WCB moisture flux. Comparison between MAC-LWP observations of cyclone-composited CLWP and aquaplanet simulations are shown in Fig. 3. To compare cyclone composites in similar meteorology conditions the cyclone composites are shown stratified into terciles of WCB moisture flux. The terciles are determined by the observational record of WCB moisture flux and

correspond to 0-2.21mm/day, 2.21-2.88 mm/day, and above 2.88 mm/day. The bounds on WCB from these terciles are also used in the presentation of the aquaplanet simulations.

The simulations carried out at convection-permitting resolution and the observations show reasonable agreement in structure and some agreement in absolute value. Both the convection-permitting and GCM-surrogate simulations generally have a lower CLWP than the observations, but this is not

surprising because no cloud-scheme is used in these simulations. That is to say, only supersaturations resolved at the model's resolution will produce cloud.  Use of a cloud scheme would increase the CLWP and bring the simulations into better absolute agreement with observations.  However, the cloud scheme would require a choice of critical relative humidity(Quaas, 2012;Grosvenor et al., 2017), which would complicate our analysis of these simulations across resolutions. The GCM-surrogate simulation has a

much lower CLWP than either the convection-permitting simulations or the observations. This is also likely to be at least partially due to the lack of a cloud scheme meaning that only convection or times where the entire grid box is saturated will be cloudy. Cyclone-centric composites of MERRA2 total precipitable liquid water are shown in Fig. 3f and agree somewhat with MAC-LWP observations, although the contrast between different WCB moisture flux regimes is not as strong and the cyclones are

significantly more diffuse. We will return to the discussion of the MERRA2 cyclone properties in the following sections to evaluate whether MERRA2 sulfate covariability with MAC-LWP CLWP is dictated by the MERRA2 CLWP.

One consistent behavior observed across the aquaplanet simulations and observations in Fig. 3 is the enhancement in CLWP with increasing WCB moisture flux. As one might expect, a greater flux of

moisture into the cyclone results in a larger total CLWP. Such a clear WCB-CLWP relationship provides a useful metric with which to stratify midlatitude cyclones. In this framework we can now ask: for a given WCB moisture flux, do variations in the aerosol that is active as CCN available to the cyclone and hence CDNC result in a different CLWP?

**3.3 The response of the mean properties of mid-latitude cyclones to changes in cloud microphysics**

As we saw in the last section, the WCB moisture flux into cyclones exerts a substantial control on the amount of liquid within the cyclone and is a quantity that we may observe remotely. We now ask the question: if we segregate cyclones into low CCN and high CCN populations will this behavior change?

5       Determining whether observed midlatitude cyclones have a higher or lower CCN available is difficult. One approach would be to use the retrieved cloud droplet number concentration (CDNC) from MODIS. This provides a good proxy for CCN (Wood, 2012), but as described in the methods section it is potentially problematic because retrieval errors relating to overlying ice cloud(Sourdeval et al., 2016), cloud heterogeneity (Grosvenor and Wood, 2014;Sourdeval et al., 2016), and low sun-angle(Grosvenor

and Wood, 2014) may spuriously bias the measurements making it difficult to interpret any observed covariation between cyclone properties and CDNC. That is to say, retrieval error may be hypothesized to lead to any covariability that we discover in our analysis.

The CDNC calculated by MODIS will suffer from retrieval errors and basing our entire analysis on it would be problematic. To avoid these ambiguities we take a similar approach to previous studies

(Boucher and Lohmann, 1995) and use both CDNC retrieved by MODIS and the sulfate mass concentration at the surface simulated by MERRA2 reanalysis(McCoy et al., 2017b;McCoy et al., 2015;McCoy et al., 2017a). This use of the sulfate mass as a proxy for MODIS observations of CDNC is advantageous because it is not susceptible to retrieval error and because MERRA2 does not have a parameterized cloud-aerosol indirect effect. If we see a similar behavior when we use MERRA2 sulfate

to stratify cyclones into low and high CDNC populations as we do when we stratify using retrieved CDNC then this covariability is not created by remote sensing retrieval biases.

Using the daily-mean MERRA2 SO4 we calculate a CDNC proxy within cyclones following the relationship established in previous studies (McCoy et al., 2017b). This gives a CDNC proxy that is calculated using the MERRA2 reanalysis and, independently, an observation of CDNC from MODIS. We

examine whether both metrics for CDNC show similar behaviors when composited around cyclone centers.

By examining cyclone-centric composites of CDNC we see an enhancement in CDNC retrieved by MODIS and inferred from MERRA2 in the southwest quadrant (for a poleward-oriented composite;

Fig. 1, and Fig. 4abc). This region has been hypothesized by previous studies to be the source of moisture and aerosol for the cyclone (Cooper et al., 2004;Naud et al., 2016;Joos et al., 2016). Based on these studies the southwest quadrant of the cyclone composited CDNC will be used to stratify cyclones by CCN and will be referred to as $CDNC_{SW}$. This region is shown in Fig. 1. Again, we note that in this study all cyclone composites are oriented so that north is toward the pole and south is toward the equator so that northern and southern hemisphere cyclones are consistently oriented.

Because of the restrictions on what retrievals of CDNC are considered reliable (Grosvenor and Wood, 2014) large regions of the cyclone composite inhabited by ice cloud may be missing, in contrast no data is missing from MERRA2 sulfate because it is a reanalysis product. Examples of cyclone composited CDNC from MODIS, and MERRA2 are shown in Fig. 4ab. While MERRA2 infers enhancement in CDNC in the southwest quadrant, MODIS shows a higher CDNC in the north (or poleward, Fig. 1) part of the composite, which is likely due to retrieval bias at low sun angles and from heterogeneous cloud. Due to the vagaries of retrieving CDNC from space in the presence of broken or icy cloud, the cyclone-composited CDNC has quite different structures depending on whether it is retrieved by MODIS or whether MERRA2 SO4 is used as a proxy for CDNC. However, the inter-cyclone variability in both cyclone-mean CDNC and $CDNC_{SW}$ retrieved by MODIS and inferred from MERRA2 are in agreement (Fig. 4d). Further, when MERRA2 is sampled where MODIS can perform a retrieval (effectively removing SO4 data when overlying ice cloud is present), the pattern of CDNC within the mean cyclone composite is in better agreement (Fig. 4cd).

Using WCB moisture flux as a measure of the meteorological condition and $CDNC_{SW}$ as a measure of CCN available to the cyclone we may evaluate the observational record and compare it to the aquaplanet simulations of CDNC enhancement. We examine the observational record of cyclone-mean CLWP by stratifying it into the top and bottom third of retrieved $CDNC_{SW}$. This is done separately using CDNC inferred from MERRA2 and retrieved by MODIS (Fig. 5cd). There is a systematic separation in mean CLWP between high and low $CDNC_{SW}$ cyclones (Fig. 6ab). The mean separation between cyclone populations is 12.7±0.7 g/m2 when MODIS is used to perform the partitioning, where the uncertainty is the 95% confidence interval assuming a normal distribution. When MERRA2 sulfate is used the difference is 15.3±0.58 g/m2. Is this behavior replicated by our idealized aquaplanet simulation?

We answer this question by comparing the low and high CDNC simulations and stratifying by WCB flux, as we did with the observations of cyclone properties. In this case we uniformly perturb the CDNC, as opposed to comparing populations within the observations. Simulations at convection-permitting resolution and low resolution are examined. When the simulations are stratified and compared in this way their behavior mimics the observations. That is to say, for a given WCB, higher $N_{acc}$ translates to a higher CLWP (Fig. 6c). The difference between the control and CDNC-enhanced simulations is more pronounced in the convection-permitting model. This may be because in the GCM-surrogate simulation aerosol-cloud interactions are not represented for convection, while in the convection-permitting simulation aerosol-cloud interactions are treated in the same way for all cloud elements. It may also reflect the cloudier base state of convection-permitting simulation. However, it is possible that the aerosol-cloud indirect effect as simulated by traditional GCMs that do not include aerosol-aware convection is systematically too weak in the midlatitudes. This is because increased model CLWP results in enhanced reflection of shortwave radiation to space (Fig. S5), although thick ice clouds may mute the enhancement of reflected shortwave radiation. Of course, assuming that missing aerosol-cloud adjustments in models indicates an overall aerosol-cloud adjustment that is too weak assumes that on average models have a reasonable representation of this mechanism when it is resolved, which is not necessarily the case.

In summary, our hypothesis, based on our analysis of idealized simulations and observed cyclones, is that *enhanced CCN should enhance CLWP in midlatitude storms for a given WCB moisture flux*. While this hypothesis is evocative, we should note a few potential caveats in our analysis.

The first potential caveat is that it is possible that the $CDNC_{SW}$ inferred from MERRA2 sulfate has somehow been affected by the observations ingested into the MERRA2 reanalysis to create a spurious increase in sulfate in cyclones with larger CLWP, although as we have noted earlier the mechanism by which this could happen is not clear. To evaluate whether this can be the case we examine the total precipitable liquid predicted by MERRA2 composited around cyclone centers. The total precipitable liquid is stratified by WCB moisture flux and then split into high and low $CDNC_{SW}$ populations utilizing MERRA2 predicted sulfate mass. Low $CDNC_{SW}$ cyclones have a higher CLWP than high $CDNC_{SW}$ cyclones at a fixed WCB moisture flux. That is to say, they have the opposite behavior displayed in observations from MAC-LWP (Fig. 6d). Based on this analysis we see that MERRA2's reanalysis is not

ingesting observations of cloud properties in such a way that it spuriously drives variations in the $CDNC_{SW}$ inferred from MERRA2 sulfate mass.

A second caveat to our analysis of cyclone properties as presented above is that there is some sensitivity to what region of the cyclone is used to characterize CDNC. If the cyclone-mean CDNC is used to stratify the cyclones instead of $CDNC_{SW}$ the separation between high and low CDNC populations changes slightly (Fig. S6ab). The mean CLWP for high CDNC cyclones is still significantly higher than the CLWP for low CDNC cyclones at 95% confidence for moisture fluxes below 5 mm/day. The position of the warm conveyor belt is sometimes not in the SW quadrant. Additional sensitivity tests using the southern half of the composite and the south-east quadrant of CDNC to stratify cyclones are shown in Fig. S7 and Fig. S8. Only use of the south-eastern quadrant (Fig. S8) for stratification results in large portions of the high and low CDNC cyclone population being indistinguishable at 95% confidence. This is in agreement with previous studies of the moisture flux (Eckhardt et al., 2004;Naud et al., 2012)- the moisture flux is not exclusively in the southwest (poleward oriented) quadrant, but is frequently in this region. Examination of the CDNC inferred from MERRA2 sulfate also supports the idea that aerosol is imported into the cyclone in the SW (with N being poleward, Fig. 1) quadrant (Fig. 4b). One possibility is that identification of frontal features (Naud et al., 2012) would better allow averaging around the element of the cyclone that carries aerosol into the cyclone. However, based on previous flow studies of aerosol within cyclones (Joos et al., 2016;Cooper et al., 2004;Naud et al., 2016) we believe that the $CDNC_{SW}$ offers a good overall representation of the importation of CCN into the cyclone and it will be used for the remainder of the analysis.

### 3.4 Differences in the structure of clouds within cyclones as a function of $CDNC_{SW}$

Having examined the difference in cyclone-mean properties between high and low $CDNC_{SW}$ populations we now examine differences in cyclone-centered cloud structure between these populations. The mean composite within each tercile of WCB moisture flux and in the high and low $CDNC_{SW}$ populations are calculated and the difference between the composites is taken.

The difference in cloud properties between high and low $CDNC_{SW}$ cyclones share features between observations and modelling, primarily an increase in the MAC-LWP CLWP in the south-west

sector of the cyclone (Fig. 7 and Fig. 8). This increase in MAC-LWP CLWP is particularly interesting as this is the region typically inhabited by open cellular convection trailing the cold front and a major source of error in simulated cyclone properties (Bodas-Salcedo et al., 2014;Naud et al., 2014;Bodas-Salcedo et al., 2012;McCoy et al., 2017d). Numerous studies have linked the dominance of open or closed mesoscale cellular convection to precipitation, and aerosol modulation of precipitation(Stevens et al., 2005;Feingold et al., 2015;Koren and Feingold, 2011;Rosenfeld et al., 2006;Mechem et al., 2012;Goren and Rosenfeld, 2012;Wang and Feingold, 2009a, b). Because of the tenuous nature of this cloud regime, and because they are typically precipitating, it is reasonable to suspect that they will be more susceptible to aerosol-driven changes in their macrophysics than either thick frontal clouds or non-precipitating clouds. It is not the intention of our investigation to examine the complex dynamics of mesoscale cellular convection, but we have chosen our observational data sets so that they do not exclude this cloud regime, and the localization of differences in CLWP between high and low CDNC$_{SW}$ cyclone populations is suggestive given the existing literature regarding both the radiative importance of these clouds(McCoy et al., 2017c) and their relation to precipitation and aerosol(Koren and Feingold, 2011). Overall, this behavior motivates future work examining this region in higher resolution and higher complexity models that can resolve these features.

Examination of the differences in observed cloud coverage between high and low CDNC$_{SW}$ cyclones exhibit a similar pattern of differences to CLWP with enhanced cloud cover in the southwest quadrant of the composites in the second and third tercile of WCB moisture flux (Fig. 9). This feature is compellingly similar to the difference in cyclone properties between high and low aerosol optical depth cyclones as shown by Naud et al. (2017). It is interesting to note that the difference in CF between high and low CDNC$_{SW}$ cyclone populations are substantially more dependent on whether observed or inferred CDNC is being used to partition the cyclone populations. For cyclones whose WCB moisture flux is in excess of 3mm/day the low and high CDNC$_{SW}$ populations diverge in much the same way when using inferred or retrieved CDNC (Fig. 10). It is unclear why the MODIS and MERRA2 partitionings of the cyclone population do not agree for CF as well as they do for CLWP (Fig. 7). However, population mean CF is different between high and low CDNC$_{SW}$ populations at 95% confidence with a mean difference of

1.38%±0.49 to 1.9%±0.49, depending on which CDNC data set is used (note that the difference in cloud fraction is given in percent cloud cover, not percentage difference).

### 3.5 Differences in albedo between high and low CDNC cyclones

As shown above, systematic differences in cyclone coverage and liquid content seem to exist between low and high $CDNC_{SW}$ populations. However, to better infer climate sensitivity using the temperature record the key variable to constrain is the change in reflected shortwave radiation due to aerosol indirect effects (Forster, 2016;Stevens, 2015;Andreae et al., 2005). The difference in cyclone-composited albedo observed by CERES between cyclones whose $CDNC_{SW}$ are in the top and bottom third of the population is shown in Fig. 11. When MODIS is used to partition the cyclone populations by $CDNC_{SW}$, the albedo increases with increasing $CDNC_{SW}$ in the western side of the cyclone and is roughly consistent with the regions whose CLWP and CF increased (Fig. 7 and Fig. 9). Cyclones in the lowest two terciles of WCB moisture flux show relatively little difference in albedo if MERRA2-inferred $CDNC_{SW}$ is used to stratify the observational record (and appear to decrease somewhat in the lowest tercile). We may speculate that this reflects poor representation of the transport of aerosol into low moisture flux cyclones by MERRA2, but the reason for this disagreement is unclear.

We contrast this difference in all-sky albedo between cyclone populations in the real world with the difference in albedo simulated in the aquaplanet simulations at low and high resolution. Differences in simulated albedo between high and low $N_{acc}$ simulations bear some general similarities to the observations (Fig. 12). Albedo increases are much more uniform throughout the entire cyclone region. There is some localization to the southwestern portion of the composite, but the difference in composites is not clearly analogous beyond the sign of the difference. This lack of structural correspondence between simulated and observed all-sky albedo differences may reflect the extremely large difference in $N_{acc}$ imposed on the simulations. This large difference in $N_{acc}$ may have led to a saturation of the enhancement in CLWP by increasing CDNC. This may also reflect the lack of structure imposed on $N_{acc}$ by cyclone dynamics in the simulations because $N_{acc}$ is not depleted by precipitation or advected by the large-scale flow, as it is in the observational record(Cooper et al., 2004).

Having inspected the differences in the structure of cloud properties and albedo between high and low CDNC$_{SW}$ populations we will now examine the difference in cyclone-mean albedo. To do this we first divide the high and low CDNC$_{SW}$ populations into 15 equal quantiles of WCB moisture flux. The mean albedo and the standard error in the mean ($SE = \sigma n$) (where sigma is standard deviation and n is the total number of observations) are calculated in each quantile (Fig. 13a). The 95% confidence interval is calculated assuming the distribution is normal. To calculate the difference in mean albedo between high and low CDNC$_{SW}$ populations in each quantile the quantile-average WCB moisture flux needs to be examined. Because the mean WCB in each quantile may be slightly different for the high and low CDNC$_{SW}$ populations, the mean and standard error for the high CDNC$_{SW}$ population is linearly interpolated so that the mean WCB moisture flux in each quantile is the same for the low and high CDNC$_{SW}$ populations. For each quantile the standard error in the difference is propagated as

$SE_{High-Low} = \sqrt{SE_{High}^2 + SE_{Low}^2}$ . The average difference in albedo across quantiles is taken. The

associated standard error in the averaged difference in albedo is calculated as $\sqrt{\sum \frac{SE_i^2}{15^2}}$. The difference and 95% confidence interval in the difference between high and low CDNC$_{SW}$ populations as a function of WCB is shown in Fig. 13b.

The mean cyclone albedo is higher for the high CDNC$_{SW}$ population at 95% confidence. When MODIS is used to retrieve CDNC the albedo is on average 0.032±0.002 higher in the high CDNC$_{SW}$ population. If MERRA2 inferred CDNC is used then the cyclone albedo is only higher for the high CDNC$_{SW}$ population when WCB moisture fluxes is greater than 2 mm/day with an average difference of 0.018±0.002.

To calculate the difference in terms of a radiative flux the difference in albedo is multiplied by the annual mean climatological downwelling SW associated with the CERES EBAF-TOA data set between 30°-80°. It is important to note that this assumes that the cyclones being affected are randomly distributed in latitude and during the year. This may be somewhat reasonable for anthropogenic pollution, but not for biogenic aerosol sources. Specifically, planktonic sulfur sources have a substantial seasonal cycle leading to their contribution in albedo occurring during the period of maximum insolation (McCoy et al.,

2015;Ayers and Gras, 1991). The difference in reflected SW provided here is only intended to act as a rough guide to contextualize the change in albedo.

Multiplying the albedo by climatological insolation yields a difference in reflected SW between high and low $CDNC_{SW}$ populations of $8.30\pm0.31$ $Wm^{-2}$ if MODIS is used to stratify cyclones and 4.62$\pm$0.33 $Wm^{-2}$ if MERRA2-inferred CDNC is used (Fig. 13b). This result does show some sensitivity to the maximum solar zenith angle considered acceptable for the 3-hourly CERES data. A maximum SZA of 30° yields values of 5.63$\pm$0.9 and 3.93$\pm$1.18 $Wm^{-2}$ for MODIS and MERRA2-inferred CDNC, respectively (Fig. S9)). A maximum SZA of 60° yields values of $6.28\pm0.48$ Wm-2 and $2.15\pm0.5$ $Wm^{-2}$ (Fig. S10). If all SZAs are included, the positions of the low and high $CDNC_{SW}$ cyclones are reversed (Fig. S11). However, the inclusion of all SZAs observed by CERES includes albedos where the SZA effect dominates, so low CF and low CLWP cyclones can have a considerably higher albedo (Fig. S2). Again, this effect is physical, but including the seasonal cycle and position of cyclones in our analysis via this effect makes it difficult to disentangle the very pronounced SZA effect from changes associated with changes in cloud properties. It is also worth noting that the difference in albedo estimated using MERRA2 SO4 to stratify cyclones is likely to have a larger sensitivity to the maximum SZA cutoff used because MODIS CDNC retrievals are not possible when the SZA exceeds 65° and so cyclones in winter are not considered in the analysis, while MERRA2 SO4 allows these cyclones to be examined.

In this analysis we have used all-sky albedo from CERES to examine the response of cyclone albedo to changes in CDNC. This variable was chosen because it does not impose a criterion for what it considers to be a cloud when calculating the albedo, as would a cloudy-sky albedo. If we were to use in-cloud albedo this would necessitate the albedo perturbation being restricted to only confidently cloudy pixels (see Marchand et al. (2010)). For example, this could exclude situations where mesoscale cellular convection was occurring as these regions would not necessarily be considered cloudy. As pointed out by previous studies (McCoy et al., 2017c), these clouds may have a significant impact on all-sky albedo. However, use of all-sky albedo may potentially conflate aerosol direct effects and indirect effects.

First we provide an estimate of how much cloud fraction differences may contribute to the difference in albedo. Because cloud fraction and albedo have a fairly linear relation in the midlatitudes on a monthly time scale (Bender et al., 2011;Bender et al., 2017) we provide a calculation of the change

in albedo related to changes in cloud cover. The observed midlatitude slope of the relation between albedo and fractional cloud cover of 0.4 from Bender et al. (2017) implies a change in albedo between the top and bottom third $CDNC_{SW}$ populations of 0.005-0.007±0.002 (Fig. 10). As shown in Fig. 10, the difference in mean cloud cover between the populations is significant at 95% confidence, but it does not appear to contribute to the majority of the effect on albedo.

Changes in scattering from aerosol could enhance clear-sky, and ultimately all-sky, albedo within cyclones. We examine the cyclone-composited clear-sky albedo observed by CERES. The albedo in cloud-free regions of the cyclone is 0.005 higher in the high CDNC cyclone population if the retrieved CDNC from MODIS is used, but is unchanged if MERRA2-inferred CDNC is used to partition the cyclone population (Fig. 14). This change in albedo implies a 1.38±0.16$Wm^{-2}$ change in reflected SW in cloud-free regions if it is scaled by the annual-mean insolation. The change in all-sky albedo is nearly an order of magnitude larger (Fig. 13). Cyclone cloud cover is usually in excess of 70% (Fig. 10) so this change in cloud-free albedo when averaged over cloudy and clear regions implies a relatively small contribution from the direct effect.

Finally, changes in CDNC in cyclones should contribute to this brightening, but based on the estimated midlatitude brightening due to changes in CDNC made in previous studies(McCoy et al., 2017b;Quaas et al., 2008;Bellouin et al., 2013;Gryspeerdt et al., 2017), it is unlikely that they contribute the entirety of the albedo difference. Overall, we provide this analysis of the difference in observed all-sky albedo to show that the high and low $CDNC_{SW}$ cyclone populations do not have the same brightness. This shows that in the midlatitude cyclone regime the adjustment in cloud macrophysical properties to a change in cloud microphysics does not act in such a way that it counteracts the forcing associated with the first indirect effect, as has been suggested to be the case in both cyclonic and non-cyclonic regimes (Stevens and Feingold, 2009;Malavelle et al., 2017;Toll et al., 2017;Seifert et al., 2012;Seifert et al., 2015;Sato et al., 2018;Michibata et al., 2016).

### 3.6 Regression model of CLWP

Given the pervasiveness of the relationships between CLWP, $CDNC_{SW}$, and WCB, we create a simple regression model of CLWP to allow us to assess how much of the variance is explained by these

parameters and the relative importance of 'meteorology' and 'aerosol'. The relationship between CLWP, WCB and $CDNC_{SW}$ shows differing behavior as a function of $CDNC_{SW}$ with a stronger increase in CLWP for a given increase in $CDNC_{SW}$ in more pristine (low $CDNC_{SW}$) storms – that is to say, examination of the average CLWP as function of $CDNC_{SW}$ and WCB moisture fluxes shows that increasing $CDNC_{SW}$ at

a fixed WCB moisture flux implies a larger increase in CLWP for $CDNC_{SW}<100$ cm$^{-3}$ (Fig. 15, Fig. S12. Using the observational record from 2003-2015 we train a regression model

$$CLWP = aWCB^b CDNC_{SW}^c - d \tag{1}$$

where WCB is in units of mm/day, CDNC is in cm$^{-3}$, and CLWP is in mm. Coefficients for the regression model trained using $CDNC_{SW}$ retrieved by MODIS and inferred from MERRA2 sulfate are shown in

Table 2. The regression model explains 62%-67% of the variance in the observed CLWP.  By using two predictors we are able to explain two-thirds of extratropical cyclone liquid water path variability.

It is interesting to consider how susceptible CLWP is to a perturbation in $CDNC_{SW}$ in the space of WCB and $CDNC_{SW}$. That is to say, what parts of the cyclone population would be more susceptible to changes in CDNC and which are effectively only sensitive to meteorology in the context of equation 1?

We illustrate this by examining the response of equation 1 to typical perturbations in each predictor.  In the context of this illustrative analysis a standard deviation is considered a typical perturbation. The standard deviation in WCB and $CDNC_{SW}$ are calculated across the data record. The coefficients for equation 1 shown in Table 2 are then used to calculate the change in CLWP for a standard deviation increase in WCB and $CDNC_{SW}$. This illustrates the relative importance of changes in aerosol (as

exemplified by $CDNC_{SW}$) and changes in meteorological environment (as exemplified by WCB moisture flux) and is visualized in Fig. 16 for equation 1 trained using MODIS $CDNC_{SW}$.

Based on the simple visualization in Fig. 16 (and Fig. S13 if the $CDNC_{SW}$ inferred from MERRA2 is used to train the model) we can see that changes in CLWP for very pristine ($CDNC_{SW}<60$cm$^{-3}$), large moisture flux cyclones (WCB>4 mm/day) due to unit standard deviation perturbation in $CDNC_{SW}$ are

estimated to be as large as 50% of those from a standard deviation perturbation in meteorology (WCB flux), while very polluted ($CDNC_{SW}>120$cm$^{-3}$), small moisture flux cyclones (WCB<2mm/day) are nearly insensitive to changes in $CDNC_{SW}$. This result is in keeping with Carslaw et al. (2013), which

demonstrated the importance of understanding low CCN regions to constrain the aerosol-cloud indirect effect. The sensitivity of our regression model to CDNC changes supports the importance of understanding CCN sources in remote, pristine regions. Averaged over the observational record, the mean relative contribution of aerosol changes to the variability in CLWP is 20% (30% if MERRA2-inferred $CDNC_{SW}$ is used) based on the observed distribution of cyclones in $CDNC_{SW}$ and WCB space. Evidently the dominant role is played by meteorology, but CDNC variability plays a non-negligible role.

### 3.7 Examination of the Holuhraun eruption case study

Recent investigation by Malavelle et al. (2017) utilizing observations and climate model simulations showed that, despite the massive emission of sulfur dioxide by the Holuhraun fissure in Iceland during September and October of 2014 (Gettelman et al., 2015;Schmidt et al., 2015), and a detectable change in cloud microphysics (McCoy and Hartmann, 2015), cloud liquid water path and coverage did not deviate detectably from their climatological behavior. As described above, based on global observations of extratropical cyclones we infer that both cloud cover and liquid water path within cyclones adjust in response to changes in CDNC- a hypothesis which is consistent with the idealized modelling we have performed. Are our results consistent with the analysis presented in Malavelle et al. (2017)?

We examine cyclones in the vicinity of Iceland (50°W-30°E and 45°N-85°N) and how cyclones in September and October 2014 differed from the climatological behavior of cyclones in this region. This region is consistent with previous modelling of trajectories originating at the Holuhraun fissure over the course of 48 hours (McCoy and Hartmann, 2015). Not every cyclone in this region interacted with the sulfate aerosol plumes from Holuhraun. To restrict the cyclone population to cyclones that might have been affected by the volcanic sulfate plume the NAME dispersion model was used to simulate the dispersion of both $SO_2$ and sulfate aerosol from Holuhraun. The average near-surface volcanic sulfate aerosol mass predicted by NAME was calculated in the southwest quadrant of cyclones during September and October of 2014. Near-surface sulfate aerosol mass concentrations in excess of 0.1 $\mu g/m^3$ were considered to indicate that a cyclone had interacted with the plume. We note that this is a very low sulfate concentration (see Fig. 3a of McCoy et al. (2018)).

Does our ability to examine cyclones during the eruptive period in relation to their WCB moisture flux reveal any additional information? If cyclones within the 50°W-30°E and 45°N-85°N study region are examined in this context it does appear that CLWP during the eruption might have been higher than the climatological mean. This is shown in Fig. 17. Cyclones during September and October for non-eruption years were used to train a power law fit to WCB moisture flux. The cyclones within the study region during September and October were split into different populations: non-eruption years; the eruption year; and the cyclones that dispersion modelling predicted to have interacted with volcanic sulfur. Two different emissions scenarios were considered in the dispersion model: emission heights set at 1500-3000m; and emissions set at 0-1500m. Anomalies relative to the climatological fit were calculated for each of the four cyclone populations. A t-test with and a non-parametric Wilcoxon rank-sums test were used to calculate if the anomalies relative to WCB in the cyclone population differed significantly from the climatology for non-eruption years. Cyclone LWP during September and October of 2014 was not unusual, but CLWP for cyclones that the NAME dispersion model predicted to have interacted with the plume were anomalously high at 95% confidence (Fig. 17). This was only the case when volcanic emissions were set at 1500-3000m height in the NAME dispersion model. The mean anomaly relative to climatology in this case was $6.51 \pm 4.43$ g/m$^2$.

While in some cases a detectable CLWP signal may be seen in this case study, these results appear to have some sensitivity to the geographical region being considered. If cyclones within 30° latitude are considered, then the cyclones flagged by either emissions height scenario have a significantly different mean CLWP than the climatology (Fig. S14). However, if cyclones spanning the entire latitude region 30°-90°N are considered the presence of several high WCB moisture flux, but relatively low CLWP, cyclones centered between 30°N and 35°N that NAME predicts to have interacted with the plume lead to the population means of CLWP no longer being distinguishable between plume-affected and unaffected cyclone populations at 95% confidence- implying a weak effect from volcanic aerosol within this population (Fig. S15).

We hypothesize that a more extensive investigation of the dispersion of sulfur from Holuhraun would allow a more conclusive identification of which cyclones really did interact with the plume. The number of possible free variables such as plume height; emissions flux from the fissure; and even the

efficiency of aerosol rain-out in the dispersion model complicate this evaluation. A more complete evaluation of the Holuhraun case study in this framework is reserved for a future work.

In summary, we find that cyclones predicted to have interacted with the volcanic plume from Holuhraun using a dispersion model had elevated CLWP relative to the climatological behavior of cyclones in that region- although this result was sensitive to some near-tropical cyclones. Direct comparison to Malavelle et al. (2017) is difficult because the present study examines clouds within midlatitude cyclone systems while Malavelle et al. (2017) aggregated anti-cyclonic and cyclonic regions. It is possible that examination of more pristine, remote marine eruptions such as those shown in Gassó (2008) and examined in Toll et al. (2017) could provide another useful constraint on aerosol-cloud adjustments as they would occur in a relatively low $CDNC_{SW}$ regime, which appears to be quite sensitive to perturbations in microphysics (Fig. 16).

### 4. Conclusions

Analysis of observed covariability between meteorology (as characterized by warm conveyor belt (WCB) moisture flux), warm cloud microphysics (as characterized by cloud droplet number concentration (CDNC)), and cyclone cloud properties is consistent with increasing CDNC leading to an increase in cyclone cloud liquid water path, fractional coverage, and ultimately albedo.

While suggestive, empirical analysis of the observational record cannot prove causality. We support this analysis by performing a set of simulations where CDNC is set at high and low values. The response of CLWP to changes in CDNC in these simulations elucidates the mechanism by which this covariability may be explained and provides support for causality flowing from enhanced CDNC to enhanced CLWP. We hypothesize that rain rates are controlled by the large-scale environment as a consequence of mass conservation in the midlatitudes. When CDNC is increased, a larger LWP is needed to give the same rain rate(Hill et al., 2015;Wood et al., 2009). The LWP adjusts to allow the rain rate to be equal to the moisture flux into the cyclone along the warm conveyor belt. This is hypothesized to lead to the observed covariance between CLWP and CDNC in the WCB region. In summary, based on the idealized simulations we have performed and our analysis of the observational record we propose that *at a higher cyclone CDNC, owing to enhanced aerosol, a larger CLWP is needed to allow rain rate out of*

*the cyclone to match the WCB moisture flux into the cyclone*. It is possible that this effect is not constrained to midlatitude cyclones and we may speculate that clouds in other regimes whose rain rate is the same have a higher LWP with increasing aerosol.

Several elements of our study are consistent with previous modelling and observational studies. An aerosol indirect effect on the clouds in midlatitude storms has been predicted by simulations of the North Pacific (Wang et al., 2014;Joos et al., 2016), and observed in the intensification of the North Pacific storm track(Zhang et al., 2007). Naud et al. (2017) and Grandey et al. (2013) diagnosed covariability between cloud cover and aerosol optical depth in extratropical cyclones. Despite using a completely different set of observations than we utilize here, we agree with the results shown in these studies. These regime-sorted analyses agree with global analysis in Gryspeerdt et al. (2016), which inferred that enhanced CCN enhanced CF in the midlatitudes. We also note that our statement that enhanced CDNC, driven by aerosol emissions, should enhance CLWP, CF, and albedo in cyclones appears to be in contradiction to the analysis conducted by Malavelle et al. (2017), which showed little response in LWP to a transient volcanic emission of sulfur from the 2014-2015 eruption of Holuhraun in Iceland. We performed dispersion model simulations of the volcanic sulfate aerosol to determine which cyclone systems were affected by Holuhraun. This analysis indicated that affected cyclones had high CLWP given their meteorological environment. Sensitivity to assumptions regarding emissions height above the volcanic fissure; sulfur flux from the fissure; and the efficiency with which precipitation removes aerosol in the dispersion model necessitates a more complete validation of this analysis in a future work.

While we suggest that there is a measurable difference in cyclone properties that is driven by microphysical changes, most of the variability in extratropical cyclones is still driven by meteorology. A regression model representation of CLWP as a function of WCB moisture flux and CDNC in the southwest quadrant of the cyclone (CDNC$_{SW}$) explains the majority (more than 60%) of observed variability in CLWP. This regression model allows us to estimate the relative importance of WCB moisture flux and CDNC$_{SW}$ to CLWP variability. The response of CLWP as inferred by the regression model to a standard deviation change in CDNC$_{SW}$ can be a significant fraction of the response to a standard deviation in WCB moisture flux when CDNC$_{SW}$ is low in pristine regions (Fig. 16), consistent

with Carslaw et al. (2013). The average contribution of $CDNC_{SW}$ relative to WCB moisture flux to CLWP variability is estimated to be 20-30%.

While we should not expect to explain all of the variability in CLWP no matter how many predictors we use, it is likely that the explained variability in our regression model could be improved by (1) a more skillful metric for moisture flux into the cyclone, (2) a more accurate observation of $CDNC_{SW}$, or (3) additional information regarding ice and mixed-phase cloud properties. In regards to point (1): we have chosen to predict moisture flux in this way so that we may observe it utilizing microwave radiometers. In regards to points (2) and (3): we note that both of these retrievals are difficult and are likely to improve as the remote sensing community examines them in more depth. Overall, explaining the majority of extratropical cyclone liquid water path variability utilizing two predictors is a useful contribution to our understanding of the midlatitudes.

Comparison of cyclone properties in the top and bottom third of the $CDNC_{SW}$ population correspond to different mean CLWP for a given WCB moisture flux, but also significant changes in cyclone cloud fraction and albedo. All-sky albedo difference between the top and bottom third of all $CDNC_{SW}$ is 0.018±0.002 (95% confidence) when MERRA2 reanalysis SO4 is used to infer CDNC and 0.032±0.002 when CDNC is retrieved by MODIS. These differences in the cyclone-mean albedo observed by CERES contribute to an in-cyclone enhancement in outgoing top of atmosphere shortwave radiation between 4.6 $Wm^{-2}$ and 8.3 $Wm^{-2}$ if the change in albedo is scaled by the annual-mean downwelling shortwave radiation between 30°-80° (Fig. 13).

The results presented here suggest that cloud adjustments in midlatitude cyclones will not reduce the negative forcing resulting from the first indirect effect. A more complete evaluation of aerosol transport into cyclones in the pre-industrial era would be necessary to offer an estimate of the forcing, but it appears that the forcing is negative in order for it to be consistent with observed covariability between microphysics and cloud properties.

**Author contributions**

DTM and PRF planned the paper and wrote the text. DTM performed data analysis and calculations. PRF created simulations in the Unified Model. DPG created the CDNC data set. BJS, AAH, and JMW created

the CASIM microphysics package. GSE created the MAC-LWP dataset. AS ran the NAME dispersion modelling. All authors contributed ideas and helped edit the paper.

### Acknowledgments

MERRA2 data was downloaded from the Giovanni data server. CERES data was downloaded through the ceres.larc.nasa.gov ordering interface. DTM and PRF acknowledge support from the PRIMAVERA project, funded by the European Union's Horizon 2020 programme, Grant Agreement no. 641727. GSE acknowledges support from the NASA MEaSUREs program (via subcontract with the Jet Propulsion Laboratory; Grant no. GG008658).

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

**Table 1 Acronyms used in this article.**

| Acronym | Description |
|---|---|
| CDNC | Cloud droplet number concentration within clouds. |
| $CDNC_{SW}$ | Cloud droplet number concentration average in the south west quarter circle of each cyclone composite. Note that southern hemisphere cyclones are flipped so that their orientation is consistent with northern hemisphere cyclones. |
| CF | Cloud fraction. |
| CLWP | Cyclone liquid water path, defined as the sum of precipitating and non-precipitating liquid. |
| LWP | Liquid water path, the integrated mass concentration of liquid in a column of atmosphere. |
| $N_{acc}$ | Accumulation mode aerosol number concentration. |
| SW | Shortwave radiation |
| SZA | Solar zenith angle. |
| WCB | Warm conveyor belt. |
| WVP | Water vapor path |

**Table 2 The coefficients for equation 1 based on using $CDNC_{SW}$ retrieved by MODIS and inferred from MERRA2 sulfate. Coefficients and 95% confidence intervals (a-d) are listed for each. The number of observations used to train the model is listed as n. The correlation coefficient, r, between predicted and observed CLWP is also listed for each model.**

|  | a | b | c | d | n | r |
|---|---|---|---|---|---|---|
| **MODIS** | 21.79±1.75 | 0.95±0.030 | 0.11±0.0062 | 18.52±3.25 | 37837 | 0.79 |
| **MERRA2** | 19.23±1.28 | 0.86±0.021 | 0.19±0.0062 | 4.53±2.69 | 49361 | 0.82 |

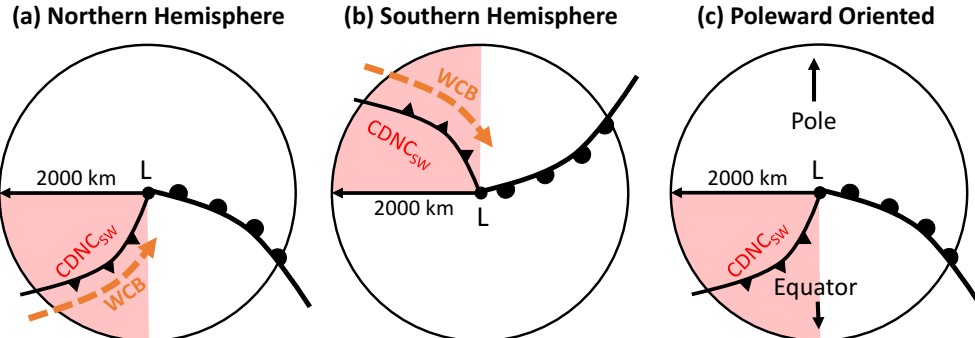

**Fig. 1 Schematic illustrations of an idealized cyclone in the northern hemisphere (a), southern hemisphere (b), and flipped so they are poleward oriented (c). All cyclone composites in this study are presented in poleward oriented format. The approximate location of the cold front is shown with triangles and the warm front is shown with half-circles. The approximate warm conveyor belt (WCB) location is indicated in orange and the low is indicated with an L. The 2000 km radius of averaging is indicated. The averaging region used to calculate CDNC$_{SW}$ is shown using red shading.**

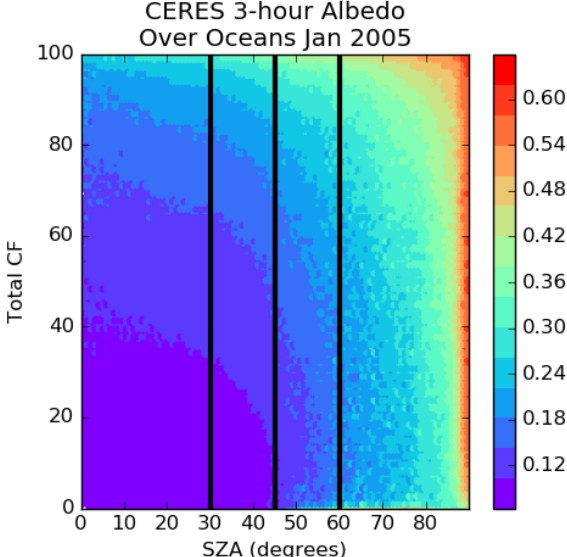

**Fig. 2 CERES 3-hourly albedo over oceans binned as a function of cloud fraction and solar zenith angle (SZA) during January 2005. Above a SZA of 45° a strong dependence of albedo on SZA is seen. The SZA cut offs used in this study of 30°, 45° and 60° are shown with vertical black lines. Example CERES albedo is shown in Fig. S1.**

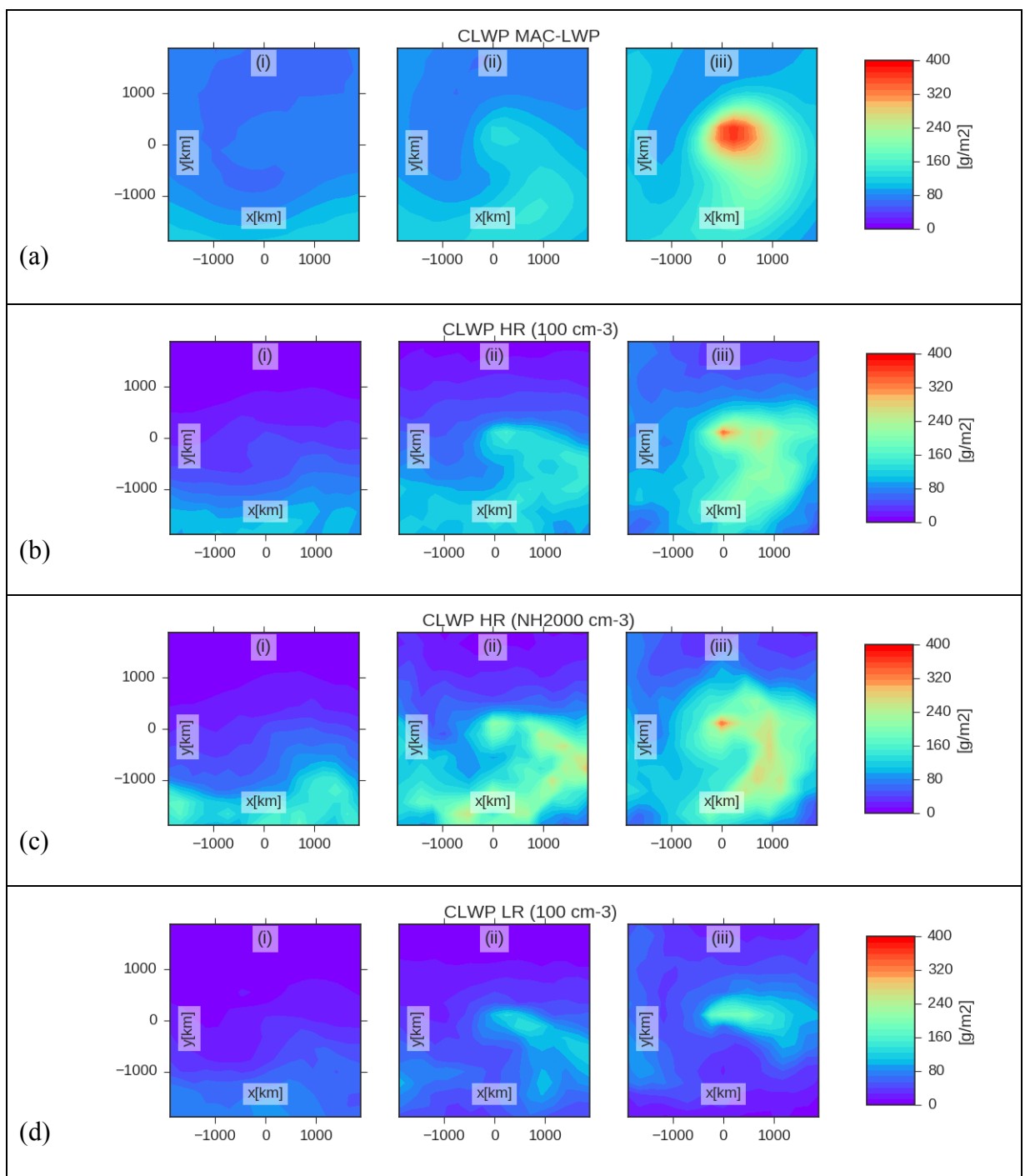

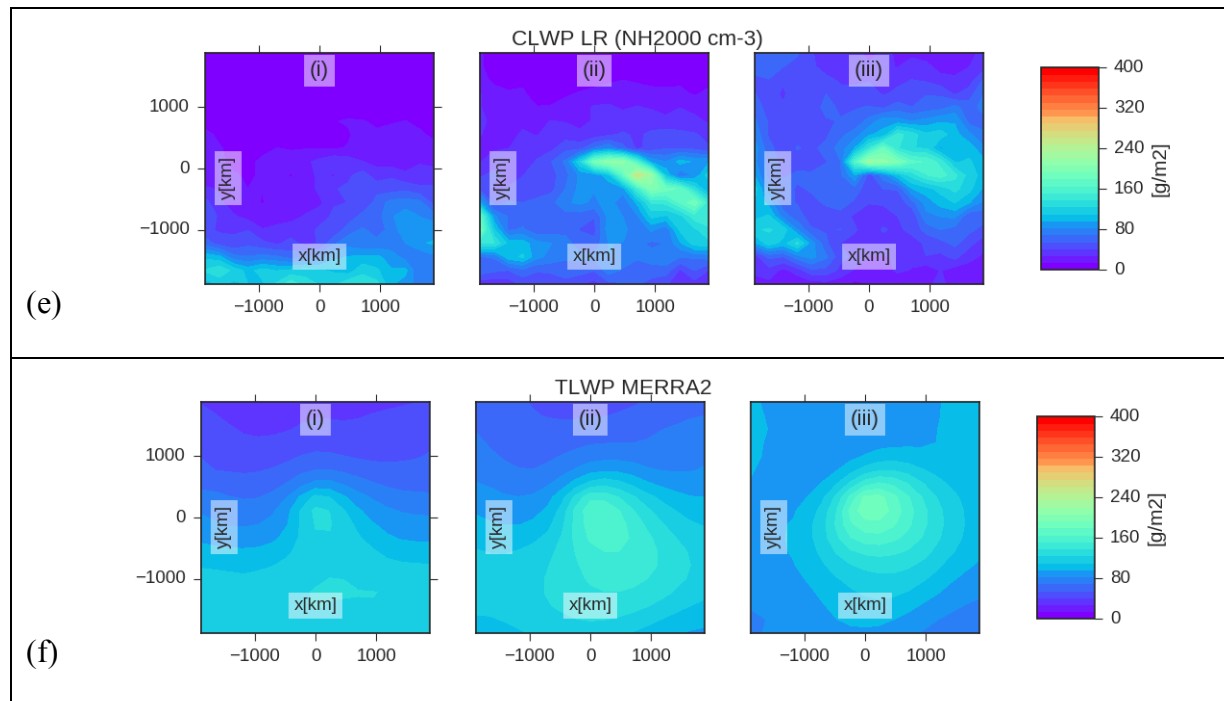

**Fig. 3** **Cyclone composites showing CLWP from (a) MAC-LWP, (b-c) the convection-permitting simulation in the control and enhanced $N_{acc}$ experiments, (d-e) the GCM-surrogate simulation in the control and enhanced $N_{acc}$ experiments, and (f) MERRA2. All composites are shown in three bins of WCB moisture flux so that cyclones with similar meteorology can be compared. The bins are terciles of observed WCB moisture flux. Bins are shown in Fig. 5a and are noted in each subplot by (i)-(iii). It should be noted that the bin edges are not recalculated for the simulations.**

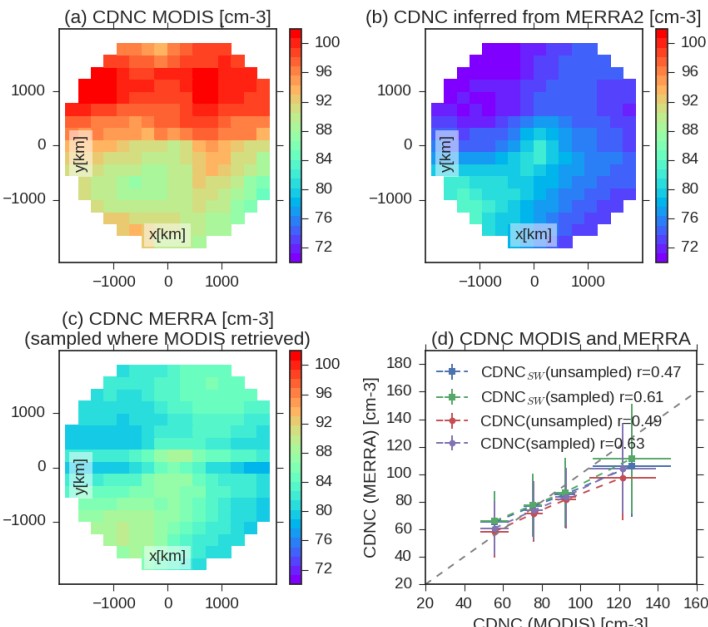

**Fig. 4 Cyclone composited CDNC (a) retrieved by MODIS, (b) inferred from MERRA2, (c) inferred from MERRA2, but sampled so that failed retrievals from MODIS are not used in calculation of the mean composite from MERRA2. Figure (d) Shows the covariability between MERRA2-inferred and MODIS-retrieved cyclone-mean and southwest quadrant (poleward-oriented) CDNC (CDNC$_{SW}$). The CDNC from MERRA2 is plotted on the y-axis and is shown binned by MODIS CDNC. Error bars show one standard deviation over each bin. The correlation between the CDNC (and CDNC$_{SW}$) from MODIS and from MERRA2 for all cyclones in the observational record is noted in the legend. CDNC (and CDNC$_{SW}$) from MERRA2 is calculated when all data is used (unsampled) and when it is sampled to correspond to MODIS.**

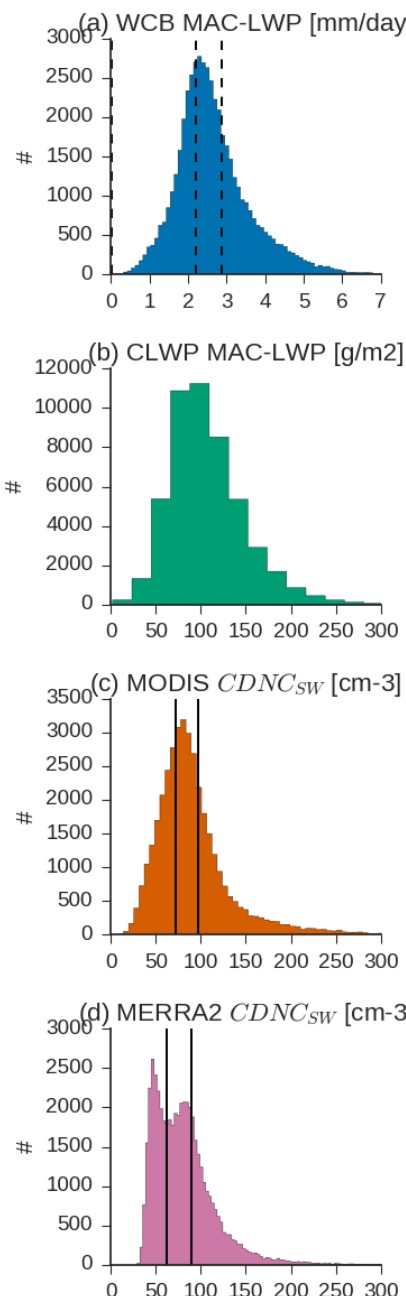

**Fig. 5 Distributions of cyclone-mean properties within the 2003-2015 observational record. Units are noted for each variable. The number of composite cyclones with that value is indicated on the ordinate. Warm conveyor belt (WCB) moisture flux is shown in (a). Cyclone LWP (precipitating and non-precipitating liquid) is shown in (b). Observations and MERRA2-inferred values of CDNC in the southwest quadrant of the cyclone (CDNC$_{SW}$) are shown in (c-d). In (c) and (d) the top and bottom third of distribution are indicated with dashed lines. In (a) edges of the WCB terciles used in Fig. 3 are shown with dashed lines.**

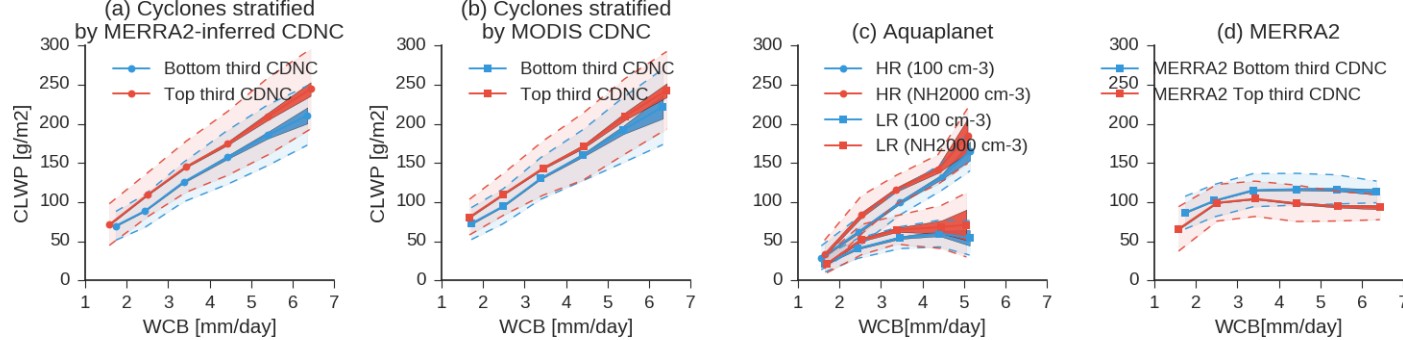

**Fig. 6 Comparison between the dependence of cyclone mean liquid water path (CLWP) on warm conveyor belt (WCB) moisture flux as a function of increasing aerosol. The CLWP is binned by WCB moisture flux. The standard deviations in CLWP across bins are shown as shading with a dashed border. The 95% confidence interval on the mean is shown as thick solid lines. Figures (a) and (b) show MAC-LWP observations from 2003 to 2015 stratified by (a) MERRA2-inferred CDNC$_{SW}$ and (b) stratified by observations of CDNC$_{SW}$ from MODIS. Panel (c) shows the simulated CLWP in a suite of global aquaplanet simulations split into low and high N$_{acc}$ simulations. In the aquaplanet simulations a high aerosol channel is added to the northern hemisphere to investigate the response of cyclone properties and surface N$_{acc}$ is noted in the legend. Aquaplanet simulations are run at convection-permitting (HR) and GCM-surrogate resolution (LR). Panel (d) shows MERRA2 total precipitable liquid stratified by MERRA2-inferred CDNC$_{SW}$. In (a,b, and d) cyclones with CDNC$_{SW}$ in the top and bottom third of retrieved CDNC$_{SW}$ (see Fig. 5) are indicated by red and blue lines.**

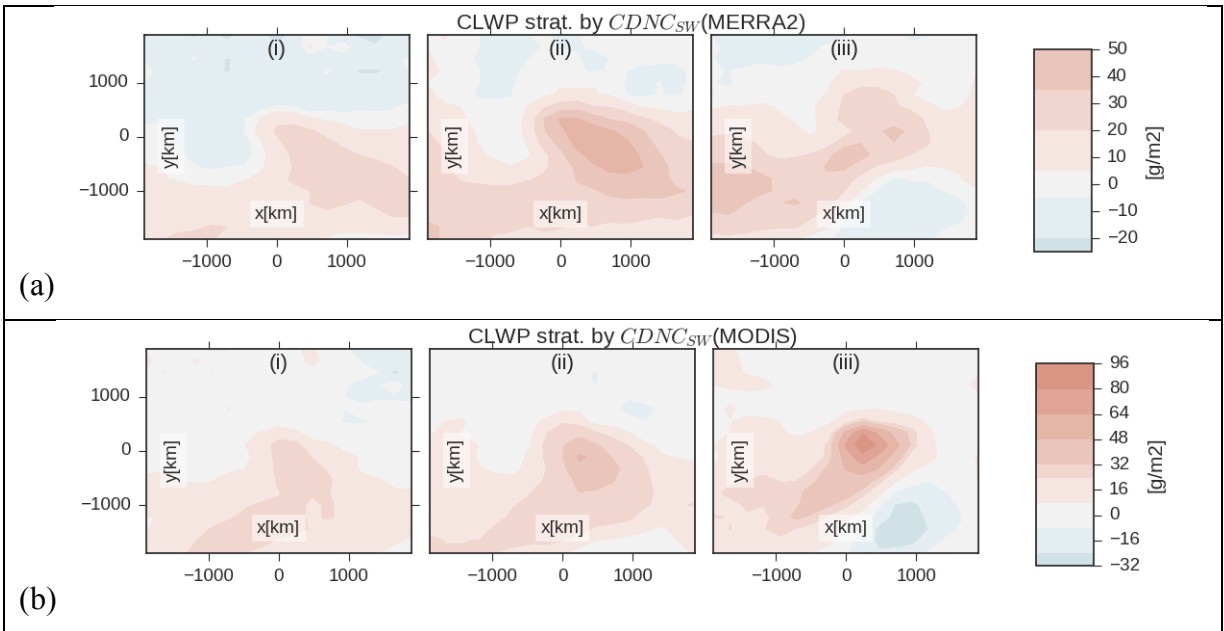

**Fig. 7 The difference in cyclone composited MAC-LWP CLWP between the top and bottom third of CDNC$_{SW}$ inferred from MERRA2 (a) and retrieved by MODIS (b). Composites are shown split into WCB quantiles as in Fig. 3.**

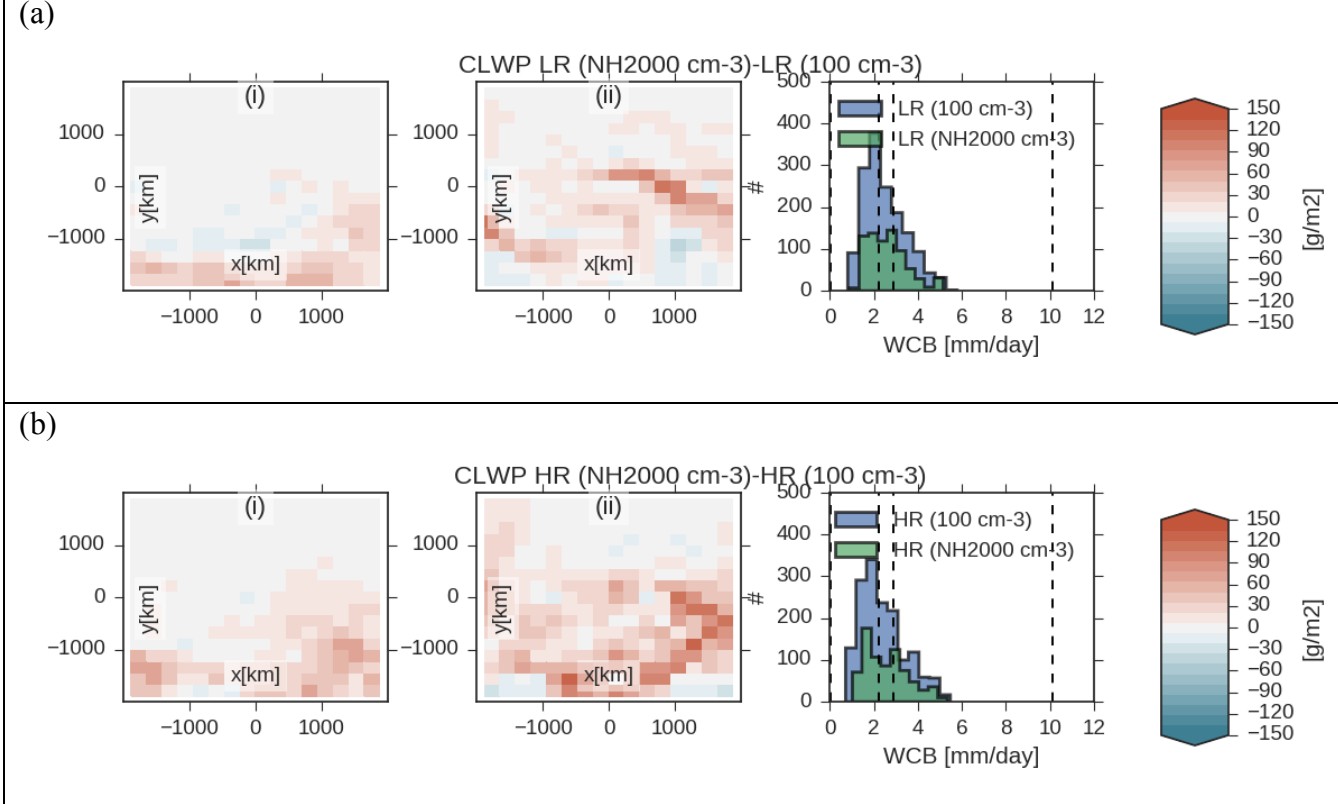

**Fig. 8** The difference in mean cyclone composites of CLWP between the high and low $N_{acc}$ simulations for (a) the GCM-surrogate low-resolution (LR) simulation and (b) the convection-permitting (HR) simulation. Differences in mean cyclone composites for different WCB regimes are shown. It should be noted that the relatively short integration time (relative to the observations) of the simulations did not yield a large number of cyclones in the top tercile of observations and only the first two WCB regimes are shown in contrast to Fig. 7. The distribution of cyclones by WCB in the simulations is shown on the rightmost plot.

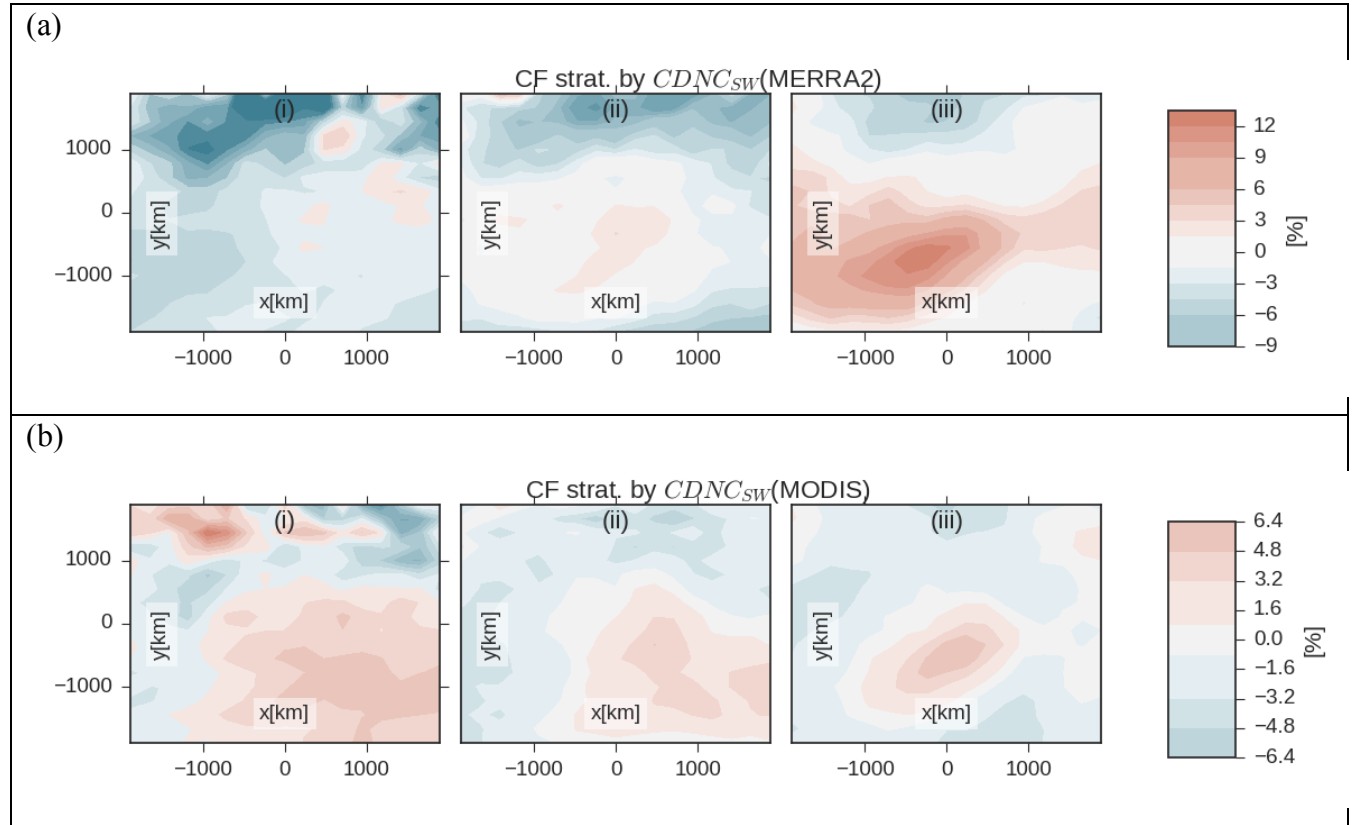

**Fig. 9 As in Fig. 7, but showing differences in cloud fraction.**

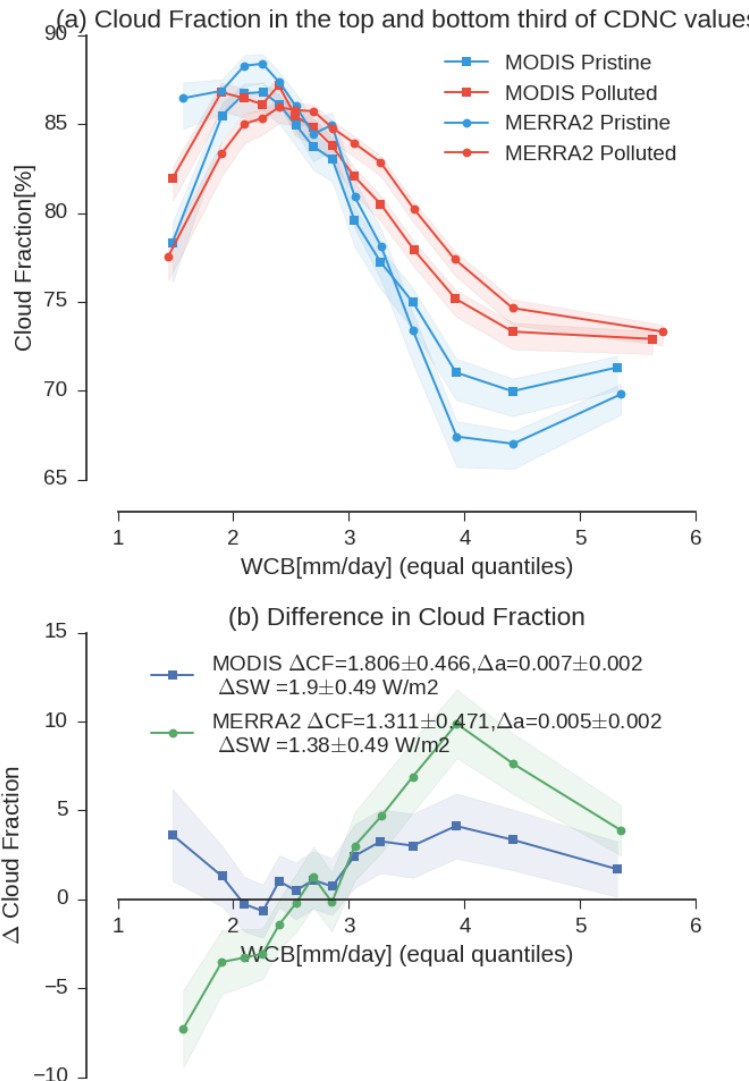

Fig. 10 (a) Cyclone-mean cloud fraction split into 15 equal quantiles and split into high and low CDNC$_{SW}$ populations. Shaded areas show 95% confidence range in the mean in each quantile. The CDNC data set used to partition the cyclone population is noted in the legend. (b) shows the difference between the high and how CDNC$_{SW}$ populations. The difference in cloud fraction between the populations and 95% confidence in the difference are noted in the legend. The relation between cloud fraction and all-sky albedo for the midlatitudes from Bender et al. (2017) is used to approximate the difference in albedo consistent with this difference in cloud cover. The difference in reflected shortwave (SW) is calculated by scaling the albedo by the annual-mean insolation between 30° and 80° latitude.

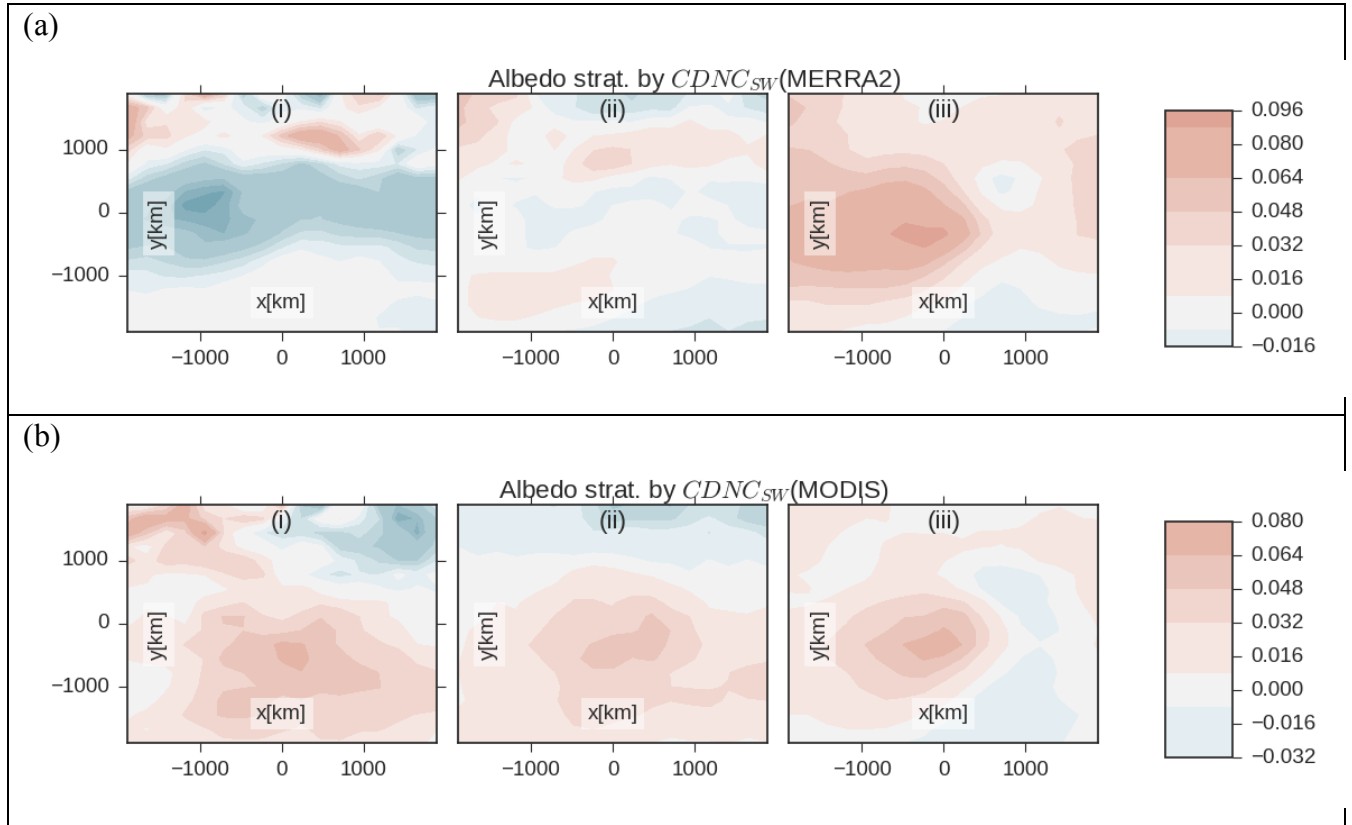

**Fig. 11 As in Fig. 7, but showing differences in albedo.**

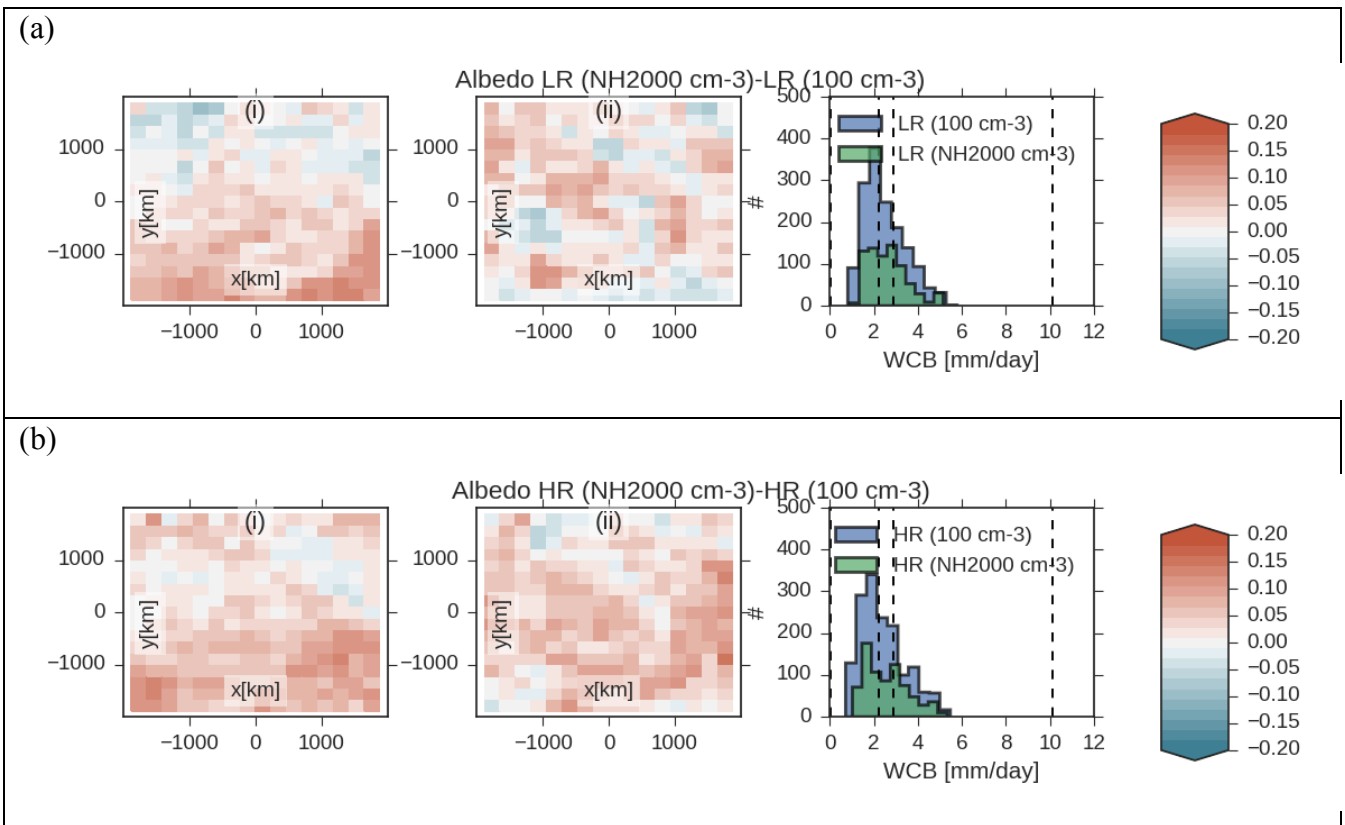

**Fig. 12 As in Fig. 8, but showing differences in albedo.**

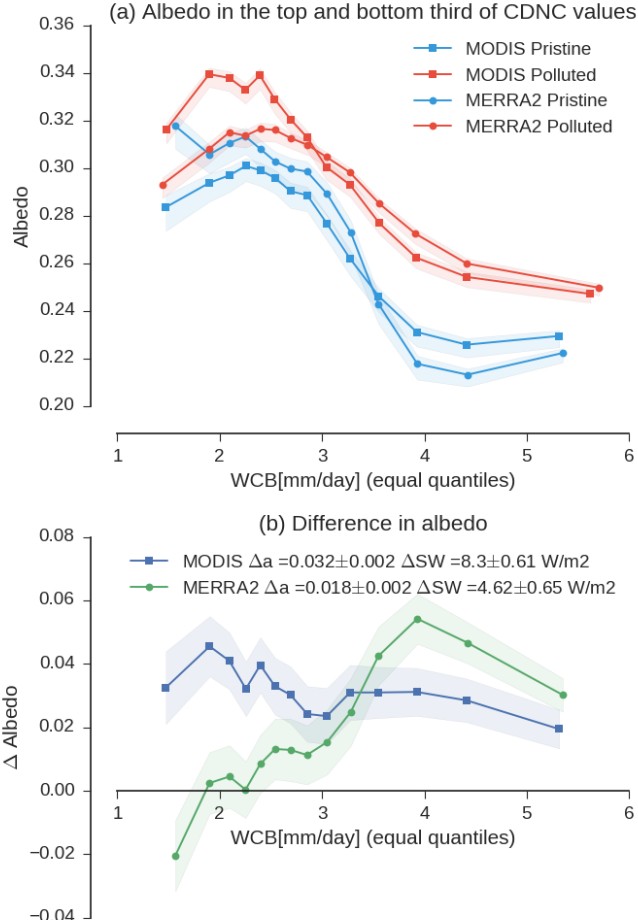

**Fig. 13 (a) Cyclone-mean albedo from CERES as a function of WCB moisture flux. Data is shown binned into equal quantiles of WCB moisture flux and separated into the top and bottom third of retrieved CDNC$_{SW}$. The 95% confidence intervals in the mean are shown using shading. Both MERRA2-inferred and MODIS-retrieved CDNC$_{SW}$ are used to partition the top and bottom third of CDNC$_{SW}$ and are noted in the legend. (b) shows the difference in albedo between the top and bottom third of retrieved CDNC$_{SW}$ as a function of WCB moisture flux. The 95% confidence interval on the difference in each quantile is shown using shading. The mean difference and 95% confidence range on the difference in albedo and estimated reflected SW based on this difference in albedo are noted in the legend. To calculate reflected SW the difference in albedo is scaled by the annual-mean climatological insolation between 30°-80°. Because albedo is a strong function of solar zenith angle (SZA) only 3-hourly measurements with SZA<45° are considered here (similar calculations using cut-offs of 30°, 60°, and 90° are shown in Fig. S9,10,11).**

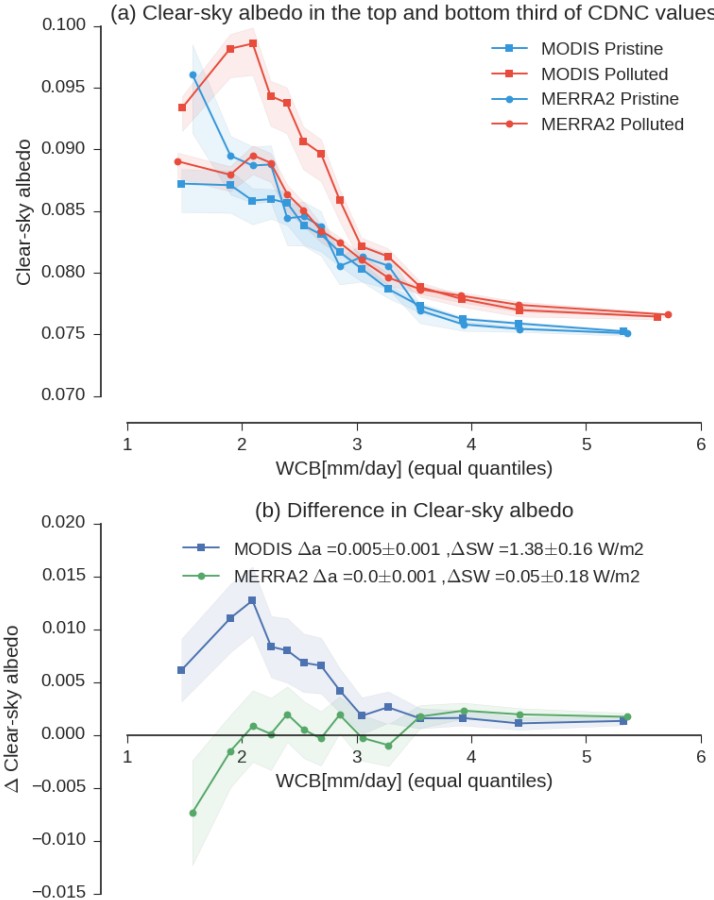

**Fig. 14** As in Fig. 13, but showing differences in cyclone-mean clear-sky albedo. The estimated difference in clear-sky SW is calculated based on the annual-mean insolation between 30-80°.

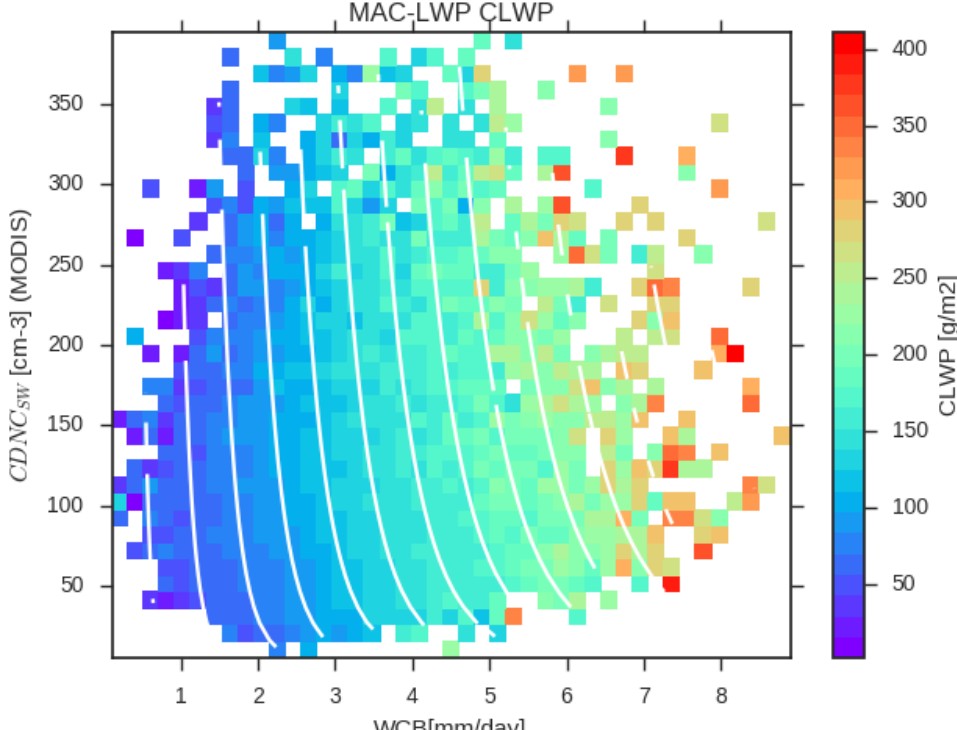

**Fig. 15 The cyclone-mean CLWP in units of g/m² of liquid water observed by MAC-LWP binned as a function of WCB moisture flux and the CDNC$_{SW}$ retrieved by MODIS. Data is binned into equal size bins for the purpose of visualizing the data record. White lines show contours of constant CLWP as predicted by Eq. 1 and the coefficients listed in Table 2. The dependence of CLWP on CDNC$_{SW}$ inferred from MERRA2 is shown in Fig. S12.**

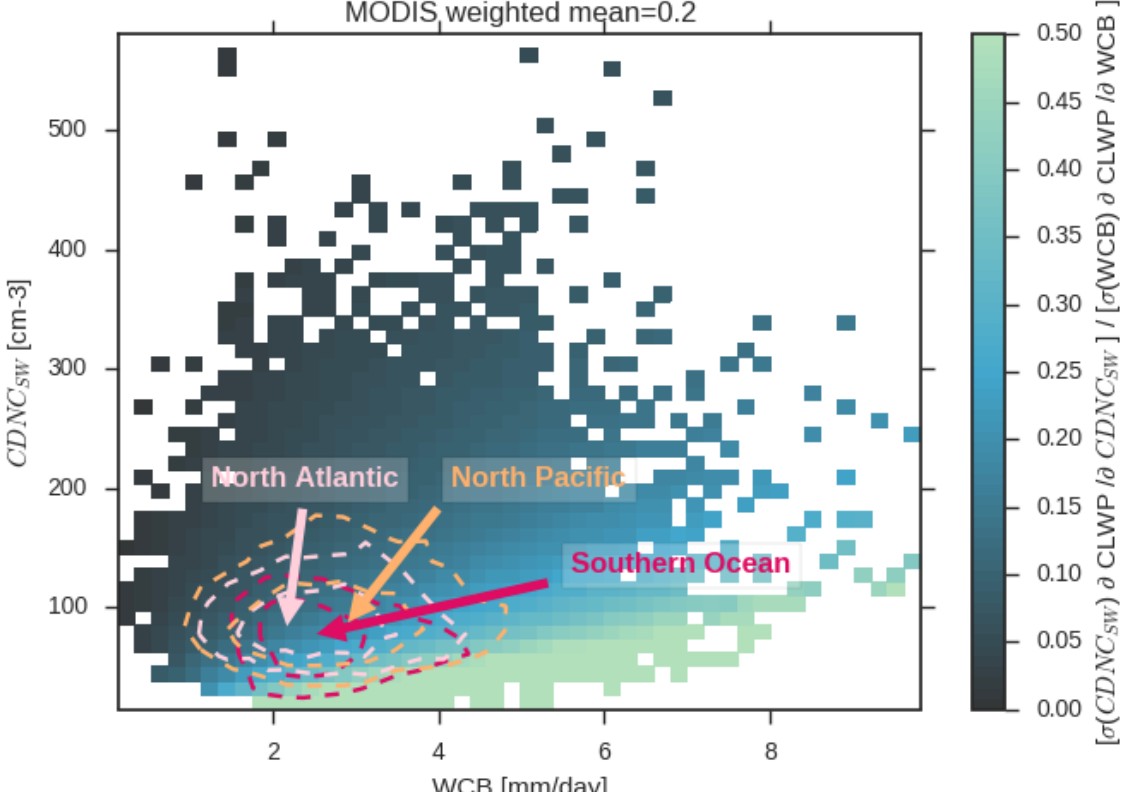

**Fig. 16** The relative contribution to CLWP of perturbations in $CDNC_{SW}$ and perturbations in WCB as estimated using Eq. 1 and the standard deviation of each predictor over the historical record. The regression model was trained using MODIS-retrieved $CDNC_{SW}$ (the same plot is shown for MERRA2-inferred $CDNC_{SW}$ in Fig. S13). The partial derivative of Eq. 1 is taken with respect to each predictor and scaled by the standard deviation of that predictor. The ratio of the partial derivative scaled by standard deviations in each of the predictors is shown using colors. The joint probability distribution of cyclones during the observational record for different ocean regions are roughly indicated using dashed lines. The joint probability distribution of all observations is used to calculate the weighted mean of the fractional contribution of perturbations in $CDNC_{SW}$ and WCB over the range of WCB and $CDNC_{SW}$ in the observational record.

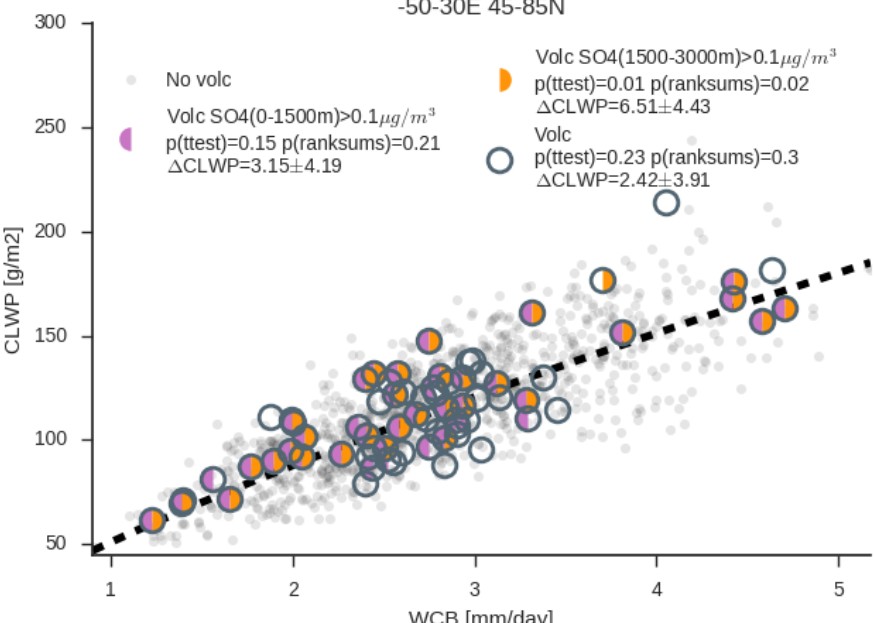

**Fig. 17 The behavior of cyclones as a function of WCB moisture flux are contrasted for September and October during the eruption of Holuhraun and during all other years. The CLWP in the North Atlantic (50°W-30°E, 45°N-85°N) in September and October is shown as a function of WCB moisture flux during all years except 2014 (grey dots), during 2014 (grey circles), and for cyclones that the NAME dispersion model predicted to have interacted with the sulfur plume from Holuhraun (purple and orange half-circles). NAME was used to simulate the Holuhraun plume assuming it extended from 0-1500m at its source (purple) or from 1500m-3000m (orange). The mean sulfate mass within the southwest quadrant of each cyclone was calculated. Only cyclones with a sulfate mass of 0.1μg/m3 were considered to have interacted with the plume. A power law fit of climatological CLWP to WCB moisture flux in the region is shown as a dashed line. This fit was used to calculate anomalies in CLWP for cyclones in September and October except 2014; for September and October of 2014; and for the cyclones that NAME predicted to have interacted with the plume. A t-test and a non-parametric rank sum test were used to evaluate the difference in means between the climatological anomalies in CLWP and the anomalies in 2014 and for the cyclones that NAME predicted had interacted with the plume. The p-values for these tests are given in the legend. Differences in means and 95% confidence intervals, assuming a normal distribution, are also given. Latitude ranges of 35°N-90°N, and 30°N-90°N are shown in Fig. S14 and S15.**

