# Peer review of "Aerosol-mid-latitude cyclone indirect effects in observations and highresolution simulations"

_Atmospheric Chemistry and Physics, 2017_

## Referee Comment (RC1) · Anonymous Referee #1 · 23 Aug 2017

Review of "The aerosol-cyclone indirect effect in observations and highresolution simulations" by McCoy et al.

This manuscript analyzes idealized simulations and observations of aerosol-cloud interactions mid-latitude cyclones. The paper is timely and of a suitable topic for Atmospheric Chemistry and Physics. I like the subject and think it would be valuable for this research to be published in some form, and think the results are quite interesting.

However, I had difficulty with this manuscript. It seems in the end like 3 half finished papers in one. The first part is 'idealized' (aquaplanet) simulations that are not sufficiently evaluated to be able to understand what can be learned. The second part is a set of observations are drawn from several sources, and probably need a bit more discussion. This is especially true because the 'observations' include essentially diagnostic

aerosol fields from a reanalysis system that does not have aerosol-cloud interactions and are used as a very coarse proxy (without explanation really) for aerosols. Finally there are separate simulations (with a different model I believe) of a recent N. Atlantic volcano, the do not seem to have very robust statistics.

This paper needs at least major revisions. Each of the pieces is quite interesting, but they are not well treated in this work, and I think deserve a more careful analysis. It really feels like these three parts were put together from separate projects, and it makes the whole incoherent. I would urge it be broken into more complete pieces.

For example, the observational analysis is very interesting but needs further development. It challenges previous work, but using different data sets and a different sampling method. Similar cyclone sampling with other datasets or means of these data sets without sampling would be very valuable for sorting out whether the different effect (little change in liquid water path with an aerosol proxy) is due to sampling or data. In addition, the blending of MERRA and observations is interesting but problematic.

Secondly, the model simulations are confusing and not fully evaluated. As noted below, I question 'convective permitting' simulations as being in a really bad part of the 'gray zone'.

Finally, the volcano work needs more sensitivity tests and does not seem robust. Introducing a totally different model to this work is a bit complicated and it's not clear what can be learned from the langrangian dispersion model.

The result is that many of the conclusions are not supported by the analysis, and previous results that contradict these conclusions are only mentioned in the introduction and not analyzed. This is not appropriate.

Specific comments are below, but I strongly suggest this be separated be developed into 3 papers. There is a lot of interesting material on the simulations and observations, but they need to be treated properly.

[Figure]

Page 1, L14: surrogate climate model? What is that? This might need a bit more description in the abstract.

Page 1, L30: it might be useful here to separate out the first and second effects in these studies for clarity.

Page 2, L14: I think you need to reference or show some model validation for the cyclone composites. Maybe it is later but it should be noted here.

Page 2, L25: does MERRA2 assimilate AMSRE or MODIS? If so, then these are co dependent.

Also, the statement here about CDNC from a sulfate regression I think means you are baking in an indirect effect.

Page 3, L20: so these simulations prescribe an indirect effect. How was this tuned?

Page 3, L28: how do these emissions compare to Malavelle et al 2017 and Gettelman et al 2015?

Page 4, L8, but also without variable aerosol sources from land that might co vary with meteorology.

Page 4, L9: This is not resolving, it may be barely even permitting. Most mesoscale meteorologists I know would not run a model in this Gray zone between 3km and 10-15km, the latter with a convection scheme, and usually 3km even with some sort of vertically coherent parameterized turbulence.

I hope there is validation of this model somewhere? How about some cyclone composite maps to compare to observations, not just WCB flux.

Page 4, L20: The ice assumption is simplistic. Is there significant ice in your cyclones? It Depends on the temperature in your aquaplanet run.

Page 4, L30: To what extent might that be a consequence of the model formulation.

Perhaps this figure needs to be in the main text?

Page 5, L2: Again, the frozen water path is not aerosol aware. If the LWP but not the IWP changes, then what does this say about how the IWP is formed in the model? This should be assessed by looking at microphysical process rates.

Page 5, L8: What does it mean that the slopes are very different, and the mean values for the model at low WCB strength are off by a factor of 2.

Page 5, L14: However, you presume that the convective permitting simulations can capture the real physics of convection at 6.5km. I don't think this is true.

Many studies with high resolution limited area models (1km horizontally and finer, with higher vertical resolution) note competing processes such as enhanced cloud depth that may offset some of the impacts. Can you comment on this?

Bottom line: without showing that your aquaplanet simulations are realistic (and they look pretty weak from the one evaluation in Figure 1), you have oversold this conclusion.

Page 5, L16: you need to explain the statement that climate sensitivity is too low. I understand the point: but this implies that people adjust the climate sensitivity to match the forcing. It may imply that sensitivity is higher in reality if the forcing is higher. It needs to be rephrased for a reader who does not understand the direction you are going in.

Page 6, L3: I understand that you have a published reference for this, but taking so4 from a reanalysis and trying to use that as a CCN proxy with observations is problematic. If MERRA2 assimilates AMSRE, then you have a potential co-variance problem.

Page 6, L9: I think you need to show cyclone composites in the main text.

Page 6, L10: Didn't Field and wood use a 2 d cyclone composite? Check.

Page 6, L13: Clarify what is observations and what is reanalysis here. It might be

interesting to also look at the CLWP in MERRA without ACI: if it also changes with CCN then there might be an issue with co variance here.

Page 6, L21: These radiation fluxes need uncertainty estimates.

Page 6, L34: except that the analysis says ice is not important but we ignore ice. So if ice was doing something you would not see it. I think it is a bit dangerous to make this assumption. How much IWP Is in the cyclones?

Page 7, L5: Where do all these values come from? Are you just looking at observations + MERRA here or is there something from the model. Also, does CLWP include a rain rain which is part of the WCB metric? What does that mean?

Page 7, L14: I am still confused about this metric. Maybe a better figure would help.

Page 7, L19: what about co-variation?

Page 8, L6: The regression model needs uncertainties on it if you are going to do this. It is not clear whether the results are significant.

Page 8, L9: Can you describe the 'Lagrangian model' NAME in more detail? What met fields drive NAME? Are these MERRA2 cyclones or observed cyclones. I think this needs a more complete treatment. This seems like the beginning of a different paper.

Page 8, L13: I'm not sure that there is a relationship here, or that the anomalies are statistically significant. Shouldn't the line increase with sulfate (i.e. CCN).

Page 8, L18: doesn't this depend on the uncertainty in the regression model (which I do not think you have described)

Page 8, L20: Malavelle et al 2017 also used 10 years of data for a climatological comparison.

Page 8, L20: how are the convective permitting simulations used here? They were aquaplanet?

Page 8, L25: But the sensitivity of convective permitting simulations is higher than observations I think?

Page 8, L29: I think a more thorough sensitivity test is necessary. Also, please check your emissions against earlier work as noted below.

Page 9, L4: but you largely prescribe these effects in idealized models: fixed CCN, no scavenging and infinite sources, fixed relationships. Of course you are going to find this.

Page 9, L6: you have said nothing about radiative forcing over the 20th century. As before: I see your logic but you need to explain it in several sentences with references.

Page 9, L8: But the Holuhraun simulations seem sensitive to emissions, and I'm not sure your statistics are robust. Again, this conclusion does not seem robust

Page 9, L10: I'm not fully clear what is idealized and what is observations in this study,

Page 9, L15: Vague, and of course there is one in the simuLations. The observational part is interesting but needs a more careful treatment.

Page 9, L17: as noted, I think this statement is not defended by the analysis and ignores a lot of previous literature on the complexities of aerosols in convection. It needs a much more thorough analysis of the simulations.

Page 14, Fig2: Are cyclones composited only over the ocean here?

Page 16, Figure 4 needs some discussion of error and/or error bars: are these lines significantly different?

Page 18, Fig6: the presentation is not that effective. It is hard to read the color scale when you use a 2 color gradient. I'm not entirely clear what this is trying to show.

Page 20, Fig8: is this statistically significant? The most 'polluted' storm is not significant. Take out 3 points and there is nothing here. I do not think this is robust.

---

## Referee Comment (RC2) · Anonymous Referee #2 · 29 Aug 2017

This manuscript examines the impact of aerosols on liquid water path in extratropical cyclones using a combination of model simulations and observations. They classify the cyclones based on the strength of the warm conveyor belt, estimated as the product between surface winds and total precipitable water (i.e. moisture flux) averaged in a 2000 km radius region centered on the minimum in sea level pressure that indicates the location of the cyclone. Then they use both MERRA-2 and MODIS information to characterize the aerosol concentration (using the cloud droplet number concentration) in the southwest quadrant of the cyclones and AMSR-E/AMSR-2 liquid water path to evaluate the correlation between cloud droplet number concentration and liquid water path in the storms. They do the same exercise with two aqua planet type simulations of differing resolution and convection treatment (one resolved, the other parameterized).
They propose a model to evaluate the expected cloud liquid water path given the moisture flux and cloud droplet number concentration and use it to evaluate the impact of the Holuhraun eruption on clouds in north Atlantic storms.

There are a number of issues in the manuscript, that would need to be addressed before it can be accepted for publication:

1. There are not enough details and a lot is left for the reader to find in other publications. The accuracy of the various derived observations would be very helpful. The paper is rather succinct, and some figures were moved to a supplement document, as if it were intended as a letter or short publication. I am not sure what the length requirements are with ACP, but it seems to me that all figures could be easily integrated in the main manuscript, and the text should be enriched with more explanations. 2. I am not convinced by the work done with CERES on the impact of the aerosols on the storm albedo (Figure 4 and associated discussion), possibly because there are not enough explanations on how the results are obtained. First it is not clear whether the WCB is constrained in the figure, then there is very succinct discussion on what actually might impact the albedo: with the warm frontal and warm conveyor belt regions of the cyclone dominating the signal and their large amount of high level, mostly ice clouds, there is little signal to be expected from changes in aerosols or low level clouds. In addition, if all cyclones are included, then the CDNC classification can be highly correlated with the cyclone properties and this would mask any impact aerosols direct and indirect effect might have. 3. More details are needed on the work of section 3.2, especially the method, the whole section is confusing and so the importance of the results somewhat degraded 4. In the title, and in the conclusions, the "aerosol-cyclone indirect effect" is mentioned. This is misleading, as this would entail an observational evidence of an impact of aerosols on the cyclone dynamics. This study is about aerosol-cloud interactions in the midlatitude using extratropical cyclones to constrain the large scale environment.

Detailed comments: 5. Page 1, line 21-22: Here you introduce the role of extratropical

cyclones: why not include their role for precipitation in the midlatitude which would be appropriate with the rest of the paper? Reference to the work of Hawcroft et al (GRL 2012), and Catto et al (GRL 2012) would make sense here.

6. Page 2, line 1: here refer to Igel et al., 2013 before Malavelle et al. 7. Page 2, line 20: "the algorithm of Field and Wood (2007)", please provide some details of what it is. 8. Page 2, line 21 onward: when you introduce the CDNC product of MODIS, some details of what it is, its strengths and limitations should be included. The same is true of the other observations/products introduced in this section. There are many observations of the same parameter that are available, so it would be good to justify a bit more why these particular ones are used. For example, cloud fraction is from CERES, why not from MODIS (which the CERES product is in fact retrieved from if I am not mistaken)? How good is the MERRA-2 reanalysis for the sulfate mass concentration product? 9. Page 2, last paragraph: how accurate is this rain water path estimate?

10. Page 3, section 2.2.2: more details on the model would be helpful. What does "NAME" stand for?

11. Page 4, line 28: you write that the cyclone-centered mean is used to obtain the cyclone moisture flux. You should justify this a bit more, as the link to warm conveyor belt is not obvious: this is not the definition used typically. One argument is that cloud and precipitation occur predominantly in the warm conveyor belt and the warm frontal region, so the signal averaged in the entire cyclone region would be dominated by these two areas. Another is that cyclone cloud and precipitation depend strongly on the strength of the cyclone (here characterized by the surface wind) and the amount of moisture ingested in the cyclone (here characterized by the total water path). References are many, but as an example, one could be given to the Field and Wood paper (2007), and/or Bauer and Del Genio (JCLI 2006) and/or Rudeva and Gulev (MWR 2011).

12. Page 5, line 1-2: "indicating that this aerosol-cyclone indirect effect acts through

the warm rain process." This is quite a leap, how do we know this is not model-specific? Also, in Igel et al. (2013), even though the total ice mass in a warm front shows very small changes with an increase in aerosol concentration, the microphysical processes differed such that the aerosol had a compensating impact on vapor depostion and riming efficiency in the mixed phase region. So could it be the case here as well? in which case you might want to change this statement as the indirect effect here would not just act through the warm rain process. And this is not an aerosol-cyclone relation, but an aerosol-cloud relation. 13. Page 5, line 14-16. In figure S3, how do we know that the change in ToA SW flux is not caused by the direct aerosol effect instead of the effect on liquid water path? 14. Page 5, line 22: "allowing for accurate observations", how accurate? There are issues in heavy rain situations with microwave radiometer retrievals of water path and wind speed, which could impact the estimate of the moisture flux and the classification used in the paper. This should be discussed, preferably as early as section 2.

15. Page 6, line 4: "highly consistent" is vague, could you be more quantitative? How is CDNC obtained when clouds are present? How often do you have retrievals in the southwest quadrant? Do you have a threshold on this number below which you do not consider the cyclone in question? 16. Page 6, line 5: because the cold front is moving with respect to the center of the cyclones, sometimes it is in the southwest quadrant, other times in the southeast quadrant, and so the aerosols could be ingested in either quadrant. Have you tried to use the southeast quadrant instead to see if the results change? 17. Page 6, line 9: Figure S4: the two composites look rather different, the two color bars should match to make the comparison easier, and a 1-1 line should be added to the (c) scatter plot. Also, why not add these three plots to Figure 2 and make it a 4-panel figure? 18. Page 6, line 16-17: why not discuss the very obvious differences in the southeast quadrant between observations and model? 19. Page 6, lines 18-26: so here the albedo is estimated with the CERES data, correct? and so is the cloud fraction? how can you have 100% cloud fraction in your cyclone area? you did not explain how this is obtained. Also, if you really have 100%CF, how do you have

MODIS CDNC? Finally the differences between MERRA-2 and MODIS are not that different in magnitude from the differences between high/low CDNC, how significant is this effect on albedo? 20. Page 6, lines 27-33: This is not very convincing, as there is no mention of the moisture flux being constrained, which means that the albedo effect could come from cyclone with low vs. high CDNC having different mean moisture flux and thus different cloud cover caused by this instead.

21. Page 7, the regression model work: I am not sure I se the link between the albedo discussion and this work. Why not introduce this before the albedo work? This regression model is obtained how, based on Figure 5? Finally, I am not sure what the implication of these results is? In particular the very last sentence is unclear, please elaborate. Line 15, and line 17, large and small cyclones do not really mean anything, you do not know anything about their spatial extent. You could use strong/weak maybe, but you would need to specify that this is in term of moisture flux strength, not winds alone.

22. Page 8, line 10: "both simulations", not clear what the two simulations are, only one is mentioned above. 23. Page 8, lines 22-26: This feels out of place, why mention the convection permitting simulations in this context? It just repeats what has been said a few times already about the merits of the high resolution vs GPM-resolution simulations.

24. Page 9, line 15: "an aerosol indirect effect on midlatitude storms" is not accurate, maybe add "clouds" after "storm"

---

## Referee Comment (RC3) · Anonymous Referee #3 · 29 Aug 2017

The manuscript by McCoy et al. investigates aerosol-cloud interactions in midlatitude cyclones over the North Atlantic using modelling and the Hohluraun eruption. I think as such the topic is interesting, but the uncertainty has to be discussed much better. My recommendation is to name the motivation and discuss major limitations of the different approaches such that the scientific evaluation of the work is easier. I hope my comments will be useful for improving the manuscript.

General comments

I recommend to provide more information/discussion on uncertainty and the motivation of some specific choices in the methods for this work. I understand that one would want to highlight the positive results that seem to provide a conclusive story, but I recommend to more openly discuss the uncertainty in such work. In my opinion, mete-

orological variability has a large impact on the perceived aerosol-cloud interaction, no matter whether we look at observations or modelling. My suggestion is to clearly highlight it for supporting an open debate and helping the reader in assessing the results. When we look for instance at the Holuhraun case, we have very few cyclones that have been affected by excessive amounts of sulphate, i.e., 10 cyclones in total according to Fig. 8. Half of these cyclones show an increase in CLWP, but that is within the range of CLWP anomalies that also naturally occur in the absence of SO4 perturbations. The other half of the cyclones with above-threshold perturbations in SO4 show, however, almost no change in CLWP and this includes the cyclone with the largest SO4 perturbation. I would state this explicitly in the text.

In addition to meteorological variability, I wonder how the regionally limited increase in aerosol affects the radiation transfer, thus the temperature gradients and possibly the cyclone/WCB statistics, based on which you construct your argument that aerosol-cloud interaction is the driver of CLWP increases. Have you analysed the changes in the temperature distributions? This would be important for understanding the physical mechanisms behind the model results.

Specific comments

The abstract could be a little longer, e.g., it does not state which model and satellite data has been used, and should more clearly state the uncertainty assessments, e.g., uncertainty in assumptions about the eruption.

p.1, l. 20: "liquid water amount and thus the albedo" The cloud albedo depends on the number and size of droplets. I also wonder whether "constraining predictions of the 21st century warming" is a good word choice as the warming will depend not only on the physics, but also on the socio-economic development. As such we will always have a spread in long-term projections into the future. In any case, citing of references would be useful here.

p.1, l. 22: "thermal contrasts" alone are not enough to form a cyclone. It might be best

to just delete that sentence.

p.3, l.7: I am a little bit surprised that both the configurations with and without explicit convection use the same vertical resolution. Maybe you can explain why you have made that choice. Would you expect the results to differ when you also change the vertical resolution?

p.3, l. 14-15: Do you mean that the exponential decay starts at the surface or above 5km? Both seems to be tricky, unless there is observational evidence for it, since aerosol is typically well mixed in the boundary layer, but only few places have a deep BL of 5 km. Maybe use cm-3 instead of /cc to be consistent with your results section.

p.3, l.18: Please clarify "non-interacting". I guess you mean that no complex aerosol parameterisation is coupled to the atmospheric model, but you prescribe the aerosol concentration as function of vertical velocity and let the aerosol interact with the radiation and clouds in the model (such a setup could also be interpreted as "interacting").

p.3, l.19: Is there a reason why you have chosen to increase the aerosol just in this channel? Such a setup generates a steep (artificial) gradient in aerosol that might change your temperature gradients and thereby the cyclones.

Section: 2.2.2: I think if you could add the uncertainty range of these estimates and maybe even systematically test the effect of such a range on your results, the work could be a much better contribution. Later in the results section you touch on that type of uncertainty. Maybe you could motivate it here already.

p.4, The first paragraph is partly redundant with the method section. Maybe you can merge the text.

p.5, l.1-2: Is this due to the simple parameterisation that you have implemented into the model? In either case I would mention it here again, because the way it is currently written suggests that what your model tells us is a fact and that fact "warm rain process" seems to contradict what one would expect for precipitation formation in midlatitude

cyclones (in reality).

p.6, l.8-9, Fig. 2: It is not clear why you get large CDNC in the cyclone center, typically a decline going outward, but than again an increase in CDNC to the southwest. Could you argue that this is something you would expect? Here I would also want to read more about the comparison of the CDNC of MERRA and MODIS to judge the quality of the re-analysis.

p.7, l.26: "enhancement of cyclone properties" I would speak of changes of cyclone properties.

p.8, l.5-6: MERRA assimilates, however, observations that have experienced a potential effect of the aerosol on the meteorology. So, this might not be as conclusive as one would hope.

p.9, l.18-20: These are big implications, but how could we know that we would get the same answer when we used other models or other volcanic eruptions, given the uncertainties and variability?

---

## Author Comment (AC1) · 30 Nov 2017

**Response to reviewers**

We thank the reviewers for their careful reading of our manuscript. Several of the comments by reviewers helped us significantly improve the paper so that is great. In response to their comments we have made several significant changes to our paper.

1. We have replaced the AMSR data set with the MACLWP data set. This reduces uncertainty related to diurnally averaging the dataset. Full details of the improvement in MACLWP data relative to AMSR data are given in Elsaesser et al. (2017).
2. We have expanded the methods section to more completely describe what has been done in the paper.
3. Based on comments from the reviewers we have found that use of the albedo from CERES from all solar zenith angles (SZAs) was introducing a substantial bias due to SZA effects. This effect is discussed in detail in Bender, Engström, Wood, and Charlson (2017) and we have taken steps to show sensitivity to this effect in our analysis.
4. Based on further analysis of the Holuhraun eruption we have decided to remove this case study from our paper for more in-depth evaluation at a future time. Our previous results were based on AMSRE/2 data, which changed overpass time in the transition from AMSRE to AMSR2, spuriously creating differences in cyclone properties within the data record. Use of MACLWP data shows that our previous analysis is less robust once the diurnal cycle is more carefully accounted for.

We will answer each reviewers comments in detail below- reviewer comments are in italics.

**Reviewer 1:**
*The second part is a set of observations are drawn from several sources, and probably need a bit more discussion. This is especially true because the 'observations' include essentially diagnostic aerosol fields from a reanalysis system that does not have aerosol-cloud interactions and are used as a very coarse proxy (without explanation really) for aerosols.*

We did not explain this aspect of the paper sufficiently. In fact, in the paper we emphasized the reanalysis aerosol giving the impression that our results are completely dependent on this. We have rewritten the methods and paper to show that our results are not qualitatively sensitive to whether we use MODIS or MERRA2 to look at CDNC. We have also tried to provide a more thorough evaluation of how MERRA2, MODIS and the OMI SO2 product covary in McCoy et al. (2017) to support these observations further. Of course the paper should stand on its own and our results using MODIS and MERRA2 are presented in parallel now. We have also expanded our discussion of the data sets and methodology.

*Secondly, the model simulations are confusing and not fully evaluated. As noted below, I question 'convective permitting' simulations as being in a really bad part of the 'gray zone'.*

Thank you- good point. We have inserted a discussion regarding pros and cons of the simulation resolution. Overall, we find the same results in a qualitative sense at both the convection permitting resolution and the GCM resolution so our results are not hugely sensitive to our choice of resolution and the contrast between the GCM and 6.8km resolution demonstrates how sensitive a single model might be to these choices in terms of its cyclone behavior.. We have

added a reference to a paper showing that within a resolution range of 1-16km the mean field statistics of parameters such as broadband fluxes and water path do not change much (Field et al., 2017).

*Finally, the volcano work needs more sensitivity tests and does not seem robust. Intro-ducing a totally different model to this work is a bit complicated and it's not clear what can be learned from the langrangian dispersion model.*

See comment above, we have removed this work as we believe our results are robust without it and the analysis appears to be sensitive to how much ocean coverage is required in the cyclone composites and how far we assume the lagrangian dispersion model is accurate for. We will return to this analysis in a future study.

*Page 1, L14: surrogate climate model? What is that? This might need a bit more description in the abstract.*

We feel that the main focus of our paper should not be examining the contrast between the GCM-resolution and convection permitting simulation because this contrast may not be highly representative of the population of GCMs and this has been deleted from the abstract. The simulation discussion in the methodology has been expanded. The surrogate climate model is just a coarsened (140km) version of the high resolution model with parametrized convection switched on. The use of parametrized convection at coarse resolution makes it more similar to a current climate GCM than the 6.8km with explicit convection. The coarse model with parametrized convection provides a convenient comparison to suggest what climate model might do, but without too many changes that make it difficult to disentangle the cause of the difference.

*Page 1, L30: it might be useful here to separate out the first and second effects in these studies for clarity.*

We noted that the first group of papers address the first indirect effect and the second group address the lifetime effect. We have rewritten the introduction to try and make it clear which papers refer to each effect.

*Page 2, L14: I think you need to reference or show some model validation for the cyclone composites. Maybe it is later but it should be noted here.*

We have added comparisons of the cyclone composites across WCB regimes between the observations, simulations, and MERRA2. There are big differences, although the convection permitting simulations look generally ok.  Since the WCB moisture flux- rain rate relation appears to be a feature of GCMs and observations (Field, Bodas-Salcedo, & Brooks, 2011) and we hypothesize that the inhibition of rain via aerosol-cloud interactions leads to divergence between low and high CCN simulations the mean-state of the simulated cyclones does not have to perfectly match the observations. The comparisons in the paper discuss the differences and similarities between the observations and simulations now.

*Page 2, L25: does MERRA2 assimilate AMSRE or MODIS? If so, then these are co dependent.*

This is a very good point- we have re-examined the MERRA2 documentation. In (McCarty et al., 2016) a list of assimilated radiances and data is made. SSM/I rain rates are retrieved through 1987-2009. SSM/I radiances are also used for this period and are cloud-cleared following (Derber & Wu, 1998). AOD from MODIS is assimilated in the aerosol analysis, but cloud properties are not (Randles et al., 2016). We have noted this in the text, but we also note that all else being equal higher LWP should imply a higher rain-rate and lower aerosol mass so the co-dependence should lead to anti-correlation between LWP and SO4. We have added analysis of the MERRA2 total precipitable liquid water path and indeed see that this is the case. That is to say, higher CDNC (from the MERRA2 SO4) corresponds to lower CLWP cyclones and vice versa. Thank you for suggesting this.

*Also, the statement here about CDNC from a sulfate regression I think means you are baking in an indirect effect.*

As noted above we use MODIS and MERRA2 SO4 as proxies for CDNC and both direct observations and the MERRA2 proxy produce similar results.

*Page 3, L20: so these simulations prescribe an indirect effect. How was this tuned?*

This indirect effect was not really tuned. In a sense the simulations are more of a 'fixed CDNC' set of simulations than a 'fixed CCN' simulations (as in Lu and Deng (2015)). Another Twomey-type activation scheme would have produced results that show the same qualitative result. We have added some verbage to the text to explain this more clearly. We have added a note that while the CDNC is sensitive to both vertical motions and CCN, the sensitivity to CCN contributes the vast majority of variability in CDNC. Thanks.

*Page 3, L28: how do these emissions compare to Malavelle et al 2017 and Gettelman et al 2015?*

Although this material has been removed we note that these emissions are somewhat more in keeping with observations than the fixed 40kT scenario in Malavelle 2017. The time varying flux is somewhat more realistic based on observed fluxes (Schmidt et al., 2015).

*Page 4, L8, but also without variable aerosol sources from land that might co vary with meteorology.*

The aerosol in the aquaplanet was constant so it can't vary.

*Page 4, L9: This is not resolving, it may be barely even permitting. Most mesoscale meteorologists I know would not run a model in this Gray zone between 3km and 10- 15km, the latter with a convection scheme, and usually 3km even with some sort of vertically coherent parameterized turbulence.*

*I hope there is validation of this model somewhere? How about some cyclone com- posite maps to compare to observations, not just WCB flux.*

Please see comments above regarding model resolution and additional plots of composites.

*Page 4, L20: The ice assumption is simplistic. Is there significant ice in your cyclones? It*

*Depends on the temperature in your aquaplanet run.*

There is significant ice, but it does not seem to be a strong function of WCB moisture flux or aerosol (see Fig S2 of the original SM). The ice number is linked to 'Cooper temperature dependence'. We do not change the ice number representation when we change the CCN. We have added a note to this effect in the text in section 3.1.

*Page 4, L30: To what extent might that be a consequence of the model formulation.*

*Perhaps this figure needs to be in the main text?*

This is a common function of GCMs and observations (Field et al., 2011; Field et al., 2008) and not a function of this particular model formulation and appears to be a consequence of mass conservation within the midlatitude cyclone systems.

*Page 5, L2: Again, the frozen water path is not aerosol aware. If the LWP but not the IWP changes, then what does this say about how the IWP is formed in the model? This should be assessed by looking at microphysical process rates.*

Please note that the aerosol exponentially decreases with height. IWP is aerosol aware, but the assumption here is that aerosol is not uniform with height throughout the atmosphere.

*Page 5, L8: What does it mean that the slopes are very different, and the mean values for the model at low WCB strength are off by a factor of 2.*

Do you mean the slopes in the observations or the simulations? It seems like the model is more sensitive to WCB at high aerosol, which seems reasonable if precipitation is being inhibited. The model mean is much lower than the observations we now note this in the paper and this is likely just because we have not used a cloud scheme, which would add additional complexity and make it harder to understand the differences between the low- and high-resolution simulation. We should note that the goal of this study was not to create a simulation with perfectly realistic cyclones, but to examine how cyclones respond to aerosol in a qualitative sense and use that insight to analyze observations in a sensible way.

*Page 5, L14: However, you presume that the convective permitting simulations can capture the real physics of convection at 6.5km. I don't think this is true.*

We don't say that the convection permitting simulation is perfectly representing convection in this section. We note that the convection parameterization, which is not aerosol-aware, is not acting on any of the clouds.

*Many studies with high resolution limited area models (1km horizontally and finer, with higher vertical resolution) note competing processes such as enhanced cloud depth that may offset some of the impacts. Can you comment on this?*

We argue that our simple model formulation appears to be consistent with the observations. We

have added a note to this effect to the discussion. Thank you.

*Bottom line: without showing that your aquaplanet simulations are realistic (and they look pretty weak from the one evaluation in Figure 1), you have oversold this conclu- sion.*

As suggested we have added more evaluation of the aquaplanet. The goal of the aquaplanet is to offer some sort of framework to analyze the models in. The main framework is the strong dependence of rain rate and by extension CLWP on WCB moisture flux. This appears to be a feature of many GCMs (Field et al., 2008). In our idealized simulations we have been able to run high and low aerosol perturbation simulations. When observations are analyzed in the same way high and low CDNC cyclones behave in a qualitatively similar sense to both our high resolution and low-resolution simulations. Based on Reviewer 1's careful analysis we have tried to prune back the conclusions and shore up the methodology.

*Page 5, L16: you need to explain the statement that climate sensitivity is too low. I understand the point: but this implies that people adjust the climate sensitivity to match the forcing. It may imply that sensitivity is higher in reality if the forcing is higher. It needs to be rephrased for a reader who does not understand the direction you are going in.*

This comment has been removed. Given the spread in aerosol-cloud forcing it is likely many models also have too strong an aerosol cloud indirect effect. Admittedly, if all climate models without aerosol-aware convection miss this effect this does make it a systematic error, but going through explaining this is convoluted.

*Page 6, L3: I understand that you have a published reference for this, but taking so4 from a reanalysis and trying to use that as a CCN proxy with observations is problem- atic. If MERRA2 assimilates AMSRE, then you have a potential co-variance problem.*

See comment in regards to this above.

*Page 6, L9: I think you need to show cyclone composites in the main text.*

See comment related to P2 L14 above- we have added figures for models and observations to the text.

*Page 6, L10: Didn't Field and wood use a 2 d cyclone composite? Check.*

They did, as does this paper. I don't understand how it relates to the material here. High and low aerosol cyclones are for the cyclone as a whole, not for a specific part of the cyclone.

*Page 6, L13: Clarify what is observations and what is reanalysis here. It might be interesting to also look at the CLWP in MERRA without ACI: if it also changes with CCN then there might be an issue with co variance here.*

Thank you for your suggestion. We did this and MERRA2 implies the opposite of the observations.

*Page 6, L21: These radiation fluxes need uncertainty estimates.*

Great point- one of the major things that we have changed in the paper is the radiation calculation. We realized that the dependence of albedo on solar zenith angle was yielding unrealistic results (high albedo cyclones having the lowest water path because they had very high SZA). We have added additional analysis exploring this to the paper. We have also added uncertainty to our analysis.

*Page 6, L34: except that the analysis says ice is not important but we ignore ice. So if ice was doing something you would not see it. I think it is a bit dangerous to make this assumption. How much IWP Is in the cyclones?*

I am not clear what you mean by this, but we do show IWP for the simulations in the SM.

*Page 7, L5: Where do all these values come from? Are you just looking at observations + MERRA here or is there something from the model. Also, does CLWP include a rain rain which is part of the WCB metric? What does that mean?*

This is from the observations. As noted in the methods section the CLWP is precipitating and non-precipitating liquid. We have chosen to look at this because it is what the microwave radiometer is sensitive to.

*Page 7, L14: I am still confused about this metric. Maybe a better figure would help.*

Thank you- we have remade this figure and tried to explain it better in the text. The idea was to point out that in the context of the regression model not all cyclones are equally sensitive to aerosol forcing. Not a hugely surprising result and really just restating Carslaw et al. (2013), but we thought it was important. We have added notation to show where different ocean regions fall on this figure.

*Page 7, L19: what about co-variation?*

This is just the weighted mean of the figure so it takes covariation into account. We have noted in the text that this analysis is hopefully  illustrative.

*Page 8, L6: The regression model needs uncertainties on it if you are going to do this. It is not clear whether the results are significant.*

Removed- see above.

*Page 8, L9: Can you describe the 'Lagrangian model' NAME in more detail? What met fields drive NAME? Are these MERRA2 cyclones or observed cyclones. I think this needs a more complete treatment. This seems like the beginning of a different paper.*

Removed- see above.

*Page 8, L13: I'm not sure that there is a relationship here, or that the anomalies are statistically significant. Shouldn't the line increase with sulfate (i.e. CCN).*

See above- removed.

*Page 8, L18: doesn't this depend on the uncertainty in the regression model (which I do not think you have described)*

See above, removed

*Page 8, L20: Malavelle et al 2017 also used 10 years of data for a climatological comparison.*

Sorry- we phrased that poorly. Removed.

*Page 8, L20: how are the convective permitting simulations used here? They were aquaplanet?*

This material has been removed.

*Page 8, L25: But the sensitivity of convective permitting simulations is higher than observations I think?*

This material has been removed.

*Page 8, L29: I think a more thorough sensitivity test is necessary. Also, please check your emissions against earlier work as noted below. Page 9, L8: But the Holuhraun simulations seem sensitive to emissions, and I'm not sure your statistics are robust. Again, this conclusion does not seem robust Page 20, Fig8: is this statistically significant? The most 'polluted' storm is not signifi- cant. Take out 3 points and there is nothing here. I do not think this is robust.*

Agreed- we found sensitivity to our assumptions in this analysis. This does not affect the conclusions in the remainder of the paper.

*Page 9, L4: but you largely prescribe these effects in idealized models: fixed CCN, no scavenging and infinite sources, fixed relationships. Of course you are going to find this.*

This is a good point. Our model by default tends toward a positive lifetime effect on LWP, although this is not a given because of interactions with other clouds and the environment – see Fig 6 and 7 of Miltenberger et al. (2017). We have added to text to make this clearer.

*Page 9, L6: you have said nothing about radiative forcing over the 20th century. As before: I see your logic but you need to explain it in several sentences with references.*

The argument here is just that the forcing is sufficiently large to be non-negligible. We have expanded our discussion to make it clearer that this is all we are saying.

*Page 9, L10: I'm not fully clear what is idealized and what is observations in this study,*

We have tried to expand our methods and discussion to make the analysis clearer.

*Page 9, L15: Vague, and of course there is one in the simuLations. The observational part is interesting but needs a more careful treatment.*

See above. We have tried to more clearly articulate the analysis and acknowledge that the model set up we have will enhance LWP with enhanced CCN, all else being equal.

*Page 9, L17: as noted, I think this statement is not defended by the analysis and ignores a lot of previous literature on the complexities of aerosols in convection. It needs a much more thorough analysis of the simulations.*

We have clarified this statement to reflect the fact that we have shown this via observations, for the first time.

*Page 14, Fig2: Are cyclones composited only over the ocean here?*

That is correct, we have expanded our methodology to more clearly articulate this.

*Page 16, Figure 4 needs some discussion of error and/or error bars: are these lines significantly different?*

We have updated our analysis of the albedo effect.

*Page 18, Fig6: the presentation is not that effective. It is hard to read the color scale when you use a 2 color gradient. I'm not entirely clear what this is trying to show.*

See above. We have tried to more clearly explain the figure.

**Reviewer 2:**
*1. There are not enough details and a lot is left for the reader to find in other publi- cations. The accuracy of the various derived observations would be very helpful. The paper is rather succinct, and some figures were moved to a supplement document, as if it were intended as a letter or short publication.*

This is correct- we have tried to more clearly articulate our research in the revised submission.

*2. I am not convinced by the work done with CERES on the impact of the aerosols on the storm albedo (Figure 4 and associated discussion), possibly because there are not enough explanations on how the results are obtained. First it is not clear whether the WCB is constrained in the figure, then there is very succinct discussion on what actually might impact the albedo: with the warm frontal and warm conveyor belt regions of the cyclone dominating the signal and their large amount of high level, mostly ice clouds, there is little signal to be expected from changes in*

*aerosols or low level clouds.*

This is a good point- we have significantly reworked our evaluation of the albedo effect, primarily due to difficulties resulting from SZA bias (see previous reviewer comments). The reviewer also makes a good point regarding ice cloud, however, assuming that the ice cloud effect on albedo is more or less randomly distributed and is unaffected by CCN then it should just add variability to the populations of low and high CCN cyclones. We have compared albedo from the high and low CCN cyclones in the SM of the original submission and we show that most of the effect in the observations and simulations is in the post cold frontal clouds. This is consistent with the proposed effect (eg CCN affecting liquid cloud cover). We have expanded this analysis and moved it to the main text.

*In addition, if all cyclones are included, then the CDNC classification can be highly correlated with the cyclone properties and this would mask any impact aerosols direct and indirect ef- fect might have.*

This is a good point regarding the direct effect. To try and better understand the possibility that we are somehow aliasing this effect in we looked at cyclone composites of CF and CLWP differences between high and low CDNC cyclones. We see that there is a general agreement between the regions where CF and CLWP enhance and where the all-sky albedo increases. Of course the partitioning of this radiative effect into components owing to changes in CF, LWP, CDNC is difficult, but it does seem like changes in cloud macrophysical properties and changes in albedo are happening in the same area so that supports the idea that these changes are driving the changes in albedo.

*3. More details are needed on the work of section 3.2, especially the method, the whole section is confusing and so the importance of the results somewhat degraded*

This has been expanded- thanks.

*4. In the title, and in the conclusions, the "aerosol-cyclone indirect effect" is mentioned. This is misleading, as this would entail an observational evidence of an impact of aerosols on the cyclone dynamics. This study is about aerosol-cloud in- teractions in the midlatitude using extratropical cyclones to constrain the large scale environment.*

Admittedly it's an aerosol-cloud effect, but we are using the cyclone as a constraint to order the meteorology so it seems reasonable to call it this since we are referring to the cyclone as the clouds that compose it. Please note that we have altered the main body of the text to reflect that it is the clouds within the cyclone changing to avoid any confusion, but we feel that completely spelling this out in the title would make it clunky.

*Detailed comments: 5. Page 1, line 21-22: Here you introduce the role of extratropical cyclones: why not include their role for precipitation in the midlatitude which would be appropriate with the rest of the paper? Reference to the work of Hawcroft et al (GRL 2012), and Catto et al (GRL 2012) would make sense here.*

Thank you. These are very good references to add. One thing we did find is that the precipitation is controlled by the WCB, not aerosol so the effect we show should not alter the total rainfall.

*6. Page 2, line 1: here refer to Igel et al., 2013 before Malavelle et al. 7.*

Done.

*7. Page 2, line 20: "the algorithm of Field and Wood (2007)", please provide some details of what it is.*

This section has been expanded- thanks.

*8. Page 2, line 21 onward: when you introduce the CDNC product of MODIS, some details of what it is, its strengths and limitations should be included. The same is true of the other observations/products introduced in this section. There are many observations of the same parameter that are available, so it would be good to justify a bit more why these particular ones are used. For example, cloud fraction is from CERES, why not from MODIS (which the CERES product is in fact retrieved from if I am not mistaken)? How good is the MERRA-2 reanalysis for the sulfate mass concentration product?*

This would be good to do. As you point out the CERES product is partially from MODIS- what was meant is that we used the data included as part of the CERES product. This has been altered to specify that the cloud fraction is originally from MODIS and geostationary satellites and the SYN1DEG product has been referenced. We have tried to provide some additional background and more complete citations on the other data products.

*9. Page 2, last paragraph: how accurate is this rain water path estimate?*

We have redone this part of the paper to use the MAC-LWP data set, which comes with the total (precipitating+non-precipitating) liquid water path already calculated. In the original paper we were just inverting the algorithm used by RSS to calculate the rain rate because the quantity that the microwave actually measures is the total liquid water path, not the liquid water path and rain rate. The rain-cloud partitioning from RSS was based on SST alone, which added ambiguity to the results since it covaries with WVP.

*10. Page 3, section 2.2.2: more details on the model would be helpful. What does "NAME" stand for?*

This has been removed.

*11. Page 4, line 28: you write that the cyclone-centered mean is used to obtain the cyclone moisture flux. You should justify this a bit more, as the link to warm conveyor belt is not obvious: this is not the definition used typically. One argument is that cloud and precipitation occur predominantly in the warm conveyor belt and the warm frontal region, so the signal averaged in the entire cyclone region would be dominated by these two areas. Another is that cyclone cloud and precipitation depend strongly on the strength of the cyclone (here characterized by the surface wind) and the amount of moisture ingested in the cyclone (here characterized by the total*

*water path). Refer- ences are many, but as an example, one could be given to the Field and Wood paper (2007), and/or Bauer and Del Genio (JCLI 2006) and/or Rudeva and Gulev (MWR 2011).*

We referenced the Field and Wood 2007 paper in this paragraph, but we will add the other references in addition to the paper used to justify the simple model used in the original FW07 paper.

*12. Page 5, line 1-2: "indicating that this aerosol-cyclone indirect effect acts through*

*the warm rain process." This is quite a leap, how do we know this is not model-specific? Also, in Igel et al. (2013), even though the total ice mass in a warm front shows very small changes with an increase in aerosol concentration, the microphysical processes differed such that the aerosol had a compensating impact on vapor depostion and riming efficiency in the mixed phase region. So could it be the case here as well? in which case you might want to change this statement as the indirect effect here would not just act through the warm rain process. And this is not an aerosol-cyclone relation, but an aerosol-cloud relation.*

This was meant to refer to the model alone, and indeed this might be different in a different model. We have updated this to reflect this.

*13. Page 5, line 14-16. In figure S3, how do we know that the change in ToA SW flux is not caused by the direct aerosol effect instead of the effect on liquid water path?*

See discussion above.

*14. Page 5, line 22: "allowing for accurate observations", how accurate? There are issues in heavy rain situations with microwave radiometer retrievals of water path and wind speed, which could impact the estimate of the moisture flux and the classification used in the paper. This should be discussed, preferably as early as section 2.*

We have deleted 'accurate' as this is not a quantifiable statement. We have extended the discussion of biases in WVP and wind speed from microwave observations.

*15. Page 6, line 4: "highly consistent" is vague, could you be more quantitative? How is CDNC obtained when clouds are present? How often do you have retrievals in the southwest quadrant? Do you have a threshold on this number below which you do not consider the cyclone in question?*

We did not fully explain- CDNC is retrieved from cloud top effective radius and optical depth. It is only retrieved for overcast 1°x1° regions. We have added additional text to clarify how the retrieval is performed. Since the SW quadrant is highly cloudy it is not too difficult to perform the retrieval. There is no lower-bound on the number of retrievals required. We also perform the same analysis with MERRA2 SO4, which never has missing data since it is reanalysis and get the same results so this does not appear to be an issue. The inter-cyclone correlation between the

CDNC$_{SW}$ calculated using MERRA2 and MODIS is shown in the supplementary figures along with the effects of resampling MERRA2 SO4 so it is only sampled when MODIS can perform a retrieval. We have added text discussing differences in cyclone composited CDNC.

*16. Page 6, line 5: because the cold front is moving with respect to the center of the cyclones, sometimes it is in the southwest quadrant, other times in the southeast quadrant, and so the aerosols could be ingested in either quadrant. Have you tried to use the southeast quadrant instead to see if the results change?*

We have expanded the discussion related to this uncertainty to also include a recalculation of our results (characterized as the low-high CDNC plot as a function of WCB) using the cyclone mean CDNC and using the SE, and South part of the composite. The difference between high and low CDNC cyclones narrows when the cyclone-mean CDNC is used. This seems reasonable as it is adding a lot of noise to the calculation. However, if the SE part of the composite is used the separation between high and low CDNC cyclones narrows considerably. If just the south part of the composite is used it narrows much less. This seems like a case that would be improved by front identification, but for the purposes of this article we will stay with our simple cyclone compositing algorithm and note that there is some sensitivity to the sector of the cyclone composite used to calculate CDNC in the cyclone. We have also added these figures to the SM and additional discussion to the main text.

*17. Page 6, line 9: Figure S4: the two composites look rather different, the two color bars should match to make the comparison easier, and a 1-1 line should be added to the (c) scatter plot. Also, why not add these three plots to Figure 2 and make it a 4-panel figure?*

Thank you- we have moved all of these figures into a 1 panel figure and have added discussion regarding why they look somewhat different in structure. Overall the inter cyclone variability agrees well and once differences in sampling are accounted for MERRA2 and MODIS agree decently well in the region where there is abundant low, liquid cloud that MODIS can retrieve CDNC from.

*18. Page 6, line 16-17: why not discuss the very obvious differences in the southeast quadrant between observations and model?*

We have expanded the discussion and comparison between model simulations and observations. Thank you.

*19. Page 6, lines 18-26: so here the albedo is estimated with the CERES data, correct? and so is the cloud fraction? how can you have 100% cloud fraction in your cyclone area? you did not explain how this is obtained. Also, if you really have 100%CF, how do you have C4*

*MODIS CDNC? Finally the differences between MERRA-2 and MODIS are not that different in magnitude from the differences between high/low CDNC, how significant is this effect on albedo?*

We found that the SZA significantly impacted these results and we have redone this calculation. The MODIS CDNC retrieval is at cloud top, not in clear sky.

*20. Page 6, lines 27-33: This is not very convincing, as there is no mention of the moisture flux being constrained, which means that the albedo effect could come from cyclone with low vs. high CDNC having different mean moisture flux and thus different cloud cover caused by this instead.*

We have now constrained our estimate by WCB. See above.

*21. Page 7, the regression model work: I am not sure I se the link between the albedo discussion and this work. Why not introduce this before the albedo work? This re- gression model is obtained how, based on Figure 5? Finally, I am not sure what the implication of these results is? In particular the very last sentence is unclear, please elaborate. Line 15, and line 17, large and small cyclones do not really mean anything, you do not know anything about their spatial extent. You could use strong/weak maybe, but you would need to specify that this is in term of moisture flux strength, not winds alone.*

We train the regression model using the observational record. Figure 5 just shows a summary of this showing the average of all the observations in predictor space to show that the lines curve a little more sharply at low CDNCSW. We have specified that by large and small we mean large and small moisture flux. We have also remade the plot and tried to make it clearer that it is just a plot summarizing the regression models fit to the data.

*22. Page 8, line 10: "both simulations", not clear what the two simulations are, only one is mentioned above.*

Removed- see above.

*23. Page 8, lines 22-26: This feels out of place, why mention the convection permitting simulations in this context? It just repeats what has been said a few times already about the merits of the high resolution vs GPM-resolution simulations.*

This has been removed.

*24. Page 9, line 15: "an aerosol indirect effect on midlatitude storms" is not accurate, maybe add "clouds" after "storm"*

That is a good point. Done.

**Reviewer 3:**

*The manuscript by McCoy et al. investigates aerosol-cloud interactions in midlatitude cyclones over the North Atlantic using modelling and the Hohluraun eruption. I think as such the topic is interesting, but the uncertainty has to be discussed much better. My recommendation is to name the motivation and discuss major limitations of the different approaches such that the scientific evaluation of the work is easier. I hope my comments will be useful for improving the*

*manuscript.*

Thanks- we have tried to more fully explore the uncertainty in our analysis and appreciate the reviewers help in improving our paper.

*General comments*

*I recommend to provide more information/discussion on uncertainty and the motivation of some specific choices in the methods for this work. I understand that one would want to highlight the positive results that seem to provide a conclusive story, but I recommend to more openly discuss the uncertainty in such work.*

*In my opinion, meteorological variability has a large impact on the perceived aerosol-cloud interaction, no matter whether we look at observations or modelling.*

We agree that meteorology controls the majority of cyclone behavior, but it does seem like once we remove variability associated with meteorological variability there is still a signal associated with aerosols. This result appears to be present in both highly idealized simulations and observations. Overall we estimate an impact of aerosols from a standardized perturbation in CDNC that is less than 30% of the response to meteorology- so meteorology still dominates cyclone behavior.

*My suggestion is to clearly high- light it for supporting an open debate and helping the reader in assessing the results. When we look for instance at the Holuhraun case, we have very few cyclones that have been affected by excessive amounts of sulphate, i.e., 10 cyclones in total according to Fig. 8. Half of these cyclones show an increase in CLWP, but that is within the range of CLWP anomalies that also naturally occur in the absence of SO4 perturbations. The other half of the cyclones with above-threshold perturbations in SO4 show, however, almost no change in CLWP and this includes the cyclone with the largest SO4 pertur- bation. I would state this explicitly in the text.*

See comments above, we have removed this.

*In addition to meteorological variability, I wonder how the regionally limited increase in aerosol affects the radiation transfer, thus the temperature gradients and possibly the cyclone/WCB statistics, based on which you construct your argument that aerosol- cloud interaction is the driver of CLWP increases. Have you analysed the changes in the temperature distributions? This would be important for understanding the physical mechanisms behind the model results.*

Removed. See above.

*The abstract could be a little longer, e.g., it does not state which model and satellite data has*

*been used, and should more clearly state the uncertainty assessments, e.g., uncertainty in assumptions about the eruption.*

Very true- we have expanded the abstract- thanks.

*p.1, l. 20: "liquid water amount and thus the albedo" The cloud albedo depends on the number and size of droplets. I also wonder whether "constraining predictions of the 21st century warming" is a good word choice as the warming will depend not only on the physics, but also on the socio-economic development. As such we will always have a spread in long-term projections into the future. In any case, citing of references would be useful here.*

Optical depth is a function of both liquid content and CDNC, not counting whether the lifetime effect changes cloud fraction. All else being equal we should expect that increasing liquid water amount should increase albedo. Since the range of possible climate sensitivity is strongly affected by the assumed strength of aerosol indirect effects if we had a tighter constraint on indirect effects we should have a tighter constraint on climate sensitivity. This statement implies that it is for a given emissions scenario, which should give a more constrained prediction based on a better constrained climate sensitivity. We have tried to expand this section to better explain what we meant.

*p.1, l. 22: "thermal contrasts" alone are not enough to form a cyclone. It might be best to just delete that sentence.*

Removed.

*p.3, l.7: I am a little bit surprised that both the configurations with and without explicit convection use the same vertical resolution. Maybe you can explain why you have made that choice. Would you expect the results to differ when you also change the vertical resolution?*

We are using the model configuration based on the operational model used for CMIP and trying to make as few changes as possible.

*p.3, l. 14-15: Do you mean that the exponential decay starts at the surface or above 5km? Both seems to be tricky, unless there is observational evidence for it, since aerosol is typically well mixed in the boundary layer, but only few places have a deep BL of 5 km. Maybe use cm-3 instead of /cc to be consistent with your results section.*

The exponential decay starts at 5km. Given that it is constant and non-interacting it is not intended to be highly physical. We also tried a constant vertical profile, but this generated a great deal of ice due to the highly simplistic ice nucleation used in the simulations at high altitudes and the results were clearly unrealistic.

*p.3, l.18: Please clarify "non-interacting". I guess you mean that no complex aerosol parameterisation is coupled to the atmospheric model, but you prescribe the aerosol*

*concentration as function of vertical velocity and let the aerosol interact with the radia- tion and clouds in the model (such a setup could also be interpreted as "interacting").*

Thank you- yes- what we meant was that precipitation does not deplete the aerosol and no new aerosol is generated. 'Fixed' would have been better. We have changed this to clarify what was meant.

*p.3, l.19: Is there a reason why you have chosen to increase the aerosol just in this channel? Such a setup generates a steep (artificial) gradient in aerosol that might change your temperature gradients and thereby the cyclones.*

This setup was chosen because it was simple to implement and didn't require any assumptions regarding the aerosol gradient. The SST is fixed in the simulations so the aerosol shouldn't change the overall temperature profile beyond changing the atmospheric temperature. We do not think that the step function of CCN should affect our results since their purpose is to give us insight into how to analyze the observations- however, this is something that we plan to look at more in the future with more simulations so we will be looking at step function and gradiated channels for the purpose of understanding model behavior more clearly.

*Section: 2.2.2: I think if you could add the uncertainty range of these estimates and maybe even systematically test the effect of such a range on your results, the work could be a much better contribution. Later in the results section you touch on that type of uncertainty. Maybe you could motivate it here already.*

Removed- see above. We will consider this in a future paper more fully examining these results.

*p.4, The first paragraph is partly redundant with the method section. Maybe you can merge the text.*

We have substantially re written both the methods section and this section and hopefully it flows better now.

*p.5, l.1-2: Is this due to the simple parameterisation that you have implemented into the model? In either case I would mention it here again, because the way it is currently written suggests that what your model tells us is a fact and that fact "warm rain process" seems to contradict what one would expect for precipitation formation in midlatitude cyclones (in reality).*

That was poorly worded, we have altered the text to make it clearer we just meant the model.

*p.6, l.8-9, Fig. 2: It is not clear why you get large CDNC in the cyclone center, typically a decline going outward, but than again an increase in CDNC to the southwest. Could you argue that this is something you would expect? Here I would also want to read more about the comparison of the CDNC of MERRA and MODIS to judge the quality of the re-analysis.*

We have expanded this discussion, but the CDNC retrieval outside of the regions of low lying cloud (such as the cold front) are not reliable and the increase toward the center of the cyclone is not likely to be robust.

*p.7, l.26: "enhancement of cyclone properties" I would speak of changes of cyclone properties.*

Altered- thanks.

*p.8, l.5-6: MERRA assimilates, however, observations that have experienced a poten- tial effect of the aerosol on the meteorology. So, this might not be as conclusive as one would hope.*

Very good point- we were concerned about this and have undertaken a separate and more extensive analysis of the MERRA2-MODIS CDNC products (McCoy et al., 2017). It does seem like there is a reasonable agreement between these products and long term changes due to pollution and volcanic degassing that cannot be explained by meteorology. We also added evaluation of the CLWP-WCB relationship partitioned into high and low CDNCSW populations (inferred from MERRA2) to our analysis and see that MERRA2 CLWP actually has the opposite behavior to the observations suggesting that the MERRA2 analysis is not baking in the increase in CLWP with increasing CDNC.

*p.9, l.18-20: These are big implications, but how could we know that we would get the same answer when we used other models or other volcanic eruptions, given the uncertainties and variability?*

You are correct that assigning a systematic bias to GCMs was not fair. Some GCMs have extremely strong lifetime effects that are sure to exceed the observations shown in our study. This sentence has been removed.

 **References:**

Bender, F. A. M., Engström, A., Wood, R., & Charlson, R. J. (2017). Evaluation of Hemispheric Asymmetries in Marine Cloud Radiative Properties. *Journal of Climate, 30*(11), 4131-4147. doi:10.1175/JCLI-D-16-0263.1

Carslaw, K. S., Lee, L. A., Reddington, C. L., Pringle, K. J., Rap, A., Forster, P. M., . . . Pierce, J. R. (2013). Large contribution of natural aerosols to uncertainty in indirect forcing. *Nature, 503*(7474), 67-71. doi:10.1038/nature12674

Derber, J. C., & Wu, W.-S. (1998). The Use of TOVS Cloud-Cleared Radiances in the NCEP SSI Analysis System. *Monthly Weather Review, 126*(8), 2287-2299. doi:10.1175/1520-0493(1998)126<2287:tuotcc>2.0.co;2

Elsaesser, G. S., O'Dell, C. W., Lebsock, M. D., Bennartz, R., Greenwald, T. J., & Wentz, F. J. (2017). The Multi-Sensor Advanced Climatology of Liquid Water Path (MAC-LWP). *Journal of Climate, 0*(0), null. doi:10.1175/jcli-d-16-0902.1

Field, P. R., Bodas-Salcedo, A., & Brooks, M. E. (2011). Using model analysis and satellite data to assess cloud and precipitation in midlatitude cyclones. *Quarterly Journal of the Royal Meteorological Society, 137*(659), 1501-1515. doi:10.1002/qj.858

Field, P. R., Brožková, R., Chen, M., Dudhia, J., Lac, C., Hara, T., . . . McTaggart-Cowan, R. (2017). Exploring the convective grey zone with regional simulations of a cold air outbreak. *Quarterly Journal of the Royal Meteorological Society, 143*(707), 2537-2555. doi:10.1002/qj.3105

Field, P. R., Gettelman, A., Neale, R. B., Wood, R., Rasch, P. J., & Morrison, H. (2008). Midlatitude Cyclone Compositing to Constrain Climate Model Behavior Using Satellite Observations. *Journal of Climate, 21*(22), 5887-5903. doi:doi:10.1175/2008JCLI2235.1

Lu, Y., & Deng, Y. (2015). Initial Transient Response of an Intensifying Baroclinic Wave to Increases in Cloud Droplet Number Concentration. *Journal of Climate, 28*(24), 9669-9677. doi:10.1175/jcli-d-15-0251.1

McCarty, W., Coy, L., R, G., A, H., Merkova, D., EB, S., . . . K, W. (2016). MERRA-2 Input Observations: Summary and Assessment. *Technical Report Series on Global Modeling and Data Assimilation, 46*.

McCoy, D. T., Bender, F. A. M., Grosvenor, D. P., Mohrmann, J. K., Hartmann, D. L., Wood, R., & Field, P. R. (2017). Predicting decadal trends in cloud droplet number concentration using reanalysis and satellite data. *Atmos. Chem. Phys. Discuss., 2017*, 1-21. doi:10.5194/acp-2017-811

Miltenberger, A. K., Field, P. R., Hill, A. A., Rosenberg, P., Shipway, B. J., Wilkinson, J. M., . . . Blyth, A. M. (2017). Aerosol-cloud interactions in mixed-phase convective clouds. Part 1: Aerosol perturbations. *Atmos. Chem. Phys. Discuss., 2017*, 1-45. doi:10.5194/acp-2017-788

Randles, C., AM, d. S., V, B., A, D., PR, C., V, A., . . . R, G. (2016). The MERRA-2 Aerosol Assimilation. *Technical Report Series on Global Modeling and Data Assimilation, 45*.

Schmidt, A., Leadbetter, S., Theys, N., Carboni, E., Witham, C. S., Stevenson, J. A., . . . Shepherd, J. (2015). Satellite detection, long-range transport, and air quality impacts of volcanic sulfur dioxide from the 2014–2015 flood lava eruption at Bárðarbunga (Iceland). *Journal of Geophysical Research: Atmospheres*, n/a-n/a. doi:10.1002/2015JD023638

---

## Referee Report (RR1)

Review: "Aerosol-mid-latitude cyclone indirect effects in observations and high-resolution simulations"

This revised version of the manuscript explores the impact of aerosols on liquid water path in extratropical cyclones. This new version is more detailed and more convincing that the previous paper, but there are still issues, mostly to do with the presentation of the results and the structure of the paper. The comments below list the various parts were rewriting or modifications are needed to make the paper clearer and more fluid.
Based on the sensitivity of the results to the source of cloud droplet number concentration, the conclusions should be somewhat expanded and in some places toned down.
Otherwise, the manuscript is acceptable for publication after minor but highly recommended revisions.

Detailed comments:

1. The document needs some cleaning and tightening as the current presentation still includes remnants of the previous version. Section 2.3.1 is the only subsection in section 2.3. Similarly 3.1 is the only subsection of this level in the results section 3.
2. In the conclusions, please discuss where the additional variability in cloud liquid water path might come from, as you indicate that 60% (and not 100%) or this variability is caused by the moisture flux and aerosol impact on cloud droplet number concentration.
3. You might want to relate your results to a recent paper by Naud et al. (2017), who explored the co-variations between aerosol optical depth and cloud cover in extratropical cyclones. In particular they examine cold and warm fronts separately, which might help explain better your assertion that the cold frontal region is where the albedo change with aerosols is concurrent with the change in liquid water path and cloud fraction. Also, while cited, the Grandey et al (2013) study who explore the impact of extratropical cyclone strength on the cloud-aerosol relationship could be further discussed in light of the results of this paper in the conclusions. Both of these studies to some extent contradict the statement in the introduction that "aerosol-cloud indirect effects have not been observed in extratropical cyclones" (L25, p2). Although neither study can establish a causal relationship, the fact remains that observations in extratropical cyclones have already been used to explore aerosol-cloud relationships.
4. In the introduction please indicate where these cyclones are. In 2.1 it is specified that they are over the oceans but a latitude range should be specified. I suppose that these are all in the northern hemisphere? Are they found in both Atlantic and Pacific oceans? Are the Mediterranean cyclones also included? At least it is specified for the simulations but at the bottom of page 12, it sounds as if the observed cyclones are sampled in both hemispheres. If indeed this is the case, then "southwest quadrant" is misleading.
5. Section 2.2.3: please add a couple of sentences to explain how the MERRA-2 sulfate mass is used to obtain CDNC.
6. Section 3.1.1: the section on the aquaplanet simulations is very short, and about half of this section is in fact about the relation between the moisture flux and precipitation, which is more general than just about model simulations. The main result here seems to

be that regardless of resolution and aerosol concentration the relationship between moisture flux and precipitation rate is unchanged. No mention of a relationship between CDNC and liquid water path is made. Both figures associated with this section are in the supplemental material. It would be preferable to have these figures in the paper since they do illustrate the discussion. This would be especially useful because in fact this discussion is confusing: on the one hand, it gives the impression that the rain rate vs moisture flux relationship does not change with the aerosol concentration (or resolution). But Fig. S3 suggest that it does. To bring this matter to rest though, the figure should be clarified by either drawing a linear regression per model configuration or constraining the data points so they would have the same WCB value per configuration. Another confusing matter is this: if WCB is kept fixed and LWP changes with aerosols, then surely precipitation should as well, no?

7. Section 3.1.2: again the title of this section is misleading, it says "observed' cyclone properties and yet the very first sentence is about comparing observations with simulations. It seems that pages 9-11 are in fact part of a single section on the simulations while a new section should be when the work using the observations alone starts (second paragraph p12)

8. Are the three WCB regimes of Figure 2 defined based on the observations? Two questions arise: are the three population very different in the number of members and would sampling issues affect the results? Are the distribution of WCB per region for observations and the different model configurations very different? Why not use the same color scale for all composites in Figure 2?

9. The discussion on why the southwest quadrant is a good place to sample for CDNC could be improved. First this quadrant is dominated by low-level clouds and so MODIS derived CDNC is probably better sampled there. It is not clear however that it would be representative of the entire cyclone, and in particular the warm sector and warm frontal zone that are dominated by high-level clouds and thus CDNC information is missing. Second the warm conveyor belt which is ingesting moisture into the cyclone tends to originate from the south east and is not always found in the southwest quadrant. So it is not clear that the "southwest quadrant is likely to be the source of moisture and aerosol for the cyclone". Figure S6 is quite important for this discussion and yet once again it is in the supplemental material. The whole discussion on how to best partition the cyclone population based on CDNC needs to be improved, it is quite confusing still. For example, it is unclear what Figure 4 is really telling us.

10. How significant is the separation in Figures 3a and 3b? based on this figure and the tests presented in Figs S7, S8, and S9, it seems that the separation is best for the southwest quadrant possibly because this is where low-level clouds dominate and ice contamination in the satellite observations is less. It seems that the results really target this specific quadrant and that little is known of the clouds that are found in other parts of the cyclones.

11. Discussion of Figure 7: here it might be worth comparing with the results of Naud et al 2017 (fig 10). Also, why not define the three WCB regimes based on terciles of the entire cyclone population? This would alleviate the small number of member issue for the 5 mm/day category?

12. Page 16, discussion on albedo effects: it is quite worrying that the difference between the MODIS and MERRA2 constrained albedo variations with aerosol are this important for low values of WCB which constitute the largest number of cyclones. This should probably be said somewhere. Unrelated: the whole discussion on albedo could probably be presented in its own subsection.
13. I do not see how Figure 11 is showing a "stronger increase in CLWP for a given increase in CDNC_SW in more pristine storms". Either there is an error in the sentence or the caption of Figure 11 needs rewriting.
14. Conclusions: last sentence. Given the observations at your disposal and the disagreements between MODIS and MERRA-2 constrained relations, I would tone down this last sentence, as I am not convinced that this is a demonstration, but rather the observation of co-variations in accordance with the expectation of the sort of effect aerosols should have (as demonstrated with the simulations though).

Typos:
Line 21, p 3: replaced "is" with "area" between "composite" and "located"
Line 17, p9: write what "CMIP" stands for.
Line 16, p14: "extent" is misleading as this is a term often used for vertical extent. "fraction" or "cover" might be more appropriate.
Line 5, p15: add "s" to "support"
Line 21-24: this sentence is too long and is missing a verb towards the end.
Line 13, p16: replace "can be" by "can display" for example

Reference:

Naud C. M., D. J. Posselt, and S. C. van den Heever, 2017: Observed covariations of aerosol optical depth and cloud cover in extratropical cyclones. J. Geophys. Res – Atmos., 122, 10,338-10,356. Doi:10.1002/2017JD027240.

---

## Author Response (AR2)

**Response to reviewer comments Aerosol-mid-latitude cyclone indirect effects in observations and high-resolution simulations**

**We thank the reviewers and editor for their consideration of our manuscript. Responses to specific comments are shown below. Responses are in bold font.**

Editor comments:
A sample of important issues raised by the reviewers that must be addressed. (I'm not implying that others don't need to be addressed.):

1) The paper should lead with the observations. The models have indirect effects 'baked in' even at the 'convection permitting' resolution so the hypothesis posed at the top of pg 12 rings a little hollow.
Please more clearly delineate observations and modeling, as requested by the reviewers.
**The paper is now organized so that the modelling and observational results are presented in parallel. We have tried to more clearly delineate observations and models. We have also tried to make it clearer that, while the first indirect effect is baked into the simulations, adjustments are not. For instance, examination of the response of clouds to aerosol as modeled by CASIM in stratocumulus(Grosvenor et al., 2017) and mixed-phase convection(Miltenberger et al., 2017) display a diverse array of adjustments to aerosol. We have added text in several parts of the paper discussing how the observational analysis cannot prove causation and how the simple modelling experiments should only be used to hypothesize a causal link.**

2) The regression analysis presents problems. First, one cannot draw conclusions about causality because of co-variability. Granted, the wording around the regression analysis doesn't suggest causality but the closing statements in the abstract and conclusions do. Second, it is very difficult to judge whether the regression fit of the proposed power law is good enough for quantitative conclusions to be drawn (only ~ 65% of variance in CLWP is captured). Fig. 11 is not helpful. Please carefully reconsider how you present the regression analysis.
**You are right, this was not clear the way it was written in the conclusions. We have tried to make it clearer that our closing statements are based on the combined evidence of our idealized model and our analysis of the observed variability in the Earth system. The first paragraph of the conclusions now reads:**

*"Analysis of observed covariability between meteorology (as characterized by warm conveyor belt (WCB) moisture flux), warm cloud microphysics (as characterized by cloud droplet number concentration (CDNC)), and cyclone cloud properties is consistent with increasing CDNC leading to an increase in cyclone cloud liquid water path, fractional coverage, and ultimately albedo. While suggestive, empirical analysis of the observational record cannot prove causality. We support this analysis by performing a set of simulations where CDNC is set at high and low values. The response of CLWP to changes in CDNC in these simulations elucidates the mechanism by which this covariability may be explained and provides support for causality flowing from enhanced CDNC to enhanced CLWP."*

**We have also tried to expand on what 65% of the variance means. We believe that Fig. 11 of the previous revision is useful in that it illustrates the fit in two dimensions and clarifies that there is some dependence on both predictors, but we agree that by itself it is only useful to orient the reader. We have provided 95% confidence intervals on the coefficients of the fit in Table 2. All the coefficients are significant at 95% confidence.**

3) The elimination of the Holhuraun volcano study, which is inconvenient for the main (relatively strong) conclusions, presents a problem, as noted by Reviewer 1. In the current version that study is only peripherally referred to. The fact that the re-working of that analysis suggests weaker evidence for aerosol-cloud interactions might require separate publication, but now that you are in possession of that knowledge it cannot simply be brushed aside. You would do better to discuss it.

**Thank you for your understanding in this regard and allowing us to clarify why we removed this case study. We responded to reviewer 1 showing our analysis of this case study. There does appear to be an effect, but we are concerned that we need to do more analysis using NAME (or another dispersion model) to better understand the uncertainty as to where the plume from Holuhraun travelled.**

**We agree with reviewer 1 that the paper is more complete with the inclusion of this case study. We have added a section on the eruption and we note that our results do not necessarily contradict the lack of response seen by (Malavelle et al., 2017). Examination of the difference in means between the climatological CLWP and CLWP during the eruption was significant in several cases, but shows sensitivity to cyclones very far from the eruption. We suspect this is related to uncertainty in the propagation and rain-out of the plume at these distances as simulated by the dispersion model. We want to run more simulations exploring the uncertainty regarding the eruption. This would necessitate that simulations with different emissions fluxes, heights, and precipitation removal efficiencies be explored. This is not feasible in the context of the current paper, which is why we removed this case from the previous version with the intent of following up on this case in the future when we could understand the uncertainty space surrounding the plume modelling. We have described these uncertainties in the text and stated the steps we will take in the next paper to clarify these uncertainties. We have also added analysis showing that CLWP was unusually high in cyclones predicted to have interacted with the volcanic plume. We hope that this gives the reader sufficient information to interpret our global analysis. All this material is in sections 2.3.2 and 3.7.**

4) Regarding the modeling, Please provide more clarity on key components of the microphysics. The dependence of Nd on aerosol and updraft is discussed at length but what autoconversion scheme is used? This is probably a more important issue than activation. Some autoconversion schemes have particularly strong dependence on drop concentration while others do not and this has significant impact on aerosol-cloud interactions.

**Shipway and Hill (2012) describes CASIM, which utilizes Khairoutdinov and Kogan (2000) to describe autoconversion. This was briefly referenced in the methods section so we have added text to clarify this:**

*"The CASIM microphysics scheme is described in Shipway and Hill (2012). The warm rain processes in CASIM is compared to other microphysics schemes in Hill et al. (2015). The cloud physics parameterization used in CASIM is described in Khairoutdinov and Kogan (2000)."*

**We have also added text to discuss how our results might be different with a different microphysics scheme. For warm rain there is modelling evidence that CASIM shows good agreement with other microphysics schemes (bin and bulk)(Hill et al., 2015) and so qualitatively, we would expect similar results, if all else is equal. Obviously, the simulations presented here include ice processes, which may change the d(rain rate)/dCDNC < 0 relationship (Koren et al., 2005) and as such may be scheme dependent. The influence of ice on aerosol cloud interactions is very important and very uncertain, as are the ice phase parametrizations(Tan et al., 2016). This is an open area of research and we have added some text about this uncertainty.**

**We also accept that dynamic feedbacks associated with microphysics are very important(Xue and Feingold, 2006;Xue et al., 2008;Hill et al., 2009) and a different microphysics scheme may lead to a different dynamical evolution. However, given the forcing in a cyclone region and the resolution in even the high-resolution simulations, we believe the using another multi-moment bulk scheme would lead to qualitatively similar answers as those presented here, assuming the ice phase processes are similar. We have added text discussing this as well.**

**This is discussed in the end of section 2.3.1:**

*"Finally, in the simulations presented in this paper we explore the response of the clouds in the UM treated by the CASIM cloud microphysics to changes in CDNC. A different cloud microphysics scheme would potentially yield a different adjustment to aerosol, but our results are unlikely to be qualitatively dependent on the simplistic activation scheme chosen here. We also acknowledge that the adjustment of cloud to aerosol in these idealized simulations will be a function of the CASIM microphysics scheme. Examination of CASIM in relation to other multi-moment schemes suggests that if the adjustment works through the warm rain process another multi-moment scheme would produce a qualitatively similar result (Hill et al., 2015). It is important to note that the simulations presented in this work include ice processes, which may affect the susceptibility of rain rate to changes in CDNC (Koren et al., 2005;Rosenfeld and Woodley, 2000). These effects may be highly dependent on the choice of microphysics scheme. Further, the representation of these effects in models is very uncertain and could substantially affect the predictions of our simulations. Another important mechanism not considered in our simulations is the dynamical feedback on aerosol-cloud interactions(Hill et al., 2009;Xue and Feingold, 2006;Xue et al., 2008). The dynamical evolution in response to changes in CDNC could change dramatically using a different microphysics scheme. We believe that our simulations are not strongly sensitive to the representation of these effects in CASIM given the resolution of the simulations and the forcing within a cyclonic system.*

*Overall, we present these simulations as an exploration of how clouds within cyclones respond to changes in CDNC through the warm rain process. These simulations are used to contextualize the observations and evaluate whether we may reproduce observational variability utilizing this idealized set of simulations."*

5) Many of the supplementary figures indeed belong in the main text. Please reconsider the distribution.

**We have moved the figures as suggested by reviewers. Several of the figures are just sensitivity tests replicating an existing figure and we have left those in the supplementary material.**

Some specific comments of my own:

1) In the opening lines of the abstract there is a conflation of 21st century climate change and aerosol effects on midlatitude cyclones. Climate change does not equate to increases in N_d (drop concentration).

**That was sloppy- we meant to say: 'infer climate sensitivity, which allows us to predict 21$^{st}$ century climate change(Andreae et al., 2005).'- we have amended this to read:**
*"Establishing how much aerosol emitted during the 20$^{th}$ century has enhanced the liquid water amount and thus the albedo of midlatitude storm systems is a key step in constraining the climate sensitivity inferred from the observational record."*

2) Pg 5, lines 1 and 14: N_d is not retrieved at cloud top. It is based on cloud top effective radius and (path integrated) cloud optical depth
**That is right, this has been amended. Thanks.**

3) Pg 6: perhaps cloud property retrievals are not subject to thresholding but you still have to define cloud fraction so that all sky albedo can be calculated. How is cloud fraction defined? Is it consistent amongst analysis and model output?
**The all-sky albedo is calculated by the SYN1DEG algorithm using the algorithms described in Doelling et al. (2013);(Doelling et al., 2016). This does use information about the cloud mask to create angular distribution models to convert radiances to fluxes(Loeb et al., 2003) and to account for diurnal variations between different satellite overpasses and geostationary data. It does not use the cloud fraction explicitly in the calculation of all-sky albedo beyond creation of the angular distribution model and using the geostationary observations to fill in the diurnal cycle. We have added text clarifying this.**

**We do utilize cloud mask from the CERES SYN1DEG data set, which is retrieved following Minnis et al. (2011). This data product is defined differently than the model, which is max-random overlap. Because of this we do not offer a comparison between model cloud fraction and CERES SYN1DEG cloud fraction.**

**Thank you for this comment. We have expanded the methods section to clarify these points:**
*"The 3-hourly data is averaged to create a daily-mean albedo and CF. CF, clear-sky albedo and all-sky albedo are provided in the CERES SYN1DEG data set edition 4(Wielicki et al., 1996;Doelling et al., 2013;Doelling et al., 2016). CF is calculated from MODIS and*

*geostationary satellites based on the  Minnis et al. (2011) cloud mask. It is used in the calculation of the albedo retrieved by CERES as described in Doelling et al. (2016) to create an angular distribution model and to interpret geostationary observations of albedo in relation to the observations from CERES. It should be noted that without utilizing a satellite simulator(Bodas-Salcedo et al., 2011) we cannot directly compare cloud fraction to the aquaplanet simulations presented in this work."*

4) Pg 7, "microphysics and aerosol explicitly interact" is a bit strong. The statement is true but a bit misleading given the resolution.
**The sentence has been amended to explain that the interaction is explicit at the resolution of the model. Our intention was to convey that this was an improvement on parameterized convection:**
*"Without convection being parameterized microphysics and aerosol explicitly interact at the model resolution"*

5) Pg 9, line 6: "disentangling". It seems like the meteorology is fixed and the aerosol is changed but this isn't what happens in nature so are you really disentangling anything of value? (see e.g., Feingold et al., PNAS 2016).
**Good point- disentangle has been removed.**

6) Pg. 10, line 15, what does "lifetime effect" mean in the context of a midlatitude cyclone. This terminology is vague at best. (See editorial comment below.)
**Lifetime has been changed to adjustment throughout the paper.**

7) Pg. 10, Fig. 2: how many simulations were performed. The observations average many initial conditions and aerosol-meteorology co-variability. Do the models all use the same initial conditions?
**This was not clearly stated in the paper. We have added the clarification:** *"Simulations were then run for 15 days. A single simulation was run at each resolution and aerosol concentration, giving a total of four simulations of 15 days each."*

8) Since aerosol concentration is fixed in the modeling, aren't you biasing your aerosol effects to be much stronger than would be the case if aerosol were washed out by the rain? If CASIM is indeed two-moment, it should remove aerosol and allow N_d to fluctuate accordingly.
**To allow CDNC to fluctuate we would need to make some decisions about aerosol sources, which would increase the complexity of our simulations.  In the context of our simple model we set CDNC and rain is inhibited by this. In the absence of dynamical feedbacks or ice processes we would expect a further enhancement in LWP and less precipitation cleaning out aerosol. This would mean that in the canonical representation of the Albrecht effect we would be somewhat under-representing our response. However, in our conceptual model we propose that rain rate is controlled by the large-scale environment and so we should not expect this fixed aerosol to affect the strength of the response as the LWP increases to push the rain rate back to be in agreement with the large-scale environment. This is good to discuss and we have added text to the aquaplanet section to clarify this assumption:**
*"It is also important to note that the fixing of CDNC at a constant value means that precipitation*

*does not affect CDNC via the removal of aerosol and thus CCN. The simulations presented here are intended to examine the adjustment in cyclone clouds to changes in CDNC as opposed to a change in aerosol fluxes. If aerosol were allowed to respond to precipitation we may speculate as to how this might affect the behavior of the cloud adjustment simulated by CASIM. As described in the following sections, the rain rate on a daily, cyclone-wide scale is determined by the large-scale environment. Subsequently, we may hypothesize that the feedback between aerosol, CDNC, and the rain rate is relatively weak, but we note that this assumption of fixed CDNC artificially removes this interaction pathway with the intent of understanding the adjustment in cloud properties to CDNC."*

9) Pg. 14, lines 8-15: Wouldn't the open cells be raining and have low N and lower domain-averaged liquid water path?

**We argue that their relatively low liquid water path and low CDNC should result in a higher susceptibility than in the frontal zones where aerosol susceptibility should be negligible. They are also raining, which is a necessary condition of having a precipitation-mediated cloud adjustment. This was not clarified in the text and we have added discussion to amend this as well as referencing the literature discussing the importance of precipitation to the open-closed cell transition:**

*"Numerous studies have linked the dominance of open or closed mesoscale cellular convection to precipitation, and aerosol modulation of precipitation(Stevens et al., 2005;Feingold et al., 2015;Koren and Feingold, 2011;Rosenfeld et al., 2006;Mechem et al., 2012;Goren and Rosenfeld, 2012;Wang and Feingold, 2009a, b). Because of the tenuous nature of this cloud regime, and because they are typically precipitating it is reasonable to suspect that they will be more susceptible to aerosol-driven changes in their macrophysics than either thick frontal clouds or non-precipitating clouds. It is not the intention of our investigation to examine the complex dynamics of mesoscale cellular convection, but we have chosen our observational data sets so that they do not exclude this cloud regime, and the localization of differences in CLWP between high and low CDNC$_{SW}$ cyclone populations is suggestive given the existing literature regarding both the radiative importance of these clouds(McCoy et al., 2017c) and their relation to precipitation and aerosol(Koren and Feingold, 2011). Overall, this behavior motivates future work examining this region in higher resolution and higher complexity models that can resolve these features."*

**Thank you for suggesting this.**

Some editorial comments:

1) What do you mean by cloud-aerosol lifetime effect? This concept is particularly poorly defined. It doesn't reflect on the lifetime of the midlatitude cyclone. I assume you mean a reduction in precipitation efficiency. Regardless, please be precise.

**Lifetime has been changed to adjustments to follow the more process-agnostic description used in the IPCC report. Thanks.**

2) Acronyms like CDNC, while widely used, are bad practice since they can much more easily be represented by symbols (N, perhaps with subscript d) as you already do for aerosol concentration. The excessive use of acronyms will make papers in our field unreadable if we are

not more careful. CLWP could be replace by L_c, SZA by \theta, CF by f_c, etc.
While I don't insist on these changes, please consider that they really help with clarity, particularly for those entering the field. (See pg. 15, what is CDNCSW??)

**That was a typo on page 15- it was supposed to be CDNC$_{SW}$. It is true that reading papers from outside the field can often be difficult. However, use of symbols can also be confusing as they have less information for people to infer their meaning. The vague use of $\lambda$ in the feedback and climate sensitivity community is one good example of this. In the interest of clarity we have added a table (table 1) of acronyms that can be consulted by the reader.**

3) CCN, again is widely used, but has no quantitative meaning unless defined at a specific supersaturation. You might as well use the same symbol as for accumulation mode aerosol number concentration if indeed they are used interchangeably.

**That is a good point, we have changed CCN to N$_{acc}$ when referring to the simulations since that is all that is being perturbed.**

4) The manuscript has some lapses in grammar (e.g., Pg 3 lines 20-22)and could benefit from some liberal use of commas in some places.

**This has been amended.**

5) Many figures and their fonts are much too small to be readable in print form.

**Figure sizes and font sizes have been increased.**

**Report #1**

The weak evidence of the Honuluraun case from the first submission has been removed completely. The rebuttal states that the revised analysis gives even poorer evidence for an aerosol effect and the case will be revisited in another article. Such poor evidence is contradicting the claim of the article "aerosol-cloud indirect effects have substantially altered clouds in extratropical cyclones" (see abstract and conclusions). I consider this bad practise that we should not pursue in science.

**We understand that the reviewer is concerned that by removing the Holuhraun case from the manuscript we are suppressing contradictory evidence. Overall, we find that the Holuhraun case study does not contradict the results presented in the body of the paper. If it did, we would agree with you that this is bad practice to remove it. Although we replicate the following discussion in the paper, we also want to present our analysis for the reviewer.**

**We examined September and October cyclones in the North Atlantic (50°W-30°E) during the eruption and in non-eruption years.  The NAME dispersion model was used to flag cyclones as being significantly affected by SO4 from Holuhraun.**

**We want to see if Sept-Oct 2014 had unusually high CLWP given the synoptic environment.  If we examine the climatological dependence of September-October CLWP on WCB (excluding 2014) for 50°W-30°E and within 20° latitude of Iceland we get the fit shown using the dashed line. A 20° latitude range is consistent with a 48-hour transit time from the fissure(McCoy and Hartmann, 2015). The cyclones during the eruption, and flagged during the eruption to have interacted with the plume are shown using open**

circles and half-circles, respectively. An average volcanic SO4 predicted by NAME of 0.1μg/m³ in the southwest quadrant of the cyclones was used to flag whether the cyclones had interacted with the plume. The anomalies in CLWP relative to the climatology were calculated using the dashed-line fit. A t-test and rank-sums test were used to evaluate whether the mean anomaly during 2014 was unusual. We found that cyclones flagged by NAME with emissions at 1500-3000m were significantly different than the climatology at 95% confidence (see legend).

[Figure]

However, we also found that this was dependent on the region being considered. If we consider all cyclones whose centers were within 30° latitude (35°-90°N) find that the result is less sensitive to the emissions height.

[Figure]

**But if we expand to latitudes from 30-90°N there are five cyclones near 30° that have a very high WCB, but a low CLWP. The negative anomaly from these cyclones results in the mean anomaly not being significant at 95% confidence.**

[Figure]

**We feel that this sensitivity of the population mean anomaly during September and October of 2014 to cyclones in excess of 30° latitude from Iceland is somewhat unrealistic. We hypothesize that expanding the dispersion modelling to consider more emissions heights, and fluxes, was well as altering the dispersion model's assumptions about its efficiency in removing aerosol via precipitation will allow us to better understand the robustness of our results. This excessively complicated the analysis in the paper and was not really feasible in the timeline of review.**

**We would like to reiterate: our removal of the Holuhraun case study was not due to the discovery of conflicting evidence- we just felt that we could do a better job understanding uncertainty in this case in a future paper using the tools we had developed in this paper. Based on the reviewers' request we have placed all of our tentative analysis of this case in the text and have outlined what simulations need to be done to constrain this case. We hope that this information will allow readers to evaluate the robustness of our results for themselves.**

**We have added this material in section 3.7 and described the plume modelling in section 2.3.2.**

Moreover, the aerosol effect on cyclone albedo that the current study suggests should not be called substantial. Substantial differences in the LWP are in my view rather associated with meteorological variability as otherwise the analysis would not need to be classified by WCBs (e.g. Fig. 2). Now that the methods are described in more detail, there seem to be additional problems with the data analysis that I also briefly comment on in the following.

**Substantial and dominant are not the same thing. We stated that meteorology dominates cyclone variability. It is hard to see how this could not be the case (see page 18 line 15 of the previous submission). We have tried to add a few more comments into the text making it clear that the variability is dominated by meteorology.**

The used all-sky albedo includes cloudy and clear pixels. Hence the analysis does not only include aerosol-cloud interaction, but also aerosol-radiation interaction that cause increased all-sky albedo. There is therefore no basis to separate the effect of aerosol on cloud albedo from the effect of aerosol on the clear-sky reflectivity. This might be primarily a problem in the cool sector of the cyclone with broken clouds that seem to show the greatest effect (P. 14 14-18). Using the term "Aerosol-cyclone indirect effect" is therefore misleading as it suggests a separation between aerosol-radiation and aerosol-cloud interaction within cyclones.

**Thank you for suggesting this. We used the all-sky albedo because we felt that it was less sensitive to thresholding than the in-cloud albedo. It is fair to note that the difference in albedo between the high and low CDNC cyclone populations could be contributed to by aerosol direct effects. We have added text to this effect to the manuscript. We have also provided analysis of the clear-sky albedo difference between cyclone populations. We find that the clear-sky difference was on the order of 0.005±0.001. The difference in all-sky albedo was nearly an order of magnitude larger (0.032-0.018). The effect of this clear-sky difference in albedo averaged in cloudy and clear regions should be relatively small compared differences in albedo due to changes in cloud properties given that midlatitude cyclones tend to have cloud fractions in excess of 70%. Thank you for suggesting this calculation. The following text has been added to the discussion:**

*" In this analysis we have used all-sky albedo from CERES to examine the response of cyclone albedo to changes in CDNC. This variable was chosen because it does not impose a criterion for what it considers to be a cloud when calculating the albedo, as a cloudy-sky albedo would. If we were to use in-cloud albedo this would necessitate the albedo perturbation being restricted to only confidently cloudy pixels (see Marchand et al. (2010)). For example, this could exclude situations where mesoscale cellular convection was occurring as these regions would not necessarily be considered cloudy. As pointed out by previous studies (McCoy et al., 2017c), these clouds may have a significant impact on all-sky albedo. However, use of all-sky albedo may potentially conflate aerosol direct effects and indirect effects.*

*First we provide an estimate of how much cloud fraction differences may contribute to the difference in albedo. Because cloud fraction and albedo have a fairly linear relation in the midlatitudes on a monthly time scale (Bender et al., 2011;Bender et al., 2017) we provide a calculation of the change in albedo related to changes in cloud cover. The observed midlatitude slope of the relation between albedo and fractional cloud cover of 0.4 from Bender et al. (2017) implies a change in albedo between the top and bottom third $CDNC_{SW}$ populations of 0.005-0.007±0.002 (Fig. 10). As shown in Fig. 10, the difference in mean cloud cover between the populations is significant at 95% confidence, but it does not appear to contribute to the majority of the effect on albedo."*

Fig. 1:
CERES albedo shows a dependence on the SZA not only for angles greater than 30 degree. Indeed there the increase in albedo with SZA is strongest, but also for smaller SZA, we can see a dependence, e.g., the same albedo of 0.18 could be associated with a cloud fraction of roughly 80-90%. This might be a problem since the aerosol effects on cloud cover has a magnitude consistent with this uncertainty (Fig. 7). Maybe there is a better way for accounting for the SZA problem than just cutting off the worst part.

**Because this is a 3D radiative transfer problem there is no easy solution to removing SZA dependence. As shown in the supplementary material, the SZA effect tends to conflict with the signal from aerosol-cloud interactions so the limiting of SZA is not creating the signal attributed to aerosol-cloud interaction.**

Specific comments:

P. 6 9-11: "potentially allowing examination of broken cloud cover"
The satellite data of the study is daily averaged that is fairly coarse since extra-tropical cyclones are mobile such that the mean smoothes the quantities under investigation, e.g., CDNC and cloud cover.
**This was unclear. What was meant was that if we had used only in-cloud retrievals of albedo we would throw out all the data from broken cloud because these do not necessarily meet the criterion required to perform a cloud retrieval (for example see Marchand et al. (2010)). By using all-sky albedo this allows us to consider contributions from sub-pixel clouds and clouds that are not surrounded by cloudy pixels. This has been expanded on in the text to make it clear that we are only considering their contribution to their albedo, we aren't picking them out.**

P. 8 19- 28: "should be thought of in the context of a 'fixed-CDNC' set of experiments"
Given the way CDNC is calculated in the simulations, CDNC is by no means fixed, but rather artificially constrained.
**The wording has been changed. Thanks.**

P. 16 8-9: "To calculate the difference in terms of a radiative flux the difference in albedo is multiplied by the annual mean climatological downwelling SW"
I like the idea, but think the estimate is rather crude. Maybe rephrase it and rather speak of typical fluxes one would expect from certain values of downwelling SW fluxes. One such value could than be the climatological mean of a certain geographical position, but I nevertheless perceive such arbitrary estimates as rather uncertain and would not rely on them. That being said, it is misleading to have these values as a key result in the conclusions without further explanation or uncertainty estimate (P. 19 1-4).
**This is a really useful insight - thanks. The way we calculated it was meant to be somewhat agnostic as to when and where the cyclones occur.  The more relevant quantity in this regard is the albedo, but we felt that the albedo is a less intuitive quantity for a broad audience.  We have changed the conclusions and abstract to state the albedo, as opposed to estimated change in SW and added discussion of this calculation to make it clear that this**

**assumes that the cyclones are randomly distributed between 30-80° throughout the year. We note that, for example, the seasonal cycle of DMS emissions in the Southern Ocean means that cyclone CDNC is highest during the period of strongest downwelling SW, so our calculation underestimates the change in SW because it ignores this source of covariability between insolation and CDNC. We now discuss that this is a quick calculation to contextualize the albedo difference and provide the difference in albedo first as you suggest. Thanks.**

*" To calculate the difference in terms of a radiative flux the difference in albedo is multiplied by the annual mean climatological downwelling SW associated with the CERES EBAF-TOA data set between 30°-80°. It is important to note that this assumes that the cyclones being affected are randomly distributed in latitude and during the year. This may be somewhat reasonable for anthropogenic pollution, but not for biogenic aerosol sources. Specifically, planktonic sulfur sources have a substantial seasonal cycle leading to their contribution in albedo occurring during the period of maximum insolation (McCoy et al., 2015). The difference in reflected SW provided here is only intended to act as a rough guide to contextualize the change in albedo."*

P.19 5-8:
And how does the estimate compare to these other studies?

**The page and line number correspond to the acknowledgements. We infer based on the content in the previous comment that this references the difference in SW. It is difficult to offer a direct comparison to other studies as they do not offer a comparable estimate of forcing from CDNC changes in the midlatitudes. We have tried to expand on comparisons between the present study and previous studies.**

**Report #2**
The authors made major changes to this manuscript, focusing it and dropping some of the analysis (i.e Volcanic simulations) as requested by the reviewers. This is a much better presentation. I am not convinced that the authors have fully responded to the reviewers and concerns about the type of model used for this study, but this paper is getting there.

I tried to look at the whole manuscript again, since the focus changed. And it STILL needs some substantial revisions.

The paper is better. I think the abstract still feels disjointed. The paper does not know whether it is a modeling study or an observation study. The problem is that the 'observations' rest on a reanalysis model, and the key 'microphysical' parameter is just a passive tracer of human emissions in that model. Again, it feels like something slapped together from a few projects. But it's one less project now (only 2), so that's fine.

**We appreciate the reviewer thoroughly re-evaluating our manuscript. As discussed extensively in the previous round of revisions the observations do not rely on MERRA2 reanalysis. The results are the same when using MODIS CDNC observations. That is to say, we could remove MERRA2 from the paper and the conclusions would be the same. We have provided supplementary analysis using MERRA2 proxy CDNC it because we are**

**concerned that there may be some residual remote-sensing biases in MODIS CDNC. The only reason we have shown results from MERRA2 is that it is not going to have retrieval error. Ultimately, the *two* estimates of CDNC (reanalysis and observations) result in the same conclusions. We have tried to further clarify that there are two CDNC estimates being used in this paper and to clarify that we are doing that because, independently they may suffer from modelling error (for MERRA2) or from retrieval bias (MODIS), but if they produce the same results then these results are at least not a function of either of these errors.**

I think it needs better organization of the results and separation between observations and models. Significantly, the manuscript should decide the order of presenting results. E.g.: section 3.1 is called:

3.1 Observed relations between meteorology, mid-latitude cyclones, and aerosol

But then immediately:

3.1.1 Aquaplanet simulated mid-latitude cyclones and their response to changes in CCN

Put this in a more logical order please. It should perhaps flow from observations to model simulations. The figures could be kept in a similar way, or separated, but I don't think the present treatment works.

**We have reorganized. Thank you. We found it very difficult to organize it as purely observations and then purely model, so we have organized things to go in parallel so we compare modeling and observations for each step of the analysis. We have expanded the number of sections in the discussion to make it easier to follow the analysis.**

Also: with the narrower focus, there should not be a need for supplementary material, and this should be brought forward.

**We feel that it is good to use the SM to examine the result of various sensitivity studies that are effectively just the same figures as the main text, but repeated with slightly different parameters. These could equally just be 'the calculation is not sensitive to X (not shown)', but it seems better to just show it in the SM.**

A few broad notes:

1. I think what I fundamentally have a problem with is applying sophisticated tools like a high resolution model in coarse ways (aquaplanet, specified and simplified aerosols and microphysics). And then saying that is what climate change will look like. I think this is valid for the WCB analysis and that partitioning is fine and interesting, but I think it is difficult to then compare that to the real world.
**We are only using the model to understand how to interpret the observations. In this case the aquaplanet is used to justify the WCB partitioning, and the idea that the CLWP should**

**adjust at a given WCB in response to changes in CDNC. Based on your suggestion we have tried to clarify that our closing statements are based on the combined evidence of observations (which cannot show causality) and simplified modelling (which does not fully capture all the processes in the real world). We describe the pros and cons of using a convection-permitting model for this type of simulation and feel that the use of a high-resolution model allows us to do a better job resolving relevant processes than a traditional GCM.**

2. Perhaps it needs to be stated that most of what is going on in mid-latitude cyclones is resolved at the scale of the model. That's not really stated, and would probably be scientifically defensible. But it does NOT look that way from the model results: convection matters, because the answers change.

**We have discussed how the simulations at resolutions from 1-16km compare as presented in Field et al. (2017). In the discussion comparing the cyclones in the simulations and in the observations we show several dissimilarities. The model grid will not be saturated some of the time when there should be a cloud. Because we don't have a cloud scheme we will never have a cloud in these cases.**

From Figure 2 and Figure 6: the MERRA2 and LR results look similar, except the Low and High CDNC have opposite effects. The high resolution aquaplanet does look like observations, but this is fundamentally different than low resolution.

**That's right. Based on this analysis MERRA2 isn't building in the CDNC effect. It is true that both the LR and MERRA2 cyclones have a much lower sensitivity to WCB moisture flux. However, we do find that even the low-resolution UM simulation has a qualitatively similar response to increased CDNC at a fixed rain rate/WCB as the high-resolution model does.**

3. Trying to pull parts of a reanalysis (like using sulfate as a proxy for CDNC) is dangerous. Not only are uncertainties in the data swept under the rug, but the reanalysis is also then being pulled in different directions. I know this has been published before and 'it works', but I think that really brushes away a lot of processes.

4. The WCB part is useful, except to the extent that ice and ice indirect effects is not really treated. But I think trying to make inferences then about the real atmosphere from the coarse analysis of the MERRA data is very dangerous.

**Please see comment above regarding the presentation of MERRA2-inferred CDNC and MODIS-observed CDNC in parallel.**

5. The analysis takes too 'high level' a view and does not dive deep enough into what is going on in the model. What is going on to make the LR and HR simulations respond differently? You imply it is convection. But you could show that in the model, by figuring out where convection is in the cyclones, and relating that to the changes in Figure 6.

**This is beyond the scope of the present analysis. Ultimately, if we just used the LR model or the HR model it would not affect our interpretation of the observations. Based on your**

comment we have rewritten the statement that differences between LR and HR are related to convection. There are several substantial differences between the LR and HR simulations- not just convection. Thanks.

Some further detailed notes:

P5, L15, "MERRA2 sulfate mass is a good predictor of CDNC as observed by MODIS". You say no cross talk: but if aerosols and clouds are convolved in MODIS, then there is cross talk because MODIS aerosol is assimilated to get SO4?

**This is discussed at length in our papers (McCoy et al., 2017a;McCoy et al., 2017b), thus we did not replicate the discussion here and just referenced the papers. Several pieces of evidence indicate that both MERRA2 SO4 and MODIS CDNC are good predictors of the true CDNC. We find this by comparing in-situ results to MODIS CDNC. They compare well, and this has been shown by other authors (Bretherton et al., 2010;Bennartz and Rausch, 2017;Painemal and Zuidema, 2011) in addition to our analysis in (McCoy et al., 2017a). We also find that long-term trends in SO4, which are consistent with long-term trends in observed SO2 emissions, are skillful at predicting long-term trends in CDNC (McCoy et al., 2017a). The MERRA2 aerosol retrieval utilizes AOD to nudge the reanalysis, but this is adjusted for swelling (Bosilovich et al., 2015;McCarty et al., 2016;Chin et al., 2002). There may be residual signal that the correction (Chin et al., 2002) does not account for, but again, this is why we reproduce everything with MODIS observations.**

Figure 2: This is a mess, because the contour intervals are all different. If you make them the same, I think you are going to end up with some wildly different results. I think that makes the analysis and comparisons very difficult.

**We note in the text that the model and observations give different looking cyclones. This is expected. We didn't utilize a cloud scheme because that would require additional tuning. We apologize for the colorbars. The color intervals and ticks should be the same now. Previously we had only set the color range so the ticks were all the same, but python clipped the colorbars in a weird way.**

Figure 3 also seems hard to compare: Aquaplanet, MAC-LWP and MERRA2 seem very different.

**As noted, we expect this. This is discussed in the beginning of section 3.1.2 from the previous round of revisions. We have made sure that both the ticks and range on all the color bars are the same now.**

P12, L17: the problem is that sulfate mass concentration in MERRA may not interact appropriately with clouds (e.g., incorporation in cloud drops, scavenging, etc) if there are no indirect effects.

**Please see comment above regarding the presentation of MERRA2-inferred CDNC and MODIS-observed CDNC in parallel.**

P15, L10: The albedo discussion here is not convincing. The model (Figure 9) looks quite different than the other analysis in Figure 8.

**We discuss that the albedo difference in the simulations is not very similar to the albedo difference in cyclone populations. We have expanded the discussion of the difference in albedo between the cyclone populations in Section 3.5.**

P15, L20: Im confused by this paragraph, and concerned at the method. The 2nd sentence on line 23 is missing a critical clause. Also: here is another place where you have mixed up observations and simulations in the flow. Perhaps this should be with observations.

**There appears to be some discrepancy with the version on ACPD and the version you are referencing. Line 23 only has one sentence. We have tried to clarify the separation between observations and simulations in the discussion.**

P16, L20: I'm not convinced a regression model works here based on the previous analysis and the different behavior with MERRA.
**Please elaborate on what aspects of the analysis you feel are not suited for a regression model. The coefficients are all significant at 95% confidence. Heuristically, if we look at CLWP as a function of the predictors it seems to depend on both pretty clearly (Fig 11). We are happy to do any other tests of the fitness of the model that you suggest. Also, see comments above regarding use of both MERRA2 and MODIS separately.**

P18, L7: This first conclusion: you come up with the same results of many models that do not match observations (e.g. Mallevele et al 2017). Why should these results with bulk microphysics be any more valid? The processes in your high resolution simulations are the same for stratiform cloud microphysics as in many global models. Why would these simulations not have the same issues?
**Based on the comments of reviewer 1 we have now shown how our results compare to (Malavelle et al., 2017). We have rewritten the conclusions substantially to more clearly express that our result is not strongly dependent on model resolution.**

P18, L18: Please comment on the limits of the regression model.
**We have added confidence intervals to the coefficients in the regression model (see table 2).**

**Report #3**
This revised version of the manuscript explores the impact of aerosols on liquid water path in extratropical cyclones. This new version is more detailed and more convincing that the previous paper, but there are still issues, mostly to do with the presentation of the results and the structure of the paper. The comments below list the various parts were rewriting or modifications are needed to make the paper clearer and more fluid.
Based on the sensitivity of the results to the source of cloud droplet number concentration, the conclusions should be somewhat expanded and in some places toned down.

Otherwise, the manuscript is acceptable for publication after minor but highly recommended revisions.

**We thank the reviewer for their time reading our revised submission. While the CDNC source does seem to have a minor quantitative effect on the results, we found that the change in CDNC (MERRA2 or MODIS) data set did not qualitatively impact the results. One thing that should be noted in the manuscript, and which might explain some of the perceived discrepancy between MERRA2 and MODIS is that because of remote-sensing constraints the MODIS observations are only available when the sun is at low solar zenith angle. This means that MODIS doesn't have observations over the Southern Ocean in winter, which is one of the lowest CCN places on Earth. MERRA2 does have the ability to predict SO4 in high latitude winter so it has a large number of low CCN cyclones. The Southern Ocean in winter in MERRA2 is just based on a climatology of DMS emissions over the remote Southern Ocean, but the mechanism explaining this behavior (no light for phytoplankton in winter and thus only sea spray) is well understood and agrees with numerous other lines of evidence (McCoy et al., 2015;Ayers and Gras, 1991;Boers et al., 1994). However, this low CDNC data that MERRA can infer doesn't disagree with the results from MODIS and only really seems to extend them to very low CDNC.**

Detailed comments:

1. The document needs some cleaning and tightening as the current presentation still includes remnants of the previous version. Section 2.3.1 is the only subsection in section 2.3. Similarly 3.1 is the only subsection of this level in the results section 3.

**Thank you, we have reordered our sections.**

2. In the conclusions, please discuss where the additional variability in cloud liquid water path might come from, as you indicate that 60% (and not 100%) or this variability is caused by the moisture flux and aerosol impact on cloud droplet number concentration.

**We have added additional discussion to this section speculating where the additional variability may originate from.**

*"While we should not expect to explain all of the variability in CLWP no matter how many predictors we use, it is likely that the explained variability in our regression model could be improved by (1) a more skillful metric for moisture flux into the cyclone, (2) a more accurate observation of CDNC_SW, or (3) additional information regarding ice and mixed-phase cloud properties. In regards to point (1): we have chosen to predict moisture flux in this way so that we may observe it utilizing microwave radiometers. In regards to points (2) and (3): we note that both of these retrievals are difficult and are likely to improve as the remote sensing community examines them in more depth. Overall, explaining the majority of extratropical cyclone liquid water path variability utilizing two predictors is a useful contribution to our understanding of the midlatitudes."*

**Thanks.**

3. You might want to relate your results to a recent paper by Naud et al. (2017), who explored the co-variations between aerosol optical depth and cloud cover in extratropical cyclones. In particular they examine cold and warm fronts separately, which might help explain better your assertion that the cold frontal region is where the albedo change with aerosols is concurrent with the change in liquid water path and cloud fraction. Also, while cited, the Grandey et al (2013)

study who explore the impact of extratropical cyclone strength on the cloud-aerosol relationship could be further discussed in light of the results of this paper in the conclusions. Both of these studies to some extent contradict the statement in the introduction that "aerosol-cloud indirect effects have not been observed in extratropical cyclones" (L25, p2). Although neither study can establish a causal relationship, the fact remains that observations in extratropical cyclones have already been used to explore aerosol-cloud relationships.

**Thank you, we were not aware of the recent Naud et al. paper, and it is very relevant. We have expanded the discussion of both papers and removed our statement in the introduction in light of this discussion.**

4. In the introduction please indicate where these cyclones are. In 2.1 it is specified that they are over the oceans but a latitude range should be specified. I suppose that these are all in the northern hemisphere? Are they found in both Atlantic and Pacific oceans? Are the Mediterranean cyclones also included? At least it is specified for the simulations but at the bottom of page 12, it sounds as if the observed cyclones are sampled in both hemispheres. If indeed this is the case, then "southwest quadrant" is misleading.

**This is discussed in the methodology. The cyclones are from 30°-90° over oceans for both hemispheres. Cyclones must have a minimum ocean coverage to be considered, and this removes Mediterranean cyclones which have too much land within the composite (although removing these cyclones didn't change our results qualitatively). As discussed on line 25 page 12, all cyclones are flipped so that north, or up, is poleward. We have added a figure illustrating this. While we recognize that it is potentially confusing to use the subtext SW, we think bottom-left is also confusing. The idea is to identify the WCB region, but sub WCB would be confusing, as would sub CF for cold-front, since no front rotation has been used. We have indicated the averaging region on the schematic in figure 1 that we have added.**

5. Section 2.2.3: please add a couple of sentences to explain how the MERRA-2 sulfate mass is used to obtain CDNC.

**Done, thank you.**

6. Section 3.1.1: the section on the aquaplanet simulations is very short, and about half of this section is in fact about the relation between the moisture flux and precipitation, which is more general than just about model simulations. The main result here seems to be that regardless of resolution and aerosol concentration the relationship between moisture flux and precipitation rate is unchanged. No mention of a relationship between CDNC and liquid water path is made. Both figures associated with this section are in the supplemental material. It would be preferable to have these figures in the paper since they do illustrate the discussion.

**The sections have been altered. Now the first section is on how WCB predicts CLWP. Thanks.**

This would be especially useful because in fact this discussion is confusing: on the one hand, it gives the impression that the rain rate vs moisture flux relationship does not change with the aerosol concentration (or resolution). But Fig. S3 suggest that it does. To bring this matter to rest though, the figure should be clarified by either drawing a linear regression per model configuration or constraining the data points so they would have the same WCB value per

configuration. Another confusing matter is this: if WCB is kept fixed and LWP changes with aerosols, then surely precipitation should as well, no?

**Linear regression for all the models has been added along with a plot of total precipitation rate. Thanks.**

**This is a good point about precipitation. We have tried to add some discussion to clarify this. At a short time scale we might expect precipitation to change within the cyclone in response to a transient change in CDNC, however the precipitation falling out of the cyclone will have to eventually have to match the moisture flux into the cyclone in order for mass to be conserved. To do this the LWP must adjust to a higher value so that rain rate increases. Text has been added at several points in the manuscript to clarify this.**

7. Section 3.1.2: again the title of this section is misleading, it says "observed' cyclone properties and yet the very first sentence is about comparing observations with simulations. It seems that pages 9-11 are in fact part of a single section on the simulations while a new section should be when the work using the observations alone starts (second paragraph p12)
**The sections have been reorganized, thanks.**

8. Are the three WCB regimes of Figure 2 defined based on the observations? Two questions arise: are the three population very different in the number of members and would sampling issues affect the results? Are the distribution of WCB per region for observations and the different model configurations very different? Why not use the same color scale for all composites in Figure 2?
**The placement of these regimes was defined in Fig 4a. We chose these regimes because the cyclone structure was very different in these three regimes so visually it seemed more interesting to the reader. We noted the number of members in each regime in the subplots. You are correct that terciles are a more natural binning. We changed the bins accordingly.**

**Apologies about the colorbars on figure 2. We set the color limits, but not the contour limits so the colorbars have the same range, but they only showed some of the ticks. We have corrected this now.**

9. The discussion on why the southwest quadrant is a good place to sample for CDNC could be improved. First this quadrant is dominated by low-level clouds and so MODIS derived CDNC is probably better sampled there. It is not clear however that it would be representative of the entire cyclone, and in particular the warm sector and warm frontal zone that are dominated by high-level clouds and thus CDNC information is missing. Second the warm conveyor belt which is ingesting moisture into the cyclone tends to originate from the south east and is not always found in the southwest quadrant. So it is not clear that the "southwest quadrant is likely to be the source of moisture and aerosol for the cyclone". Figure S6 is quite important for this discussion and yet once again it is in the supplemental material. The whole discussion on how to best partition the cyclone population based on CDNC needs to be improved, it is quite confusing still. For example, it is unclear what Figure 4 is really telling us.
**Figure 4 is just there to help visualize the number and distribution of cyclones and where the different CDNC$_{SW}$ and WCB populations are demarcated. It's not a key figure, but it seemed helpful for readers to orient themselves.**

**We have moved figure S6 into the main text.**

**We acknowledge that the conveyor belt does not always originate in the south-west (poleward oriented), but this appears to be the case in the majority of cases and in the literature (Eckhardt et al., 2004;Naud et al., 2012). We have shown the sensitivity of our analysis to this assumption in the SM. This is one possible explanation of why our regression model doesn't have a higher explained variance, and we have included this discussion in the MS. We can also see that SO4 is higher in this quadrant, indicating that the aerosol ingestion predicted by MERRA2 is in this quadrant. As you suggest, moving this to the main text would be useful.**

10. How significant is the separation in Figures 3a and 3b? based on this figure and the tests presented in Figs S7, S8, and S9, it seems that the separation is best for the southwest quadrant possibly because this is where low-level clouds dominate and ice contamination in the satellite observations is less. It seems that the results really target this specific quadrant and that little is known of the clouds that are found in other parts of the cyclones.
**The populations have overlapping standard deviations as one might expect (shaded area), but the means of the populations are different at 95% confidence based on the standard error in the mean (solid lines) and a normal distribution. We note that this result is not sensitive to whether the satellite CDNC is being used or if the inferred reanalysis CDNC (which doesn't have ice cloud contamination) is used. This is now stated more clearly. Thank you.**

11. Discussion of Figure 7: here it might be worth comparing with the results of Naud et al 2017 (fig 10). Also, why not define the three WCB regimes based on terciles of the entire cyclone population? This would alleviate the small number of member issue for the 5 mm/day category?
**Our results seem very consistent with the results in (Naud et al., 2017). Thank you for bringing this paper to our attention. We have added comparison with this paper. We have repeated the analysis using equal terciles.**

12. Page 16, discussion on albedo effects: it is quite worrying that the difference between the MODIS and MERRA2 constrained albedo variations with aerosol are this important for low values of WCB which constitute the largest number of cyclones. This should probably be said somewhere. Unrelated: the whole discussion on albedo could probably be presented in its own subsection.
**The albedo discussion has been moved to its own section. Thank you for this suggestion. Use of terciles does make the albedo difference quite a bit more pronounced when MODIS CDNC is used- it is still only really present in the top tercile of WCB moisture flux, which is interesting. However, we show that for WCB moisture flux in excess of 3 mm/day the two populations clearly diverge in respect to the cyclone-mean albedo. We speculate that this may reflect difficulties in the representation of cyclone dynamics for low moisture flux cyclones by MERRA2. MODIS is not affected by this issue.**

13. I do not see how Figure 11 is showing a "stronger increase in CLWP for a given increase in CDNC_SW in more pristine storms". Either there is an error in the sentence or the caption of Figure 11 needs rewriting.
**We just meant that the white lines bend more at low CDNC$_{SW}$. We have tried to clarify this in the text.**

14. Conclusions: last sentence. Given the observations at your disposal and the disagreements between MODIS and MERRA-2 constrained relations, I would tone down this last sentence, as I am not convinced that this is a demonstration, but rather the observation of co-variations in accordance with the expectation of the sort of effect aerosols should have (as demonstrated with the simulations though).
**There is relatively minor disagreement between the relations predicted by MODIS and MERRA2 (as in Figure 3). However, we have added discussion to clarify our reasoning and make it clear that we are making a statement based on empirical analysis of observations. We find that similar results can be replicated based on an idealized model and have a theory that we think explains the mechanism, but ultimately we can't show causality as in a lab study.**

Typos:
Line 21, p 3: replaced "is" with "area" between "composite" and "located"
Line 17, p9: write what "CMIP" stands for.
Line 16, p14: "extent" is misleading as this is a term often used for vertical extent. "fraction" or "cover" might be more appropriate.
Line 5, p15: add "s" to "support"
Line 21-24: this sentence is too long and is missing a verb towards the end.
Line 13, p16: replace "can be" by "can display" for example
**Thank you for your careful reading of our manuscript and your comments. These typos have been fixed.**

Andreae, M. O., Jones, C. D., and Cox, P. M.: Strong present-day aerosol cooling implies a hot future, Nature, 435, 1187, 2005.
Ayers, G. P., and Gras, J. L.: Seasonal relationship between cloud condensation nuclei and aerosol methanesulphonate in marine air, Nature, 353, 834-835, 1991.
Bennartz, R., and Rausch, J.: Global and regional estimates of warm cloud droplet number concentration based on 13 years of AQUA-MODIS observations, Atmos. Chem. Phys. Discuss., 2017, 1-32, 10.5194/acp-2016-1130, 2017.
Boers, R., Ayers, G. P., and Gras, J. L.: COHERENCE BETWEEN SEASONAL-VARIATION IN SATELLITE-DERIVED CLOUD OPTICAL DEPTH AND BOUNDARY-LAYER CCN CONCENTRATIONS AT A MIDLATITUDE SOUTHERN-HEMISPHERE STATION, Tellus Series B-Chemical and Physical Meteorology, 46, 123-131, 10.1034/j.1600-0889.1994.t01-1-00004.x, 1994.
Bosilovich, M., Akella, S., Coy, L., Cullather, R., Draper, C., and Gelaro, R.: MERRA-2. Initial evaluation of the climate, Tech. Rep. Ser., Global Modeling and Data Assimilation, RD Koster, ed., NASA/TM-2015-104606, 2015.
Bretherton, C. S., Wood, R., George, R. C., Leon, D., Allen, G., and Zheng, X.: Southeast Pacific stratocumulus clouds, precipitation and boundary layer structure sampled along 20° S

during VOCALS-REx, Atmos. Chem. Phys., 10, 10639-10654, 10.5194/acp-10-10639-2010, 2010.

Chin, M., Ginoux, P., Kinne, S., Torres, O., Holben, B. N., Duncan, B. N., Martin, R. V., Logan, J. A., Higurashi, A., and Nakajima, T.: Tropospheric Aerosol Optical Thickness from the GOCART Model and Comparisons with Satellite and Sun Photometer Measurements, Journal of the Atmospheric Sciences, 59, 461-483, 10.1175/1520-0469(2002)059<0461:TAOTFT>2.0.CO;2, 2002.

[revised manuscript text omitted]

First, weWe examine how our key predictors of cyclones behavior, $CDNC_{SW}$ and WCB moisture flux, differed inin September and October of 2014 in thethe vicinity of Iceland (50°W-30°E and 45°N-85°N) and how September and October 2014 differed from the climatological behavior of cyclones in this region from their climatological behavior. This region is consistent with previous modelling of trajectories originating at the Holuhraun fissure over the course of 48 hours (McCoy and Hartmann, 2015). The climatological September-October WCB moisture flux and $CDNC_{SW}$ in is shown in Fig. S14, and Fig. S15. This is compared to the population mean behavior of cyclones during September and October of 2014 in the region of Iceland. This shows relatively small increases in $CDNC_{SW}$. However, notNot every cyclone in this region interacted with sulfurthe sulfate aerosol plumes from Holuhraun. To restrict the cyclone population to cyclones that might have been affected by the volcanic sulfate plume the NAME

dispersion model was used to simulate the dispersion of both $SO_2$ and sulfate aerosol from Holuhraun. The average near-surface volcanic sulfate aerosol mass predicted by NAME was calculated in the southwest quadrant of cyclones during September and October of 2014. Near-surface Sulfate aerosol mass concentrations in excess of 0.1 $\mu g/m^3$ were considered to indicate that a cyclone had interacted with the plume. ~~The WCB moisture flux and CDNC$_{SW}$ observed during 2014 is shown in Fig. S14 and S15 for emissions beginning at 0-1500m and 1500-3000m. In the context of the dependence of CLWP on CDNC$_{SW}$ and WCB moisture flux inferred from global observations, difference in CDNC$_{SW}$ during 2014 relative to the climatological CDNC$_{SW}$ did not infer a massive change in CLWP. This is consistent with the small change in in-cloud LWP observed by Malavelle 
[revised manuscript text omitted]
 found a consistent prediction of enhanced enhancement in cloud cover with with enhanced aerosol column optical depthstudies. These regime-sorted analyseis agree with global analysis in Gryspeerdt et al. (2016), which inferred that enhanced CCN enhanced CF in the midlatitudes. We also note that our statement that enhanced CDNC, driven by aerosol emissions, should enhance CLWP, CF, and albedo in cyclones appears to be in contradiction to the multi-pronged analysis conducted by

10   Malavelle et al. (2017), which showed little response in LWP to a transient volcanic emission of sulfur from the 2014-2015 eruption of Holuhraun in Iceland. However, we find that this small change in LWP is not inconsistent with the dependence of CLWP inferred in our study from 13 years of global dataMalavelle et al. (2017). We performed dispersion modellingmodel simulations of the volcanic plumesulfate aerosol to determine which cyclone systems had interacted with the plumewere 
[revised manuscript text omitted]

---

## Author Response (AR3)

**Response to reviewer comments: acp-2017-649**
**D.T.McCoy@leeds.ac.uk**

**We thank the editor for their help improving our paper and thank them, and the reviewers for their time and effort in carefully reading our paper. Responses are in bold font.**
Abstract:
"When CDNC is increased in the simulations, the CLWP consistent with a given WCB moisture flux increases." This is a key point of this paper but the wording here is not clear, especially since this is the first time it is introduced. I urge you to reword it along the lines used in the conclusion: "When CDNC is increased a larger LWP is needed to give the same rain rate. The LWP adjusts to allow the rain rate to be equal to the moisture flux into the cyclone along the warm conveyor belt."
**Thank you- that is clearer and it is the key result shown in this paper. We have reworded as suggested.**

Pg 4: "Cyclone centers" (not Cyclones centers)
**Corrected.**

Pg 5: "crosstalk" is technical. Please avoid this terminology and just state what the issue is.
**Agreed- we have altered the section to read:**
> *One possible caveat in our analysis is that the radiative signal used to retrieve LWP may partly arise from upwelling radiation due to wind roughening of the ocean surface or emission from WVP. In such cases, LWP is biased in one direction, while wind and/or WVP may be biased in an opposite direction (Elsaesser et al., 2017). However, retrievals of WVP and wind speed have been shown to be unbiased relative to in situ observations and thus such issues are likely minimal (Mears et al., 2001;Wentz, 2015;Trenberth et al., 2005;Meissner et al., 2001;Elsaesser et al., 2017).*

Pg 6: The power law form of the relationship between Nd and sulfate is clearer.
**We have changed the equation to be formatted as in Boucher and Lohmann (1995). Thank you- that is clearer.**

Pg 10: The discussion of fixed Nd is unnecessarily complicated and needs some work. Please state more suscinctly that you are assuming replenishment = removal and discuss implications (e.g. what are the conditions under which it is more likely to be true. You could back this up with some simple calculations along the lines of Wood (doi:10.1029/2012JD018305).
**We have tried to add some additional discussion to clarify the model set up and that no processing is being done so we are assuming that replenishment of removed aerosol is instantaneous. As in Wood et al. (2012) we assume a time-invariant aerosol concentration- and this does seem to work in the regions examined in that study- this is the case over long time scales, otherwise we would have no aerosol or would be choking, but this doesn't work when the depletion is quick. We have added some text to note that this set up is extremely simplified and will not address some features of the cyclone (frontal rain bands for example), which will add variability to the cyclone structure.**

Pg 11: The discussion of "dynamical feedback" is out of place. First it's not even defined,

leaving the reader to wonder what you are referring to. Second, one cannot expect coarse resolution simulations that use saturation adjustment to represent these effects. Please streamline this discussion and remove unnecessary references.

**Thank you, we have clarified this and shortened the statement to read: "Lastly, the evaporation-entrainment feedback on aerosol-cloud interactions (Hill et al., 2009;Xue and Feingold, 2006;Xue et al., 2008) is not well represented in these simulations due to model vertical grid resolution and boundary layer treatment."**

Pg 12: "as it should be a consequence of mass conservation". This is a statement that could be contracted to clarify the point.
Why not: "This consistency across models of varying spatial resolution and observations of real-world cyclones seems reasonable because once in equilibrium, the water mass flux that goes into the cyclone must be precipitated out." (or similar)
**Thank you- this is clearer and has been inserted.**

Pg 17 and 18 (and other instances): MODIS does not "observe" Nd; rather MODIS measurements of tau and r_e are used to "retrieve" Nd.
**That is a good point. The jargon surrounding satellite products is confusing. We have changed instances of 'observed' to 'retrieved' as you suggest because it clarifies what MODIS is doing.**

Pg 19: I'm still not convinced that CASIM doesn't "bake in" positive LWP responses to Nd at the grid spacings employed in this model. I do agree that CASIM should be able to represent other responses at higher resolution. See, e.g., recent NICAM results by Y. Satoh et al.
**This is fair. We agree that if we examine a precipitating grid box with fixed liquid water content and increase the CDNC then the precipitation will be inhibited and on the next time step there will be more liquid. However, the subsequent evolution of the cloud system can be complex. In response to this comment we have added discussion to clarify that for a given precipitating grid box an increase in CDNC will equate to more liquid on the next time step and that because of the way CASIM is formulated and the resolution we are using we might miss out on some of the feedbacks that might lead to a less intense LWP response to CDNC:**

> *In the context of the CASIM cloud scheme used here we note that an increase in LWP in response to CDNC is guaranteed for a precipitating grid box (all else being equal). That is to say, if we examine a precipitating grid box in the model with a given liquid water content and instantaneously increase the CDNC, on the subsequent time step the grid box will have increased its liquid content because precipitation will be inhibited. If there is no precipitation the liquid content will remain unchanged. This is a common feature of warm clouds in models(Hill et al., 2015), and appears in the LWP response simulated by higher horizontal resolution instances of the CASIM model(Grosvenor et al., 2017). While some LWP reduction effects such as evaporation entrainment will not be as efficacious in CASIM due to vertical grid resolution and boundary layer treatment, Miltenberger et al. (2018) showed, using CASIM, that the subsequent evolution of the clouds in the context of a realistic forcing may yield decreased LWP in response to increased CDNC through interaction with the environment and between*

*clouds. In summary, CASIM's vertical resolution and boundary layer treatment make it less likely that mechanisms such as the evaporation-entrainment feedback will be as efficacious and the LWP response to enhanced CDNC might be less pronounced in a different model that is able to capture these effects.*

Pg 21: 3.4 "Differences in the structure of clouds within cyclones as a function of Nd"
**Corrected.**

Pg 26 (top) "position… is reversed" or "positions… are reversed"
**Corrected.**

Pg 26: "as would a cloudy sky albedo"
**Corrected**

Pg 28 (and elsewhere) I think you could be more quantitative than "majority". Why not "up to two thirds"?
**That makes sense, corrected.**

Pg 30: "anomalous at the 95% confidence level" (and anomalous in what regard?
**Corrected.**

Pg 30: "While evocative…" evocative in what regard? Increasing LWP? Please be clear.
**Expanded**.

Pg 31, line 3: If you mean that "no longer being different at the 95% confidence level" means "no aerosol effect on LWP" then please be clear.
**Expanded.**

Pg 31: "Overall, we find agreement with the results presented in Malavelle et al. (2017) in that models [that] strongly adjust LWP in response to this eruption are likely to over-predict aerosol-cloud adjustments."
This sentence states the obvious.
**Fair enough. Removed.**
Pg 32 and abstract. I think this is the key message of this paper (at higher aerosol a larger LWP is required to allow the rain rate to equal the moisture flux) and that you can do a better job conveying this.
**Thank you- we have rewritten our conclusions and abstract statements as suggested.**
Pg 33: There are still quite a few grammatical lapses that one might hope the technical editors will pick up, but in lieu of that, a careful read would be helpful. Language can definitely be tightened in various places.
**We have tried to read through the paper as thoroughly as possible and correct grammatical errors.**

**We again thank the editor for their time in reading through our paper and all their help.**

**References:**

[revised manuscript text omitted]